# Simultaneous proteome localization and turnover analysis reveals spatiotemporal features of protein homeostasis disruptions

Jordan Currie[1], Vyshnavi Manda[1], Sean K. Robinson [1], Celine Lai[2], Vertica Agnihotri[3], Veronica Hidalgo[1], R. W. Ludwig[1], Kai Zhang [4], Jay Pavelka[1], Zhao V. Wang [4], June-Wha Rhee[3], Maggie P. Y. Lam [1,5,6] & Edward Lau [1,6] ✉

The spatial and temporal distributions of proteins are critical to protein function, but cannot be directly assessed by measuring protein bundance. Here we describe a mass spectrometry-based proteomics strategy, Simultaneous Proteome Localization and Turnover (SPLAT), to measure concurrently protein turnover rates and subcellular localization in the same experiment. Applying the method, we find that unfolded protein response (UPR) has different effects on protein turnover dependent on their subcellular location in human AC16 cells, with proteome-wide slowdown but acceleration among stress response proteins in the ER and Golgi. In parallel, UPR triggers broad differential localization of proteins including RNA-binding proteins and amino acid transporters. Moreover, we observe newly synthesized proteins including EGFR that show a differential localization under stress than the existing protein pools, reminiscent of protein trafficking disruptions. We next applied SPLAT to an induced pluripotent stem cell derived cardiomyocyte (iPSC-CM) model of cancer drug cardiotoxicity upon treatment with the proteasome inhibitor carfilzomib. Paradoxically, carfilzomib has little effect on global average protein half-life, but may instead selectively disrupt sarcomere protein homeostasis. This study provides a view into the interactions of protein spatial and temporal dynamics and demonstrates a method to examine protein homeostasis regulations in stress and drug response.

Protein turnover is an important cellular process that maintains the quality and quantity of protein pools in homeostasis, and involves fine regulations of the rates of synthesis and degradation of individual proteins. A close relationship exists between turnover kinetics with the spatial distribution of proteins. Cellular organelles including the cytosol, endoplasmic reticulum (ER), and mitochondria are equipped with distinct quality control and proteolytic mechanisms that maintain protein folding and regulate protein degradation in a localization-dependent manner[1–3]. Newly synthesized proteins need to be properly folded and trafficked to their intended subcellular localization through subcellular targeting and sorting mechanisms, whose capacity has to be coordinated to match the rate of protein synthesis[4–6]. A mismatch between temporal synthesis rate and spatial localization capacity can lead to ER stress and subsequently mislocalization of newly

[1]Department of Medicine, University of Colorado School of Medicine, Aurora, CO 80045, USA. [2]Stanford Cardiovascular Institute, Stanford University, Stanford, CA 94305, USA. [3]Department of Medicine, Division of Cardiology, City of Hope Comprehensive Cancer Center, CA 91010 Duarte, USA. [4]Department of Diabetes and Cancer Metabolism, Beckman Research Institute, City of Hope National Medical Center, Duarte, CA 91010, USA. [5]Department of Biochemistry and Molecular Genetics, University of Colorado School of Medicine, Aurora, CO 80045, USA. [6]Consortium for Fibrosis Research and Translation, University of Colorado School of Medicine, Aurora, CO 80045, USA. ✉e-mail: edward.lau@cuanschutz.edu

synthesized proteins[7]. Disruption of protein turnover and homeostasis is broadly implicated in human diseases including cardiomyopathies, cancer, and neurodegeneration[7,8]. In stressed cells, misfolded proteins accumulate and trigger the unfolded protein response (UPR), which signals to suppress protein synthesis and promote protein folding and proteolysis to restore proteostasis. In parallel, UPR invokes a spatial reorganization of the proteome, such as the transient translocation of UPR pathway mediators to the nucleus during acute stress response, and the retrotranslocation of misfolded ER proteins to the cytosol for proteasomal clearance under endoplasmic-reticulum-associated protein degradation (ERAD).

We wonder how the spatial and temporal dynamics of proteins are regulated in conjunction with UPR. Advances in mass spectrometry methods now allow the turnover rate and subcellular localization of proteins to be measured on a large scale. The turnover rate and half-life of proteins can be measured using stable isotope labeling in cells and in intact animals followed by mass spectrometry measurements of isotope signatures and kinetics modeling to derive rate constants[9-12]. Quantitative comparison of turnover rates provides a temporal view into proteostatic regulations and can implicate new pathological signatures and pathways over steady-state mRNA and protein levels[13-15]. In parallel, spatial proteomics methods have allowed increasing power to discern the subcellular localization of proteins on a large scale[16-20]. In recent work using a differential solubility fractionation strategy and mass spectrometry, we observed broad substantial rearrangement of proteins across three subcellular fractions in an acute paraquat challenge model of UPR in the mouse heart, consistent with protein differential localisation being an important layer of proteome regulation under proteostatic stress[21]. Nevertheless, an integrated strategy that can simultaneously measure protein turnover kinetics and spatial information has thus far not been realized.

In this work, we describe an experimental strategy and computational analysis workflow to perform simultaneous proteome localization and turnover (SPLAT) measurements in baseline and stressed cells. SPLAT builds on prior work in spatial and temporal proteome profiling by combining dynamic SILAC labeling, differential ultracentrifugation, TMT labeling, and kinetic modeling to measure changes in the turnover dynamics and subcellular distributions under perturbation within a single experiment and on a proteome scale. Applying SPLAT to human AC16 cardiac cells under thapsigargin- and tunicamycin-induced UPR, we delineate prominent spatiotemporal changes in the proteome, including membrane transporter localization, possible endomembrane trafficking disruption, and stress granule formation. We also applied the method to human induced pluripotent stem cell derived-cardiomyocytes (iPSC-CM), and acquired data suggesting the proteasome inhibitor cancer drug carfilzomib may exert cardiotoxic adverse effects by selectively impairing sarcomeric protein turnover.

## Results
### Simultaneous acquisition of turnover and spatial information using a double labeling strategy
We reason that we can use a hyperplexing strategy to simultaneously encode temporal and spatial protein information through isotope labels in the MS1 and MS2 levels, respectively. Hence, we designed a workflow that combines dynamic SILAC metabolic labeling in cultured cells, with TMT labeling of spatially separated fractions to simultaneously measure new protein synthesis as well as subcellular localization under baseline and perturbation conditions (Fig. 1a). To determine the rate of protein turnover during control, thapsigargin, and tunicamycin conditions, a dynamic SILAC strategy was used to measure the rate of appearance of post-labeling synthesized protein. Briefly, cells were pulsed with a lysine and arginine depleted media supplemented with heavy labeled lysine and arginine concurrently with drug treatment to label post-

treatment synthesized proteins and derive fractional synthesis rates through kinetic modeling.

Upon harvesting, the cells were fractionated to resolve subcellular compartments. We adopted a protein correlation profiling approach. In particular, the LOPIT-DC (Localisation of Organelle Proteins by Isotope Tagging after Differential ultraCentrifugation) method[17] uses sequential ultracentrifugation to enrich different subcellular fractions from the same samples, which facilitates ease of adoption and reproducibility. Briefly, the cells were lysed under gentle conditions and then sequentially pelleted through ultracentrifugation steps, which pellets subcellular fractions based on their sedimentation rate and which is a function of particle mass, shape, and volume. The ultracentrifugation fractions were each subsequently solubilized, and the extracted proteins were digested and further labeled with tandem mass tag (TMT) isobaric stable isotope labels. The acquired mass spectrometry data therefore carries temporal information in the dynamic SILAC tags and spatial information in the TMT channel intensities (Fig. 1b).

To process the double isotope encoded mass spectrometry data, we assembled a custom computational pipeline comprising database search and post-processing, and quantification for dynamic SILAC and TMT data (Fig. 1c). The turnover kinetics information from the dynamic SILAC data is analyzed using a mass spectrometry software tool we previously developed, Riana[11], which integrates the areas-under-curve of mass isotopomers from peptides over a specified retention time window, then performs kinetic curve-fitting to a mono-exponential model to measure the fractional synthesis rates (FSR) of each dynamic SILAC-labeled (K and R containing) peptide. We then used the pyTMT tool[21] to perform TMT label quantification, correct for isotope contamination (Supplementary Table 1), and assign ultracentrifugation fraction abundance to peptides and proteins (see Methods). The data were then used for temporal kinetics summaries using the MS1 encoded information and subcellular localization classification from the MS2 encoded information (Fig. 1d). By separately analyzing heavy and light peptides, the subcellular spatial information of the heavy (new) and light (old) subpools of thousands of proteins can be mapped simultaneously in normal and perturbed cells.

### Protein turnover kinetics regulations under unfolded protein response vary by cellular compartments
We applied SPLAT to identify protein spatiotemporal changes in human AC16 cells under UPR induced by 1 μM thapsigargin for 16 h. Thapsigargin at the dosage and duration used is a common and robust model to induce ER stress and integrated stress response in cardiac and other cell types through the inhibition of sarco/endoplasmic reticulum Ca2 + -ATPase (SERCA). Thapsigargin treatment at 16 h robustly induced known ER stress markers[22] including BiP/HSPA5, HSP90B1, PDIA4 (limma FDR adjusted $P < 0.01$) (Fig. 2a). Three biological replicate SPLAT experiments were carried out for normal and thapsigargin-treated AC16 cells ($n = 3$ each). We analyzed the spatial fractionation patterns of the proteins following ultracentrifugation and TMT labeling, and classified the subcellular localization of proteins using a Bayesian model BANDLE as previously described[23]. A spatial classification model is trained separately for each treatment using a basket of canonical organelle markers (see Methods), which showed clear separation in PC1 and PC2 in each condition (Supplementasry Fig. 1). The ultracentrifugation profiles of each cellular compartment are highly consistent across treatments and replicates (Supplementary Fig. 2). To minimize the potential ratio compression that can result from MS2-based TMT quantification, we employed extensive two-dimensional fractionation and narrow isolation window, and verified that identified MS2 spectra had high precursor ion purity (median purity 92–93%) (Supplementary Fig. 3). We further performed a direct comparison of MS2 and MS3 based quantification on an identical sample (control replicate 2) (Supplementary Fig. 4), which confirmed

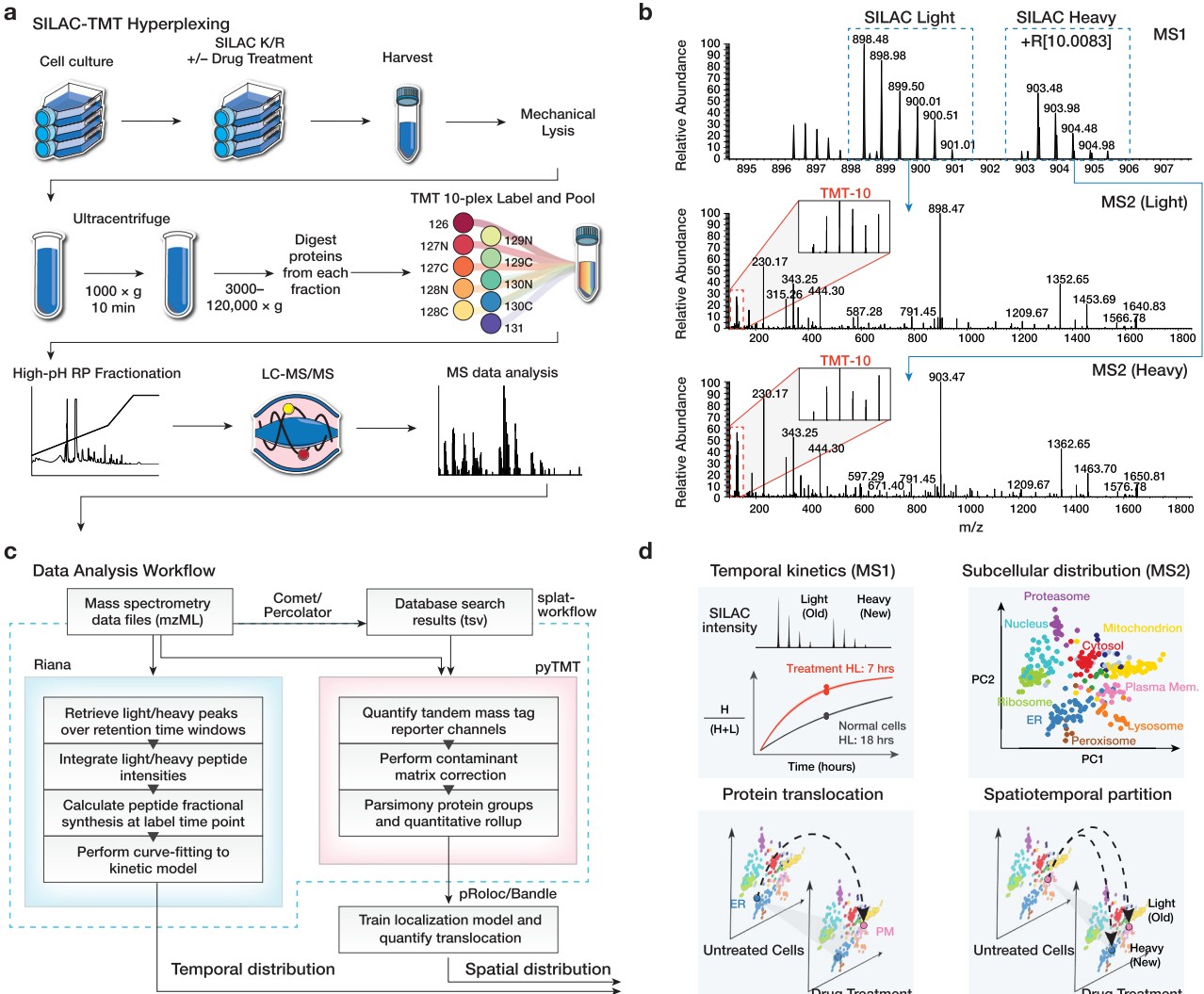

**Fig. 1 | Overview of the SPLAT strategy. a** Experimental workflow. Control, thapsigargin-treated, and tunicamycin-treated human AC16 cardiomyocytes were labeled with $^{13}C_6^{15}N_2$ L-Lysine and $^{13}C_6^{15}N_4$ L-Arginine dynamic SILAC labels. For each condition, 3 biological replicate SPLAT experiments were performed ($n = 3$). After 16 h, cells were harvested and mechanically disrupted, followed by differential ultracentrifugation steps to pellet proteins across cellular compartments. Proteins from the ultracentrifugation fractions were digested and labeled using tandem mass tag (TMT) followed by mass spectrometry. **b** Dynamic SILAC labeling allowed differentiation of pre-existing (unlabeled, i.e., SILAC light) and post-labeling (heavy lysine or arginine, i.e., +R[10.0083]) synthesized peptides at 16 h. The light and heavy peptides were isolated for fragmentation separately to allow the protein sedimentation profiles containing spatial information to be discerned from TMT channel intensities. **c** Computational workflow. Mass spectrometry raw data were converted to mzML format to identify peptides using a database search engine. The mass spectra and identification output were processed using Riana (left) to quantitate the time-dependent change in SILAC labeling intensities and determine the protein half-life, and using pyTMT (right) to extract and correct TMT channel intensities from each light or heavy peptide MS2 spectrum. The TMT data were further processed using pRoloc/Bandle to predict protein subcellular localization via supervised learning. **d** Temporal information and spatial information are resolved in MS1 and MS2 levels, respectively. SPLAT allows the subcellular spatial information of the heavy (new) and light (old) subpools of thousands of proteins to be quantified simultaneously in normal and perturbed cells. HL: Half-life.

that MS2-based quantification produced acceptable spatial resolution, consistent with previous observations[24].

In total using MS2-based TMT quantification, we mapped the subcellular profiles of 4360 protein features (i.e., 1,820 new and 2,540 old proteins) in normal AC16 cells across 3 biological replicate experiments using a stringent two-peptide filter at 1% FDR, with 1946 old proteins and 1,462 new proteins assigned to one of 12 subcellular localization with >95% confidence after removing outliers (see Methods) (Fig. 2b; Supplementary Data 1). The accuracy of the spatial classification is supported by the observation that 69.5% of assigned proteins in normal AC16 cells contain matching cellular component annotation in Gene Ontology despite the current incompleteness of annotations (Fig. 2c) and 71.6% of proteins match their localization annotation in thapsigargin-treated cells (Supplementary Fig. 5). From

the associated SILAC data of the proteins with spatial information, we further quantified and compared the turnover kinetics of 2516 proteins (Supplementary Data 2); hence we were able to acquire proteome-wide spatial and temporal information in matching samples from a single experiment.

Considering the temporal kinetics data, we observed a proteome-wide decrease in fractional synthesis rates under thapsigargin challenge compared with normal cells (median protein half-life 46.9 vs. 19.9 h; Mann-Whitney test $P < 2.2\text{e}{-}16$) (Fig. 2d). This slowdown is consistent with the extensive shutdown in protein translation due to ribosome remodeling under integrated stress response[25,26], shown here by the decreased rate of SILAC incorporation into proteins. Notwithstanding the overall slowdown, we also observed a wide range of protein turnover rates in both the untreated and thapsigargin

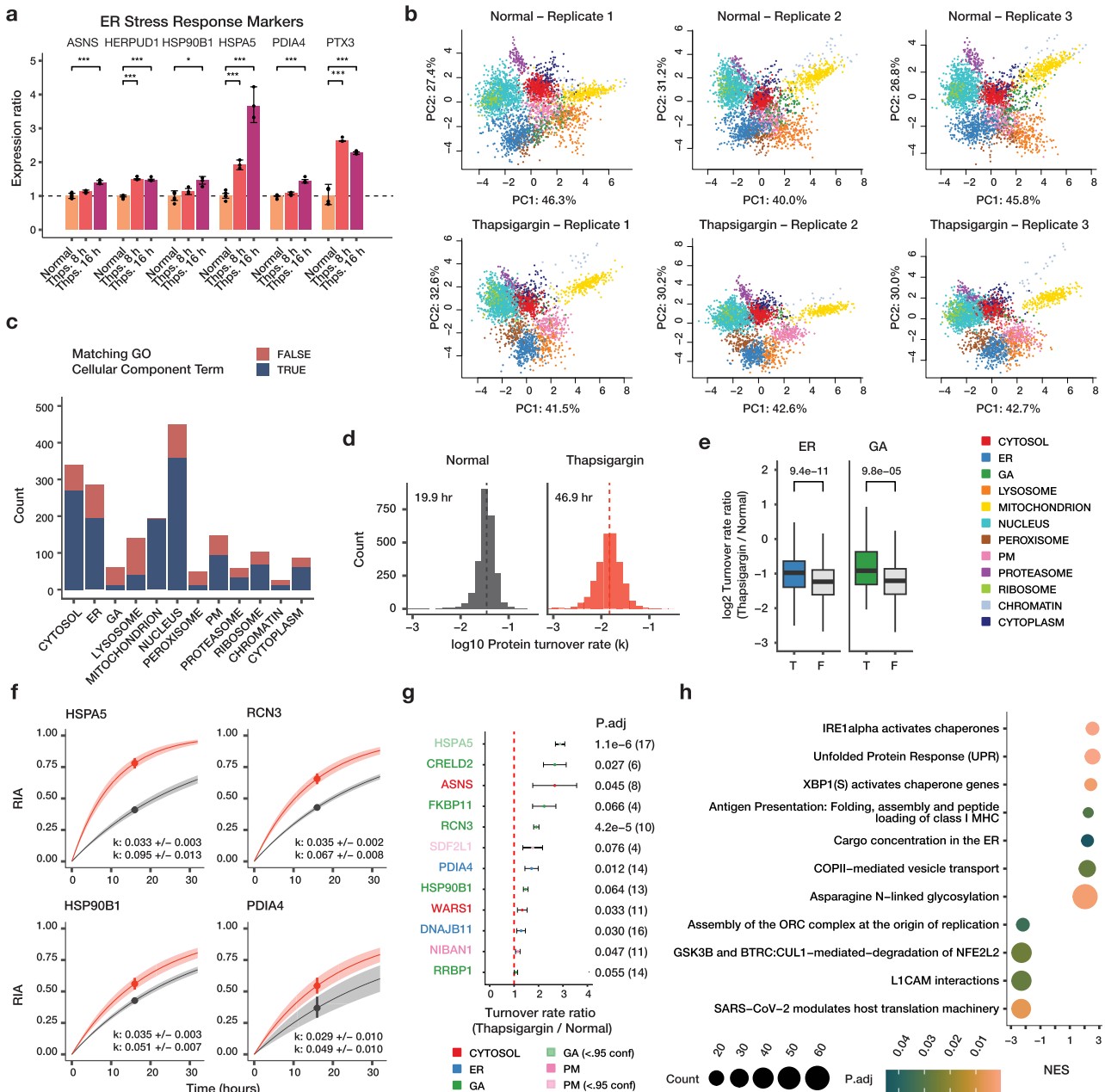

**Fig. 2 | Simultaneous measurements of spatial and temporal kinetics under UPR. a** Bar graphs of expression ratios of known ER stress markers upon 8 and 16 h of thapsigargin (Thps.) ($n = 6$ normal; $n = 3$ thapsigargin). *: $P < 0.01$; **: $P < 0.001$; ***: $P$ d $< 0.0001$; limma multiple-testing corrected (FDR adjusted) P (two-sided). Error bars: expression ratios ± s.d. **b** PC1 and PC2 of proteins spatial map showing the localization of confidently allocated proteins in normal and thapsigargin-treated AC16 cells. Data point: protein; color: subcellular compartment classification. **c** Distribution of light (unlabeled) protein features in each of the 12 subcellular compartments ($n = 3$); fill color represents whether the protein is also annotated to the same subcellular compartment in UniProt Gene Ontology Cellular Component terms. **d** Histograms of determined log10 protein turnover rates in control and thapsigargin cells ($n = 3$). Text overlay: median half-life. **e** Boxplot of log2 turnover rate ratios in thapsigargin /normal cells for proteins localized to the ER (blue) (T) or not (F); or the Golgi (GA; green). P values: Mann–Whitney test (two-sided). A Bonferroni corrected threshold of 0.05/13 is considered significant. Center line: median; box limits: interquartile range; whiskers: 1.5x interquartile range; $n = 286, 2234, 62, 2458$ proteins over 3 independent experiments per group. **f** First-order protein kinetic curves in normal (gray), and thapsigargin treated (red) AC16 cells of 4 known ER stress markers with elevated turnover (HSPA5, RCN3, HSP90B1, PDIA4). Point: best-fit k; line: first-order kinetic curve of k; bands: fitting s.e. **g** Turnover rate ratios (thapsigargin vs. normal) of top proteins with elevated temporal kinetics in UPR. Color: compartment; P.adj: Mann–Whitney test (two-sided) with Benjamini-Hochberg multiple-testing correction. Dashed line: 1:1 ratio; point: median ratio; range: median ± MAD/median of ratios; n=number of peptide observations (parenthesis) over 3 biologically independent samples per group. **h** Gene set enrichment analysis (GSEA) of turnover rate ratios in thapsigargin treatment. Color: multiple-testing corrected (FDR adjusted) P in GSEA (two-sided); x-axis: normalized enrichment score (NES). Size: proteins in the gene set.

treated conditions that differ by the assigned subcellular compartment (Supplementary Fig. 6). Changes in protein kinetics following thapsigargin also varies by compartment, with ER and Golgi proteins having significantly less slowdown of protein kinetics compared to

protein in other compartments (Mann-Whitney test P: 9.4e–11 and 9.8e–5, respectively; « 0.05/13) (Fig. 2e). On an individual protein level, out of the 2516 proteins measured, 1542 showed significant changes in temporal kinetics (Mann-Whitney test, FDR adjusted P value < 0.1), but

the vast majority of these proteins show decreases in turnover as expected, with only 12 proteins showing significant increased temporal kinetics. Among these are the induced ER stress markers BiP/HSPA5, HSP90B1, and PDIA4 (Fig. 2f; Supplementary Data 3) but also other ER and Golgi proteins that may be involved in stress response (Fig. 2g). SDF2L1 (stromal-cell derived factor 2 like 1) is recently described to form a complex with the ER chaperone DNAJB11 to retain it in the ER[27]. In control cells, we found that SDF2L1 has a basal turnover rate of 0.027/hr. Upon thapsigargin treatment, its turnover rate increased to 0.048 /hr (adjusted P: 0.07). DNAJB11 also experienced accelerated kinetics (1.28-fold in thapsigargin, adjusted P 0.029) hence both proteins may be preferentially synthesized during UPR. On a proteome level, gene set enrichment analysis (GSEA) of temporal kinetics changes show a preferential enrichment of proteins in unfolded protein response (FDR adjusted P: 4.1e−4), ER to Golgi anterograde transport (FDR adjusted P 0.036) and N-linked glycosylation (FDR adjusted P: 1.7e−3) but a negative enrichment of translation-related terms (Fig. 2h). Overall, protein kinetic changes are modestly correlated with protein abundance changes (Supplementary Fig. 7), suggesting that AC16 cells actively regulate protein synthesis and degradation kinetics in normal and stressed conditions beyond changes in protein abundance.

## Changes in protein subcellular distribution under ER stress

We next analyzed the spatial proteomics component of the data to find proteins that change in their subcellular localization following thapsigargin treatment. To do so, we used a Bayesian statistical model implemented in the BANDLE package to estimate the differential spatial localization of proteins. In total, we identified 1,306 protein features (687 light and 619 heavy) with differential localization in thapsigargin under a stringent filter of BANDLE differential localization probability > 0.95 with an estimated FDR of 0.0018 (0.18%), and further filtered using a bootstrap differential localization probability of > 0.95. We then further prioritized 330 pairs of differentially localized proteins where the light and heavy features both show confident differential localization (Supplementary Data 2, Supplementary Data 4). The differential localization of these 330 proteins recapitulate previously established relocalization events in cellular stress response, capturing the migration of caveolae toward the mitochondrion under cellular stress[28], and the engagement of EIF3 to ribosomes in EIF3-dependent translation initiation in integrated stress response[29] (Supplementary Fig. 8), thus supporting the confidence of the differential localization assignment.

From the results, we discerned three major categories of differential localization behaviors in ER stress that revealed insights into proteome-wide features of UPR. First, we observed the externalization of proteins toward the plasma membrane (Fig. 3a). The large neutral amino acid transporter component SLC3A2 is localized to the lysosome fraction in normal cells (Pr > 0.999) but in thapsigargin-treated cells is localized confidently to the plasma membrane (BANDLE differential localization probability >0.999) (Fig. 3b). Showing similar behaviors are SLC7A5, the complex interacting partner of SLC3A2; and two other amino acid transporters SLC1A4, and SLC1A5 (Fig. 3c); whereas the ion channel proteins SLC30A1, ATP1B1, ATB1B3 and ATP2B1 also showed confident localization toward the cell surface (Fig. 3c). The change in localization of SLC3A2 is corroborated by immunostaining (Fig. 3d), which shows a decrease in co-localization between immunostaining signals of SLC3A2 and lysosome marker LAMP2 upon thapsigargin treatment (Fig. 3e).

Second, thapsigargin treated AC16 cells are associated with an increase in proteins classified into the peroxisome fraction including proteins whose locations changed from the ER, Golgi, and the nucleus in normal cells (Fig. 3f). In mammalian cells, ER and peroxisomes are spatially adjacent; the peroxisome associated fractions sediment prominently at 5000–9000 × g (F3 and F4) in the LOPIT-DC protocol (Supplementary Fig. 2), marked by canonical peroxisome markers

PEX14 and ACOX1 (Supplementary Data 2). However, although this compartment was trained using peroxisome markers, the majority of proteins categorized into this compartment are not annotated to be in the peroxisome whereas 30 out of 49 (61%) of proteins in the control cells allocated to this compartment were annotated also as endosome, including canonical markers EEA1 and VPS35L. We thus refer to this compartment hereafter as peroxisome/endosome. Moreover, proteins that become differentially localized to this fraction in thapsigargin include known stress granule proteins UBAP2, USP10, CNOT1, CNOT2, CNOT3, CNOT7, CNOT10, ZC3H7A, and NUFIP2 (Fig. 3g and Supplementary Fig. 9), which show high-confidence localization to the peroxisome/endosome fraction, and are known RNA binding proteins that participate in phase separation, consistent with stress granule formation in UPR. Notably, LMAN1, LMAN2, SCYL2, and SNX1 are RNA-binding proteins that are not currently established stress granule components and show identical differential localization patterns, nominating them as potential participants in RNA granule related processes in AC16 cells for further studies (Supplementary Fig. 9). Other proteins in this fraction include the ER-to-Golgi transport vesicle proteins GOLT1B, GOSR2, RER1, and NAPA (Supplementary Fig. 9).

Lastly, we see evidence of proteins from the ER and Golgi targeted to the lysosome (see Tunicamycin section below). Thus taken together, the spatial proteomics component of the data reveals a complex network of changes in protein spatial distribution during UPR.

## Partition of newly synthesized and pre-existing protein pools

We next considered the interconnectivity of temporal and spatial dynamics, namely whether some localization changes are contingent upon protein pool lifetime, such as where the light (old) protein does not change in spatial distribution but the heavy (new) protein displays differential localization upon UPR. Because the spatial profiles of the light and heavy proteins are acquired independently, this experimental design allowed us to examine whether old and new proteins are localized to identical cellular locales. In both normal and thapsigargin-treated cells, we found that the independently measured spatial profiles of light (pre-existing) proteins and their corresponding heavy SILAC (newly synthesized) counterparts are highly concordant, with a normalized spatial distribution distance (see Supplementary Methods) of 0.020 [0.015–0.030], compared to 0.117 [0.080–0.155] in random pairs of pre-existing proteins (1614 light-heavy pairs, Mann–Whitney $P < 2.2e−16$) in normal cells, and 0.028 [0.019–0.041] and 0.122 [0.081–0.161] in thapsigargin-treated cells (1614 light-heavy pairs, Mann-Whitney $P < 2.2e−16$) (Fig. 4a). This robust agreement provided an additional independent confirmation on the accuracy of the spatial measurements. Consistently, among heavy-light protein pairs with confidently assigned subcellular localization, the heavy and light proteins are assigned to the identical subcellular compartment in 93% and 89% of the cases in normal and thapsigargin-treated AC16 cells, respectively (Fig. 4b). We focused on the unusual cases where the spatial distribution distances between the heavy and light proteins increased noticeably following thapsigargin treatment, as they may be indicative of localization changes that are dependent upon time since synthesis. These include two proteins EGFR and ITGAV with an uncommon increase in heavy-light spatial distances (Z: 2.00 and 2.24, respectively) (Fig. 4c). Epidermal growth factor receptor (EGFR/ErbB1/HER1) is a receptor tyrosine kinase with multiple subcellular localizations and signaling roles, and is implicated in cardiomyocyte survival[30]. Following a variety of stressors, EGFR is known to be inactivated by intracellular trafficking, including being internalized to the early endosome and lysosome following oxidative stress and hypoxia in cancer cells[31]. In the spatial proteomics data, the spatial distribution of EGFR borders the lysosome and plasma membrane fractions, which we interpret as EGFR having potential multiple pools including a cell surface fraction (Fig. 4d). In the thapsigargin-treated cells, the light-heavy spatial distance of EGFR increased from 0.014 in normal cells to

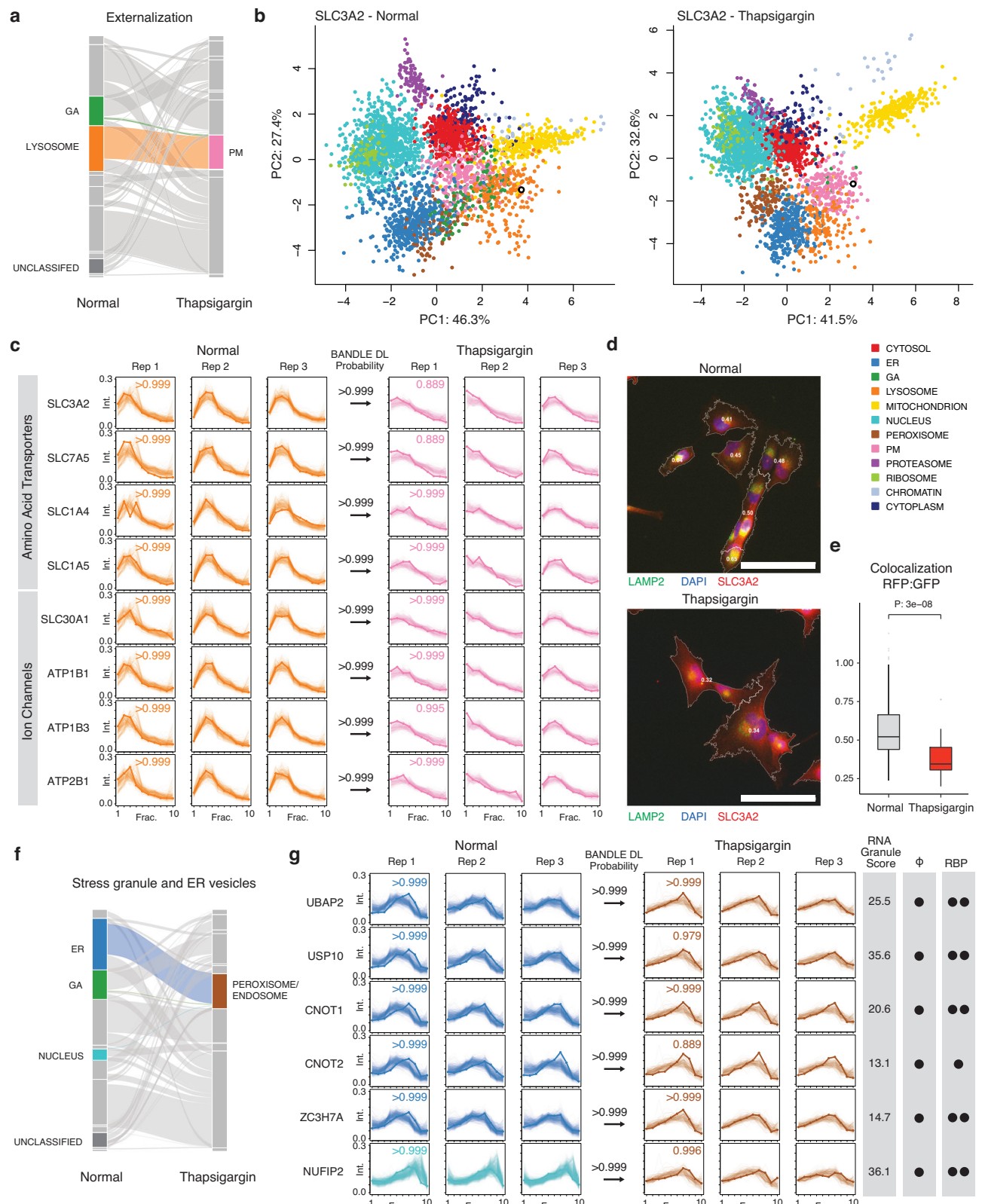

0.052, and the dynamic SILAC labeled pool (heavy) becomes internalized toward the ER (BANDLE differential localization probability: >0.999) but not the pre-existing (light) pool (Fig. 4e). The data therefore suggests that the internalization of EGFR under thapsigargin is likely to be due to endomembrane stalling or redistribution upon new protein synthesis, possibly leading to fewer new EGFR molecules reaching the cell surface. Likewise, in thapsigargin-treated AC16 cells,

newly-synthesized ITGAV (integrin subunit alpha V) shows a partition from the plasma membrane fraction to the ER fraction but not the old/existing protein pool, concomitant with an increase in spatial distribution distance from 0.017 to 0.057 between old and new proteins (Fig. 4f–g). With the function of integrins as cell surface receptors that function in intracellular-to-extracellular and retrograde communication, the ER localization of newly synthesized ITGAV, such as due to

**Fig. 3 | SPLAT captures extensive protein localization changes under UPR.**
**a** Alluvial plot of differential localization (DL) events (> 0.99 BANDLE DL probability; estimated FDR < 1%) following thapsigargin treatment showing a cohort of proteins moving from the Golgi apparatus (GA) and lysosome towards the plasma membrane (PM) (*n* = 3). **b** Spatial map for SLC3A2 (open black circle) in normal (left) and thapsigargin-treated (right) AC16 cells, showing its lysosomal assignment in normal cells and PM assignment in stressed cells. Colors: allocated subcellular compartment. **c** Ultracentrifugation fraction profile of SLC3A2 and other amino acid transporters SLC7A5, SLC1A4, SLC1A5 and ion channel proteins SLC30A1, ATP1B1, ATP1B3, and ATP2B1 with similar migration patterns. X-axis: fraction 1 to 10 of ultracentrifugation. Y-axis: relative channel abundance. Bold lines: protein of interest; light lines: ultracentrifugation profiles of all proteins classified to the compartment. Colors correspond to subcellular localization in panel **b** and all AC16 data in the manuscript; numbers within boxes: BANDLE allocation probability to compartment; numbers at arrows: BANDLE DL probability. **d** Immunofluorescence of SLC3A2 (red) against the lysosome marker LAMP2 (green) and DAPI (blue). Numbers in cell boundary: colocalization score per cell. Scale bar: 90 μm. **e** Colocalization score (Mander's correlation coefficient) between SLC3A2 and LAMP2 decreases significantly (two-tailed unpaired t-test P value: 3e−8) in thapsigargin, consistent with movement away from lysosomal fraction (*n* = 205 normal cells, *n* = 32 thapsigargin treated cells). Center line: median; box limits: interquartile range; whiskers: 1.5x interquartile range; points: outliers. **f** Alluvial plot showing the migration of ER, GA, and nucleus proteins toward the peroxisome/endosome containing fraction in thapsigargin treated cells. **g** Ultracentrifugation fraction profile of stress granule proteins UBAP2, USP10, CNOT1, CNOT2, ZC3H7A, and NUFIP2. RNA Granule Score 7 or above is considered a known stress granule protein. Phi: predicted phase separation participation. Circle denotes a prediction of True within the database. RBP: Annotated RNA binding protein on the RNA Granule Database. One circle denotes known RBP in at least one data set; two circles denote known RBP in multiple datasets.

stress-induced stalling of protein trafficking along the secretory pathway, could indicate a decrease in integrin signaling function through spatial regulation rather than protein abundance. To partially corroborate the partial redistribution of EGFR, we performed immunocytochemistry imaging of EGFR subcellular distribution in AC16 cells with or without thapsigargin (Fig. 4h). Thapsigargin treatment did not increase cell size (Fig. 4i), and whereas there is an increase in immunofluorescence signal of EGFR in thapsigargin (Fig. 4j), this signal is distributed preferentially to the interior of the cell such that there is a significant reduction in the ratio of mean intensity at cell borders over the whole cell in thapsigargin vs. untreated cells (0.673 vs. 0.746, *n* = 93 and 71 cells, Mann-Whitney P: 1.7e−4) (Fig. 4k), consistent with a partial redistribution of EGFR toward an internal pool. Taken together, these examples demonstrate the SPLAT strategy can be used to distinguish time-dependent differential localization of proteins such as due to the trafficking of newly synthesized proteins.

## Spatiotemporal proteomics highlights similarities and differences of ER stress induction protocols

We next investigated the protein spatiotemporal features of AC16 cells under the treatment of tunicamycin, another compound commonly used to induce ER stress in cardiac cells[32,33] by inducing proteostatic stress via inhibition of nascent protein glycosylation. Three biological replicate SPLAT experiments were performed in tunicamycin-treated cells to resolve the temporal kinetics and subcellular localisation of proteins (Supplementary Figs. 1, 2, 5–7, 10). Tunicamycin treatment at 1 μg/mL for 16 h robustly induced the known ER stress response markers BiP/HSPA5, HSP90B1, PDIA4, CALR, CANX, and DNAJB11 (limma FDR adjusted *P* < 0.10) (Fig. 5a) demonstrating effective ER stress induction. Overall, tunicamycin treatment led to a lesser slowdown of temporal kinetics than thapsigargin (average protein half-life 32.6 h) (Fig. 5b; Supplementary Data 5). As in thapsigargin treatment, the kinetic changes following tunicamycin are different across cellular compartments, with ER and Golgi proteins having relatively faster kinetics than other cellular compartments (Mann-Whitney test P: 1.3e−13 and 6.6e−5; Bonferroni corrected threshold 0.05/13); whereas the greatest reduction was observed among proteins localized to the lysosome, a compartment closely linked to glycosylation and recycling of glycans (Mann-Whitney test P: 1.8e−10) (Fig. 5c). Gene set enrichment analysis (GSEA) of turnover rate ratios revealed a significant positive enrichment of UPR proteins (adjusted P 3.5e−3) and DNA repair terms (e.g., processing of DNA double-strand break ends; adjusted P 0.017) and a negative enrichment of translation-related terms (Fig. 5d). Compared to thapsigargin treatment however, no significant enrichment of glycosylation and vesicle transport related terms were found in tunicamycin. Inspection of individual protein kinetics changes likewise revealed both similar induction of the ER stress response markers HSPA5, HSP90B1, and

PDIA4 as in thapsigargin treatment, but other stress response genes PDIA3 and NIBAN1 are not induced in thapsigargin (Fig. 5e). On the other hand, RCN3 (reticulocalbin 3) is an ER lumen calcium-binding protein that regulates collagen production[34] and shows increased temporal kinetics in thapsigargin (Fig. 2f) but not in tunicamycin (ratio 0.76 over normal; Supplementary Data 5), altogether reflecting potential differences in stress response modality to a different ER stress inducer.

Parallel to the less prominent changes in vesicle transport, tunicamycin treatment also led to fewer protein differential localizations than thapsigargin, with 620 differentially localized features (including 282 light proteins and 338 heavy proteins) at BANDLE differential localization probability >0.95, corresponding to an estimated FDR of 0.35%, and thresholded by bootstrapping differential localization probability >0.95; from which we highlighted 109 proteins where the heavy and light versions both showed confident differential localization. The spatial data revealed a high degree of similarity but also notable differences with thapsigargin-induced ER stress. We found that in both tunicamycin and thapsigargin treatment, there was evidence of lysosome targeting from other endomembrane compartments, including: RRBP1, a ribosome-binding protein of the ER, GANAB, a glucosidase II alpha subunit integral to the proper folding of proteins in the ER, FKBP11, a peptidyl-prolyl cis/trans isomerase important to the folding of proline-containing peptides, IKBIP, an interacting protein to the IKBKB nuclear kinase, and MANF, a neurotrophic factor which has relations to ER stress-related cell death when its expression is lowered (BANDLE differential localization probability >0.999)[35] (Fig. 5f). Among these proteins was the collagen synthesis enzyme P3H1 in both thapsigargin and tunicamycin. Interestingly, prior work found no correlation between the protein abundance of collagen-modifying enzymes with the known reduction of collagen synthesis in chondrocytes and fibroblasts under ER stress[36]. The results here suggest that the functional decline may instead correlate with a change in the subcellular localization of collagen-modifying enzymes in AC16 cells. Tunicamycin treatment also induced old-new protein partitions in EGFR and ITGAV as observed in thapsigargin (Supplementary Fig. 11). Notably, although tunicamycin also induced the movement of proteins toward the peroxisome/endosome fraction, different proteins are involved, including the stress response proteins DNAJB11, DNAJC3, DNAJC10, and PDIA6 as well as other proteins EMC4, EMC8, VAPA, and VAPB (Supplementary Fig. 12) which further outlines different modalities of cellular response toward two different ER stress inducers. The differentially localized stress response proteins DNAJB11, DNAJC10, and PDIA6 also showed a significant acceleration in temporal kinetics in tunicamycin (Mann-Whitney test, FDR adjusted *P* < 0.10; Supplementary Data 6) which is consistent with specific production of the proteins followed by shuttling to subcellular location for their function during stress response.

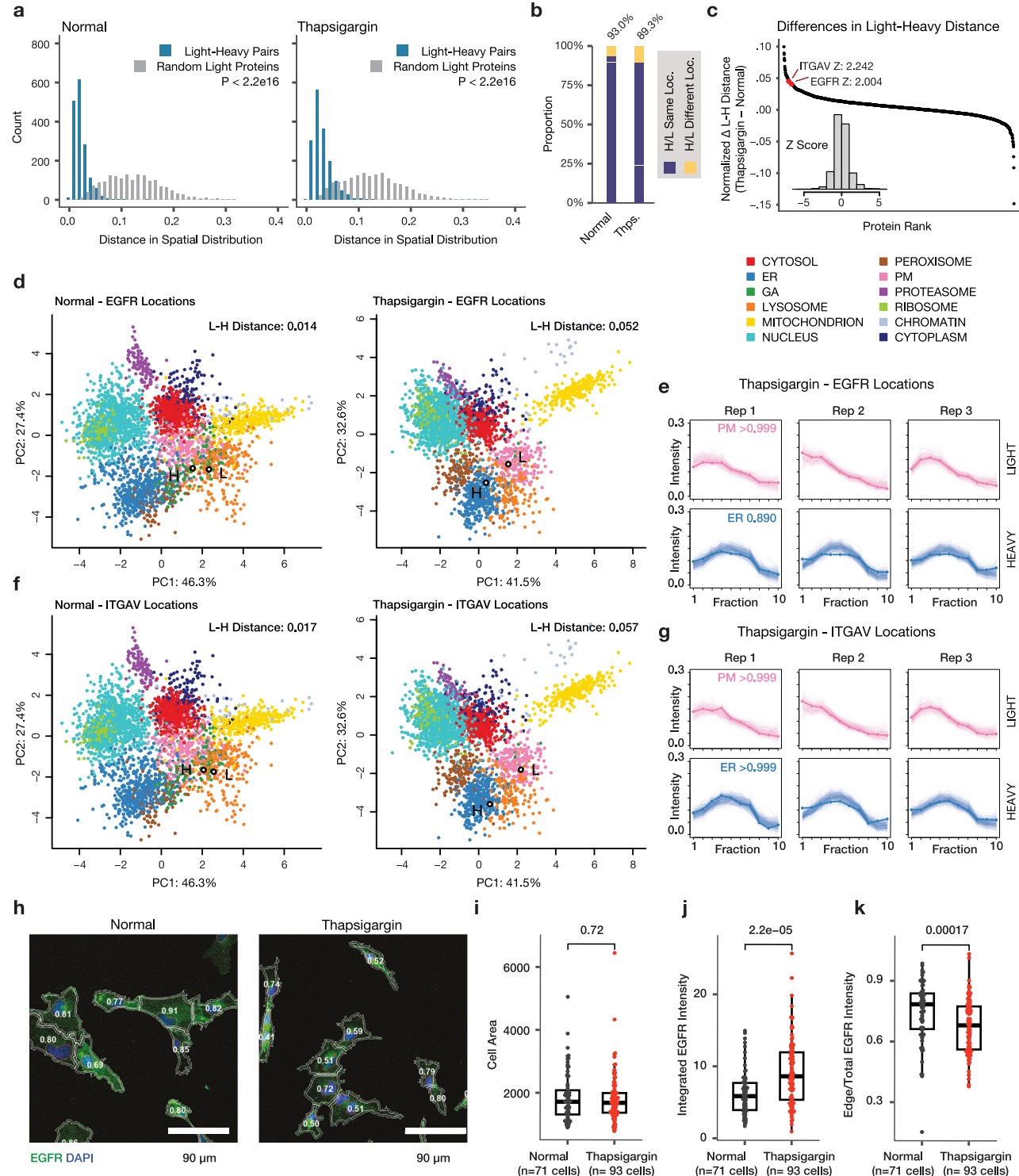

**Application of SPLAT to the mechanism of cardiotoxicity in iPSC-CM models**

We next assessed the applicability of SPLAT toward a different, non-proliferating cell type (human iPSC-derived cardiomyocytes [iPSC-CMs]) and its utility for interrogating the mechanism of cardiotoxicity following proteasome inhibitor treatment (Fig. 6a). The ubiquitin proteasome system is responsible for the degradation of over 70% of cellular proteins. Compounds that inhibit proteasome function, including carfilzomib, are widely used as cancer treatment and have led to remarkable improvement in the survival of multiple myeloma patients. Mechanistically, carfilzomib functions by binding to and irreversibly inhibiting the proteasome catalytic subunit PSMB5[37], leading to the accumulation of unfolded proteins in cancer cells. Importantly, despite its efficacy as a cancer treatment, carfilzomib also leads to cardiotoxic adverse effects including heart failure (<20%), arrhythmia (<10%), and hypertension (11-37%) in a significant number of patients[38]. This cardiotoxicity has been modeled in vitro by exposure of 0.01 – 10 μM carfilzomib to human iPSC-CMs[39], yet the molecular mechanisms of carfilzomib cardiotoxicity remain incompletely understood. To examine the protein spatiotemporal changes upon carfilzomib-mediated proteasome inhibition in cardiac cells, we differentiated contractile iPSC-CMs using a small molecule based

**Fig. 4 | SPLAT reveals protein-lifetime dependent differential localization.**
**a** Histogram of distances in light and heavy proteins in normalized fraction abundance profiles in normal and thapsigargin-treated (Thps.) AC16 cells. X-axis: the spatial distribution distance of two proteins is measured as the average euclidean distance of all TMT channel relative abundance in the ultracentrifugation fraction profiles across 3 replicates; y-axis: count. Blue: distance for 1,614 quantified light-heavy protein pairs (e.g., unlabeled vs. SILAC-labeled EGFR). Grey: distribution of each corresponding light protein with another random light protein. *P* value: Mann–Whitney test (two-sided). **b** Proportion of heavy-light protein pairs with confidently assigned localization to the same location (purple) in normal (left; 93%) and thapsigargin (right; 89%) cells. **c** Ranked changes in heavy-light pair distance in thapsigargin treatment. The positions of EGFR and ITGAV are highlighted. Inset: Z score distribution of all changes. **d** Spatial map of the light and heavy EGFR in normal and thapsigargin-treated cells. Each data point is a light or heavy protein species. Color: assigned compartment. Numbers: euclidean distance in fraction

profiles over 3 replicates. **e** Corresponding fraction profiles; x-axis: ultra-centrifugation fraction; y-axis: fractional abundance. Post-labeling synthesized EGFR is differentially distributed in thapsigargin and shows ER retention (blue), whereas the preexisting EGFR pool remains to show a likely cell surface localization (pink) after thapsigargin. **f**, **g** As above, for ITGAV. **h** Confocal imaging of EGFR immunofluorescence supports a partial relocalization of EGFR from the cell surface toward internal membranes following thapsigargin. Numbers: ratio of EGFR at the plasma membrane vs. whole cells. Blue: DAPI; green: EGFR; scale bar: 90 μm. **i** Cell areas; Mann-Whitney two-sided P: 0.72. Center line: median; box limits: interquartile range; whiskers: 1.5x interquartile range. **j** Total EGFR intensity per cell; Mann-Whitney two-sided P: 2.2e-05. Center line: median; box limits: interquartile range; whiskers: 1.5x interquartile range. **k** Edge/total intensity ratios in normal and thapsigargin-treated AC16 cells (*n* = 71 normal cells; *n* = 93 thapsigargin cells; Mann–Whitney two-sided P: 1.7e−4). Center line: median; box limits: interquartile range; whiskers: 1.5x interquartile range.

protocol, and treated the cells with 0.5 μM carfilzomib. To verify toxicity modeling, we measured iPSC-CM viability and phenotypes under 0 to 5 μM carfilzomib for 24 and 48 h. Under the chosen treatment (0.5 μM for 48 h), iPSC-CMs showed sarcomeric disarray consistent with prior observations on the cardiotoxic effects of carfilzomib (Fig. 6b) but maintained viability (82%) (Fig. 6c), while showing significant decreases in oxygen consumption (Fig. 6d), basal respiration (Fig. 6e), and maximal respiration (Fig. 6f), whereas higher doses are accompanied with drops in viability at 48 h and an increase in proton leak (Fig. 6g). ATP production at the 0.5 μg dose was significantly lower than untreated cells at both 24 and 48 h (Fig. 6h). Therefore cardiotoxicity due to carfilzomib can be recapitulated in a human iPSC-CM model, consistent with prior work in the literature.

From the untreated and carfilzomib-treated (0.5 μM, 48 hrs) human iPSC-CMs, we constructed a protein subcellular spatial map that takes into account several features of the iPSC-CM cell type, including the inclusion of cell junction and desmosome proteins, as well as a sarcomere protein compartment, that are not apparently recognized as discrete compartments in the prior spatial maps (Supplementary Fig. 13). In addition, the 40 S and 60 S ribosomes showed clear separation in iPSC-CM unlike in AC16 cells and are hence classified separately. This separation is consistent with less active protein translation in this cell type. In total, we mapped the subcellular localization of 5047 protein features including 2680 old proteins and 2367 new proteins using a stringent two-peptide filter at 1% FDR, including 2010 old proteins and 1737 new proteins assigned to one of 13 subcellular localization with >95% confidence after removing outliers (Fig. 6i; Supplementary Data 7). The iPSC-CM spatial map achieved similar levels of concordance with known cellular component annotations as in AC16 cells (70.8% with known annotation matching the assigned compartment in normal iPSC-CM; 63.0% in carfilzomib-treated cells) (Supplementary Fig. 14). The iPSC-CM map has comparable spatial distance between light and heavy protein pairs as in AC16 cells, and 87.6% heavy-light protein pairs map to the same compartment in the baseline, supporting that there is sufficient spatial resolution to resolve subcellular compartment differences in this cell type (Supplementary Fig. 14).

Among proteins with spatial information, we compared the temporal kinetics of 2648 proteins. Unexpectedly, there was little overall slowdown of protein temporal kinetics with the median protein half-lives being 97.4 h and 100.0 h in normal and carfilzomib-treated cells, respectively (Fig. 6j; Supplementary Data 8), suggesting that at 48 h following proteasome inhibitor treatment, the observed cellular toxicity is not directly explainable by a drop of per-protein average in global protein degradation. At 48 h of carfilzomib treatment in iPSC-CMs, proteasome chymotrypsin-like activities are partially suppressed but significant partial activities are also observable (Fig. 6k); whereas other proteolysis mechanisms may also compensate for proteasome inhibition, including a suggestive increase in autophagy (P: 0.053)

(Fig. 6l). Inspection of the spatial data revealed that the changes in protein kinetics upon carfilzomib are localization specific, with a significant reduction in chromatin/sarcomere protein turnover rate, and significant increase for the proteasome compartment under carfilzomib treatment (Fig. 6m).

Notably, on an individual protein level we find that the majority of proteins with increased protein kinetics belong to subunits of the regulatory 19 S complex rather than the core 20 S complex (Fig. 7a) suggesting possible changes in 26 S proteasome activity and target engagement. In addition to proteasome subunits, the temporal kinetics revealed a robust induction of chaperons HSP90AA1/HSP90A, HSP90AB1/HSP90B, HSPA4, and BAG3; and ERAD associated proteins VCP and UFD1 (Fig. 7b, Supplementary Data 8). Within the mitochondrion, quality control proteins HSPD1, HSPE1, and CLPB are induced (Fig. 7a). In contrast, among proteins that show reduced turnover in carfilzomib treatment are major sarcomeric proteins MYH6, MYH7, MYBPC3, MYL7; as well as proteins classified to the cell junction compartment dystrophin (DMD) and utrophin (UTRN), and the desmosome complex protein desmoplakin (DSP) (Fig. 7a, c).

We observed some interconnectivity of spatial and temporal changes, with 23 out of 339 pairs of differentially localized proteins also showing significant kinetic changes. BAG3, a muscle chaperone important for sarcomere turnover[40], shows elevated kinetics (Fig. 7b) and a partition away from the soluble cytosol compartment (Fig. 7d) toward an expanded compartment in carfilzomib that co-sediments with Golgi markers. Inspection of existing annotations show that the majority of categorized proteins are not canonical Golgi proteins but contain cytoplasm and endosome terms; hence it likely represents a less soluble cytoplasmic fraction consistent with a lower abundance in the last ultracentrifugation step (Supplementary Fig. 13, Supplementary Fig. 14, Supplementary Data 9). This is consistent with the known dynamic partitioning of BAG3 between the cytosol and myofilament fractions for its function[41]. Secondly, we find accelerated temporal kinetics of PA200/PSME4 proteasome activator (Fig. 7b) in conjunction with a change in localization from the nuclear compartment in baseline toward the proteasome compartment upon carfilzomib (Fig. 7e). The PSME4/PA200 proteasome activator is known to bind with the 20 S/26 S proteasome complex to stimulate proteolysis and has a putative nuclear localization signal[42]. The change in localization is, therefore consistent with increased binding with the proteasome complex. In parallel, the proteasome activator PA28/PSME3 also relocalizes to the proteasome upon carfilzomib (Supplementary Fig. 15), altogether suggesting a remodeling of proteasome configuration upon carfilzomib.

Taken together, the spatiotemporal proteomics data here identified major proteostatic pathways induced in carfilzomib, involving a potential remodeling of the proteasome, induction of chaperones and ERAD proteins, and mitochondrial protein quality control mechanisms that may be important for preserving function. On the other hand, a

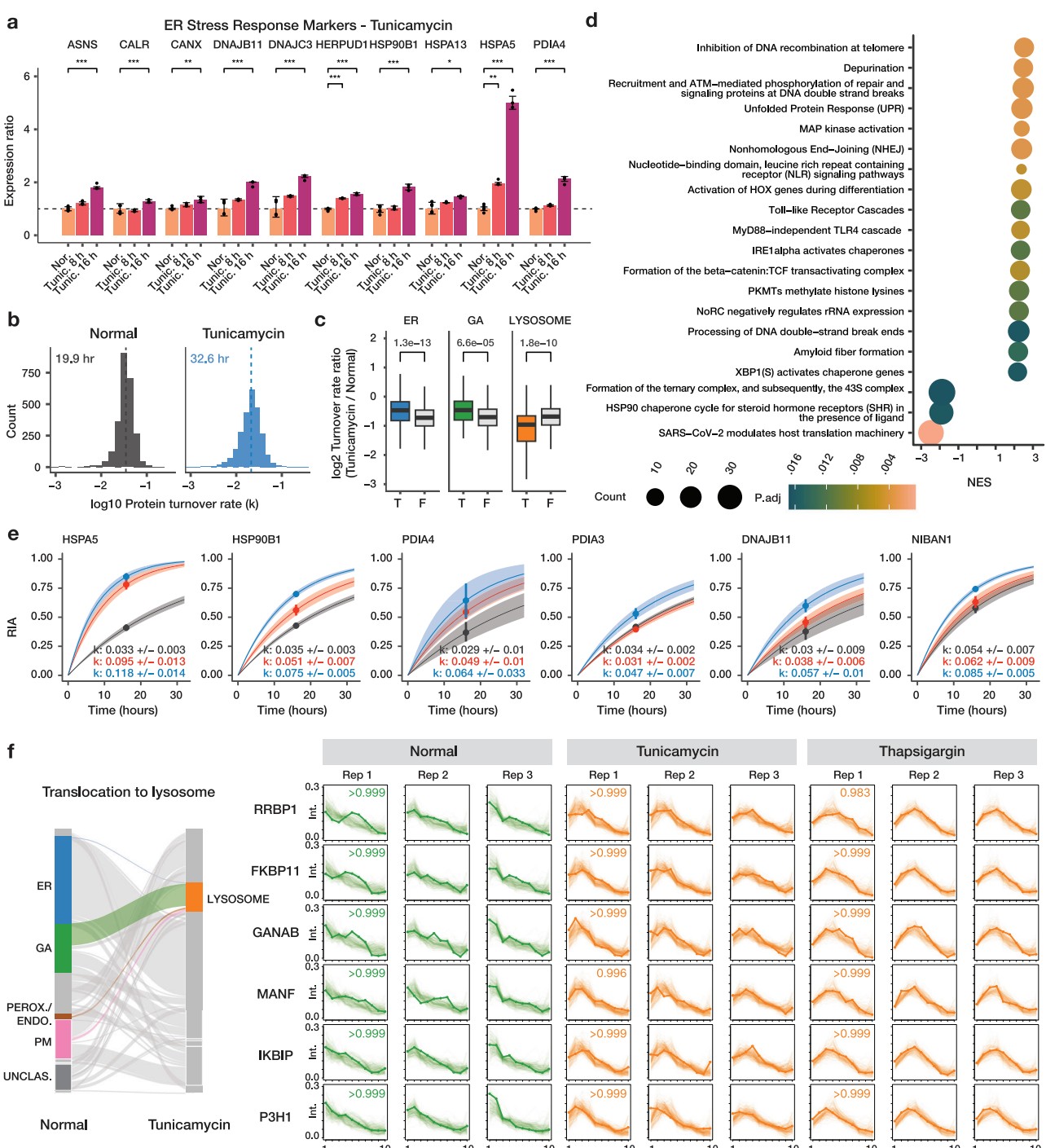

**Fig. 5 | Comparison of ER stress induction methods. a** Bar charts showing activation of known ER stress markers upon tunicamycin (Tunic.) treatment for 8 h and 16 h. X-axis: ER stress markers; y-axis: expression ratio ($n = 6$ normal AC16; $n = 3$ tunicamycin). *: $P < 0.01$; **: $P < 0.001$; ***: $P < 0.0001$; multiple-testing corrected (FDR adjusted) P (two-sided). Error bars: expression ratios ± s.d. **b** Histograms of the determined log10 protein turnover rates in control and tunicamycin-treated cells ($n = 3$). **c** Boxplot showing the log2 turnover rate ratios in tunicamycin over normal AC16 cells for proteins that are localized to the ER (T) or not (F); Golgi apparatus (GA), or the lysosome. P values: two-tailed t-test. Center line: median; box limits: interquartile range; whiskers: 1.5x interquartile range; $n = 286$, 2234, 62, 2458, 139, 2381 proteins over 3 biological independent samples. A Bonferroni corrected threshold of 0.05/13 is considered significant. **d** Gene set enrichment analysis (GSEA) of turnover rate ratios in tunicamycin treatment. Color: multiple-testing corrected (FDR adjusted) P in GSEA (two-sided) x-axis: normalized

enrichment score (NES). Size: proteins in the gene set. **e** Example of best fit curves in the first-order kinetic model at the protein level between normal (gray), tunicamycin (blue) and thapsigargin (red) treated AC16 cells showing known ER stress markers with elevated turnover in both ER stress inducers (HSPA5, HSP90B1, and PDIA4) as well as stress response proteins with elevated turnover only in tunicamycin (PDIA3, DNAJB11, NIBAN1). Point: best-fit k; line: first-order kinetic curve of k; bands: fitting s.e. **f** Alluvial plot showing the migration of ER, GA, and peroxisome/endosome proteins toward the lysosome (left). On the right, the ultracentrifugation fraction profiles of the differentially localized proteins RRBP1, FKBP11, GANAB, MANF, IKBIP, and P3H1 are shown that are targeted toward the lysosome in both tunicamycin and thapsigargin treatment (BANDLE differential localization probability > 0.95). Numbers in boxes are the BANDLE allocation probability in each condition ($n = 3$).

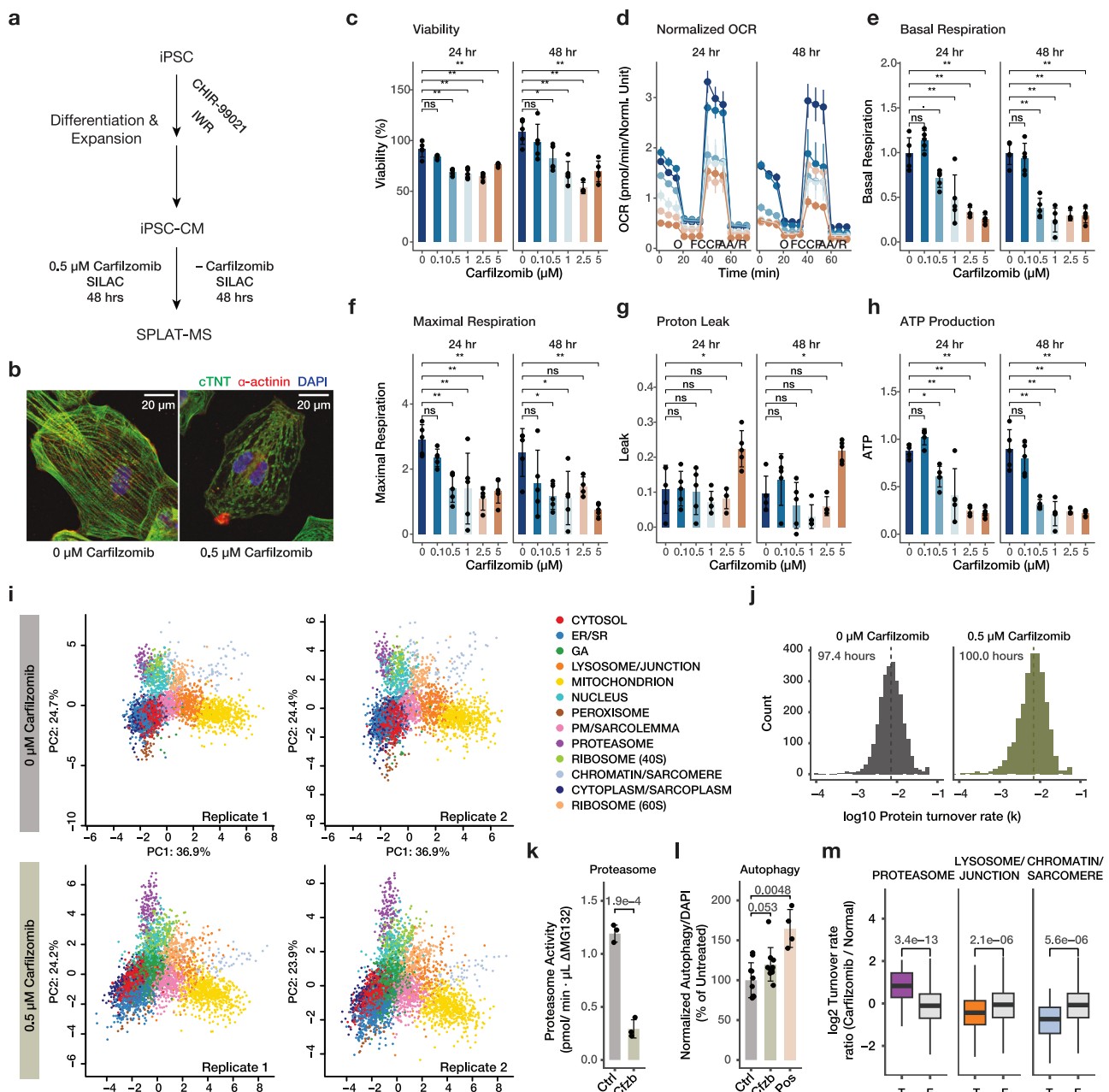

**Fig. 6 | Applicability in human iPSC-derived cardiomyocytes. a** Schematic of human iPSC-CM differentiation, carfilzomib treatment, and SPLAT analysis. **b** Confocal microscopy images showing sarcomeric disarray in iPSC-CMs upon 48 hrs of 0.5 μM carfilzomib; green: cTNT, red: alpha-actinin; blue: DAPI; scale bar: 20 μm. **c–h** Cell viability (%), normalized Seahorse oxygen consumption rate (OCR; pmol/min), basal respiration, maximal respiration, proton leak, and ATP production upon 0–5 μM carfilzomib for 24 or 48 hrs; •: $P < 0.1$; *: $P < 0.05$; **: $P < 0.01$, ANOVA with Tukey's HSD post-hoc (two-sided) at 95% confidence level; $n = 5$ biologically independent experiments. Error bars: mean ± s.d. for bar charts in panels **c, e, f, g, h**; mean ± s.em. for the OCR graph in panel **d**. Colors in panel **d**: dosage, same as panel **c**. O: Oligomycin; AA/R: Antimycin A/Rotenone. **i** Spatial map with 13 assigned subcellular localizations in iPSC-CMs at the baseline (top) and upon 0.5 μM carfilzomib treatment ($n = 2$). Source data are provided as a Source Data file.

**j** Histogram of log10 protein turnover rates (**k**), with median half-life of 97.4 h and 100.0 h in normal and carfilzomib-treated iPSC-CM. **k** Proteasome activity in iPSC-CMs treated with 0 (Ctrl) vs. 0.5 μM carfilzomib (Cfzb) for 48 h. P value: two-tailed t-test; $n = 3$ biologically independent samples; error bar: mean ± s.d. **l** Autophagy assay for iPSC-CMs treated with 0 (Ctrl) vs. 0.5 μM carfilzomib (Cfzb) for 48 h, and positive control (Pos); data were normalized to DAPI and normal cells. P value: two-tailed t-test; $n = 10$ biologically independent samples; error bar: mean ± s.d. **m** log2 Turnover rate ratios between carfilzomib-treated and untreated iPSC-CM from the spatiotemporal proteomics data. P values: Mann–Whitney test (two-sided); with a Bonferroni threshold of 0.05/14 considered significant. Center line: median; box limits: interquartile range; whiskers: 1.5x interquartile range; $n = 60, 2572, 175, 2457, 51, 2581$ proteins over 2 biologically independent samples.

preferential decrease of temporal kinetics in sarcomere and desmosome proteins suggests that the interruption of protein quality control and turnover in these important cardiomyocyte components may be principal sites of lesion in carfilzomib cardiotoxicity. Finally, we assessed the protein-level expression profiles in the hearts of mice

treated with carfilzomib for 2 weeks to model cardiac dysfunction (Supplementary Methods). Notably, we find differential protein abundance analysis showed that MHC-β (MYH7) and desmoplakin (DSP) are the 1st and 5th most significantly up-regulated proteins among 3379 quantified proteins in the hearts of mice treated with

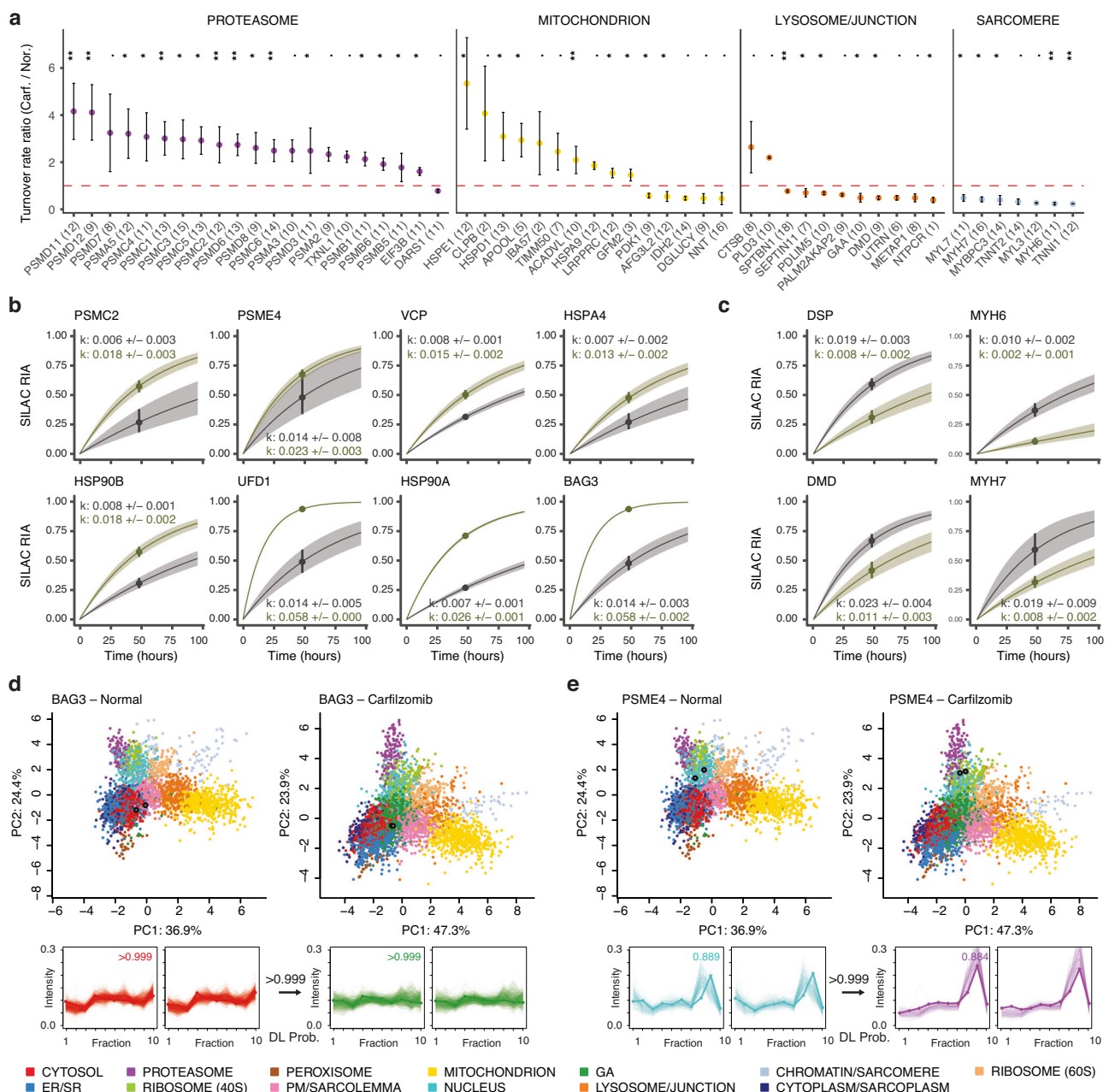

**Fig. 7 | Proteostatic pathways and lesions in carfilzomib mediated cardiotoxicity in iPSC-CMs. a** Changes in protein turnover rates between carfilzomib vs. normal iPSC-CMs across selected cellular compartments; •: $P < 0.1$; *: $<0.05$; **$<0.01$; Mann-Whitney test (two-sided) with Benjamini-Hochberg multiple-testing correction. Dashed line: 1:1 ratio; point: median ratio; range: median ± MAD/median of ratios; n=number of peptide observations (parenthesis) over 2 biologically independent samples per gorup. Source data are provided as a Source Data file. **b** Kinetic curve representations of proteins with accelerated temporal kinetics in carfilzomib, including PSMC2 which corresponds to the ratio in panel A, as well as additional ERAD proteins and chaperones; gray: normal iPSC-CM; green:

carfilzomib. Point: best-fit k; line: the first-order kinetic curve of k; bands: fitting s.e. **c** Kinetic curve representations of slowdown of protein kinetics in DSP, DMD, MYH6, and MYH7, corresponding to the ratios in panel a. Point: best-fit k; line: first-order kinetic curve of k; bands: fitting s.e. **d**, **e** Spatial map (PC1 vs. PC2) and ultracentrifugation fraction profiles of **d**. BAG3 and **e**. PSME4 in normal and carfilzomib-treated human iPSC-CM, showing a likely differential localisation in conjunction with kinetics changes. White-filled circles: light and heavy BAG3 or PSME4 in each plot. The kinetic curves of BAG3 and PSME4 are in panel **b**. Numbers at arrows correspond to BANDLE differential localization (DL) probability.

carfilzomib (Supplementary Fig. 16), consistent with their accumulation following proteasome inhibition and suggesting the possibility that similar proteostatic lesions may underlie cardiotoxicity mechanism in vivo.

## Discussion

Advances in spatial proteomics have opened new avenues to discover the subcellular localization of proteins on a proteome scale. Thus far

however, few efforts have linked the spatial dynamics of the proteome to other dynamic proteome parameters, which hinders a multi-dimensional view of protein function[43,44]. Co-labeling of SILAC (MS1) and TMT (MS2) tags have been used to increase the channel capacity of quantitative proteomics experiments[45]. Here we leveraged this extended labeling capacity to encode spatial and temporal information in the same experiment (Fig. 1d). We chose to use differential ultra-centrifugation to resolve protein spatial distribution because of its

ability to yield consistent pelleted fractions across replicates. However, this approach also has some limitations. For instance, the differential ultracentrifugation method employed here requires ~$10^7$ cells. Due to the large number of cells required, we focused on acquiring data at a single SILAC time point per treatment (16 h post thapsigargin or tunicamycin; 48 h post carfilzomib), which was selected based on the drug treatment models but also needed to be sufficient to capture the median half-life of proteins in the cell types studied (Figs. 2d and 6j). Although there is no inherent limit to the number of dynamic SILAC labeling time points that can be investigated, in practice the collection time points need to be prioritized for different cell types with distinct intrinsic protein turnover rates. The inclusion of earlier time points might provide insight into early differential localization events, e.g., during acute phase ER stress response.

Another limitation is the lower resolution of differential ultracentrifugation-based fraction separation compared to gradient-based separation, which can lead to some subcellular compartments not being resolved, e.g., lysosome from cell junction in iPSC-CMs. This may be improved in future work couples turnover analysis to gradient-based sedimentation approaches with higher spatial resolution. Protein correlation profiling-based techniques generally also face challenges in recognizing proteins with multiple localizations or partial redistributions. For instance, the multi-functional ERAD protein p97/VCP is known to have multiple subcellular localizations, but its precise spatial profile is difficult to interpret from the TMT data and is unclassified to any compartment. Because differential localization may be sub-stoichiometric, this can lead to lower confidence in the classification of location. Under stress, some organelles may sub-compartmentalize or fragment, which is currently difficult to trace with LOPIT/protein correlation profiling approaches. The ability to observe some additional biologically relevant differential localization events may require the development of spatial proteomics methods that combine orthogonal separation principles or the use of proximity labeling-based alternatives.

Lastly, the double labeling design of SPLAT requires independent MS2 acquisitions of light and heavy SILAC labeled peptides to acquire their TMT profiles separately, which can decrease the depth and data completeness of mass spectrometry-based proteomics analysis. However, this approach has the advantage of being able to resolve proteome-wide turnover spatial distributions in the same experiment and allowing separate observations of the spatial distribution of new and old protein pools. Despite each of the SILAC labeled pairs (light and heavy) and their associated TMT profiles being separately quantified by mass spectrometry as independent ions, the spatial profiles of light and heavy proteins are highly similar under baseline conditions (Fig. 4a, b) which provides additional assurance of spatial assignments.

Despite these limitations, we were able to integrate dynamic time-resolved stable isotope labeling kinetic analysis with differential ultracentrifugation-based subcellular proteomics to characterize proteome-wide spatial and temporal changes upon perturbation. Applying the SPLAT workflow to human AC16 cardiac cells under thapsigargin and tunicamycin induced ER stress and the associated UPR, we observed that both ER stress inducing drugs led to a global suppression in turnover rate, consistent with the reduced translation known to be caused by ER stress. The temporal kinetics data of individual proteins revealed the coordinated activation of known and suspected stress mediators through their increased kinetics, particularly concentrated in the ER and Golgi compartments (Fig. 2e). At the same time, the spatial proteomics profiles revealed substantial endomembrane remodeling and hundreds of differentially localized proteins under ER stress. A recent work also reported ~75 relocalized proteins under acute low-dose thapsigargin (250 nM for 1 h) in U-2 OS cells, although the experimental approach was more optimized toward mRNA detection[46]. The spatial data here add to an emerging view of dynamic protein regulation under cellular stress, illustrating a differential localization of RNA binding proteins to stress granules, targeting of specific proteins toward lysosomes, as well as membrane trafficking of ion channels and amino acid transporters. UPR activation is known to induce the biosynthesis of non-essential or partly-essential amino acids despite protein synthesis suppression[47]; the recycling of lysosomal lysine and arginine regulates the sensitivity to ER stress[48]; whereas deprivation of amino acids is known to activate downstream pathways of UPR including CHOP in vitro[49]. Moreover, the knockdown of SLC3A2 has been found to suppress the activation of ER stress response pathways including ATF4/6 induction[50], together suggesting amino acid transporters may function in ER stress response. The data here indicate that these transporters may in turn be regulated by their spatial localization beyond steady-state abundance. Other changes are found that may be specific to stressors. Tunicamycin treatment leads to a further decrease in turnover of lysosomal proteins compared to other organelles. The lysosome plays an important roles in the recycling of glycans, a process which may be slowed under ER stress in response to the decrease in glycosylation.

Changes in protein spatial distribution can occur due to a relocation of an existing protein, where in a protein may respond to a signaling cue such as a post-translational modification status and subsequently migrate to a subcellular location. Alternatively, an alternate localization of newly synthesized proteins can also drive spatial redistribution. By comparing the spatial distributions of new and old protein pools separately, we were able to observe a partitioning of new and old protein pools including epidermal growth factor receptor (EGFR/ErbB1/HER1). ErbB family proteins are required for both normal heart development and the prevention of cardiomyopathies in the adult heart. EGFR is a receptor tyrosine kinase of this family capable of triggering multiple signaling cascades, and can be activated via both ligand dependent and ligand independent pathways. Upon ER stress induction, EGFR immunofluorescence showed a partial distribution away from the plasma membrane. Although immunofluorescence cannot distinguish between old and new protein pools, this partial differential distribution is consistent with the mass spectrometry data showing partial differential localization, involving the newly synthesized heavy protein pool. We hypothesize that this result reflects a ligand-independent trafficking of newly synthesized protein, rather than ligand-dependent activation and internalization that is agnostic to protein lifetime. Of note, ligand-independent activation and internalization of EGFR has been previously induced via both starvation and tyrosine kinase inhibitor treatment, leading to cellular autophagy[51], hence the observed spatial distribution may carry functional significance to protective cellular response.

We further applied SPLAT to a different, non-proliferating cell type, namely human iPSC-derived cardiomyocytes, which have gained increasing utility for modeling the cardiotoxic effects of cancer drugs, including carfilzomib. In the carfilzomib experiment, the data from SPLAT revealed a surprising similarity in global turnover rates between control and treatment. These observations are consistent with a compensatory rescue of proteasome abundance and activity previously observed in proteasome inhibition by carfilzomib[39,52] or bortezomib[53] in other cell types. Proteasomes are known to be regulated by negative feedback mechanisms[53–55], which could explain the lack of change in proteome-wide half-life differences and instead suggest that toxicity may derive from more specific cellular lesions. We identified a significant reduction in turnover in sarcomere proteins, which may be particularly sensitive to interruptions in proteasome activity and, moreover may account for the bulk of turnover flux in iPSC-CMs given their high abundance. Finally, the induction and differential localization of proteostatic pathway proteins BAG3 and PA200/PSME4 in cardiac cells may be explored as potential targets to ameliorate proteostatic disruptions and cardiotoxic effects. The SPLAT method may therefore be useful for understanding the function and behaviors of proteins inside the cell and may provide new insight

into the mechanisms that regulate protein stability and localization in stress, disease, and drug treatment.

## Methods

All animal procedures were conducted in accordance with the protocol approved by the Institutional Animal Care and Use Committee (IACUC) of the City of Hope National Medical Center. Additional methods can be found in the Supplementary Information file.

### AC16 cell culture, metabolic labeling, UPR induction

AC16 cells procured from Millipore between passage numbers 11 and 16 were cultured in DMEM/F12 supplemented with 10% FBS and no antibiotics. Cells were maintained at 37 °C with 5% $CO_2$ and 10% $O_2$. For isotopic labeling, SILAC DMEM/F12 (Thermo Scientific) deficient in both L-lysine and L-arginine was supplemented with 1% dialyzed FBS and heavy amino acids $^{13}C_6{}^{15}N_2$ L-Lysine-2HCl and $^{13}C_6{}^{15}N_4$ L-Arginine-HCl (Thermo Scientific) at concentrations of 0.499 mM and 0.699 mM, respectively. Light media was switched to heavy media, and cells were labeled for 16 h prior to harvest. UPR was induced with 1 μM thapsigargin (SelleckChem) or 1 μg/mL tunicamycin (Sigma) at the same time as isotopic labeling.

### Cell harvest, differential centrifugation, and isobaric labeling

Cell harvest and subcellular fractionation were performed based on the LOPIT-DC differential ultracentrifugation protocol as described in Geladaki et al. [17]. Briefly, AC16 cells were treated, harvested with trypsinization, washed 3× with room temperature PBS, and resuspended in a detergent free gentle lysis buffer (0.25 M sucrose, 10 mM HEPES pH 7.5, 2 mM magnesium acetate). 1.5 mL of suspension at a time was lysed using an Isobiotec ball-bearing homogenizer with a 16 μM clearance size until ~80% of cell membranes were lysed, as verified with trypan blue (approximately 15 passes through the chamber). Lysates were spun 3 times each in a 4 °C swinging bucket centrifuge 200 × *g*, 5 min to remove unlysed cells. The supernatant was retained and used to generate the 9 ultracentrifugation pellets using spin parameters shown in Supplementary Table 2.

The supernatant generated in the final spin was removed and all pellets and the final supernatant were stored at −80 °C until proceeding. Supernatant was thawed on ice and precipitated in 3× the volume of cold acetone overnight at −20 °C. This was used to generate pellet 10 by centrifuging at 13,000 × *g* for 10 min at 4 °C. Excess acetone was removed and the pellet was allowed to dry briefly before resuspension in a resolubilization buffer of 8 M urea, 50 mM HEPES pH 8.5, and 0.5% SDS with 1x Halt Protease and Phosphatase Inhibitor Cocktail (Thermo Scientific). The suspension was sonicated in a Bioruptor with settings 20× 30 s on 30 s off at 4 °C.

Pellets from the ultracentrifugation fractions 1 to 9 were resuspended in RIPA buffer with Halt Protease and Phosphatase Inhibitor Cocktail (Thermo Scientific) with sonication in a Bioruptor with settings 10x 30 s on 30 s off at 4 °C. Insoluble debris was removed from all samples (1-10) by centrifugation at 14,000 × *g*, 5 min. The protein concentration of all samples was measured with Rapid Gold BCA (Thermo Scientific). The samples were digested and isobarically tagged using the iFASP protocol [56]. A total of 25 ug protein per sample in 250 uL 8 M urea was loaded onto Pierce Protein Concentrators PES, 10 K MWCO prewashed with 100 mM TEAB. The samples were again washed with 8 M urea to denature proteins and remove SDS. The samples were washed with 300 uL 100 mM TEAB twice. The samples were then reduced and alkylated with TCEP and CAA for 30 min at 37 °C in the dark. CAA and TCEP were removed with centrifugation and the samples were washed 3x with 100 mM TEAB. Samples were digested atop the filters overnight at 37 °C with mass spectrometry grade trypsin (Promega) at a ratio of 1:50 enzyme:protein. A total of 0.2 mg of TMT-10plex isobaric labels (Thermo Scientific) per differential centrifugation fraction were equilibrated to room temperature

and reconstituted in 20 μL LC-MS grade anhydrous acetonitrile. In each experiment, labels were randomly assigned to each fraction (Supplementary Table 3, Supplementary Table 4) with a random number generator to mitigate the possible batch effect. Isobaric tags were added to peptides still atop the centrifugation filters and incubated at room temperature for 1 h with shaking. The reactions were quenched with 1 μL 5% hydroxylamine at room temperature for 30 min with shaking. Labeled peptides were eluted from the filters with centrifugation. To further elute labeled peptides 40 μL 50 mM TEAB was added and filters were again centrifuged. All 10 labeled fractions per experiment were combined and mixed well before dividing each experiment into two aliquots. Aliquots were dried with speed-vac and stored at −80 °C.

### Liquid chromatography and mass spectrometry

A total of 13 SPLAT mass spectrometry experiments were performed, which constituted 3 biologically independent samples of each of normal, thapsigargin, and tunicamycin-treated AC16; and 2 mass spectrometry experiments of each of normal and carfilzomib-treated iPSC-CMs. One aliquot of peptides per experiment was reconstituted in 50 μL 20 mM ammonium formate pH 10 in LC-MS grade water (solvent A) for high pH reverse phase liquid chromatography (RPLC) fractionation. The entire sample was injected into a Jupiter 4 μm Proteo 90 Å LC Column of 150 × 1 mm on a Ultimate 3000 HPLC system. The gradient was run with a flow rate of 0.1 mL/min as follows: 0–30 min: 0%–40% Solvent B (20 mM ammonium formate pH 10 in 80% LC-MS grade acetonitrile); 30–40 min: 40%-80% Solvent B; 40–50 min: 80% Solvent B. Fractions were collected every minute and pooled into a total of 20 peptide fractions, then dried with speed-vac.

The dried fractions were reconstituted in 10 μL each of pH 2 MS solvent A (0.1% formic acid) and analyzed with LC-MS/MS on a Thermo Q-Exactive HF orbitrap mass spectrometer coupled to an LC with electrospray ionization source. Peptides were separated with a PepMap RSLC C18 column 75 μm x 15 cm, 3 μm particle size (Thermo-Scientific) with a 90 minute gradient from 0 to 100% pH 2 MS solvent B (0.1% formic acid in 80% LC-MS grade acetonitrile). Full MS scans were acquired with a 60,000 resolution. A stepped collision energy of 27, 30, and 32 was used and MS2 scans were acquired with a 60,000 resolution and an isolation window of 0.7 m/z.

### Mass spectrometry data processing and turnover analysis

Mass spectrometry raw data were converted to mzML format using ThermoRawFileParser v.1.2.0[57] then searched against the UniProt Swiss-Prot human canonical and isoform protein sequence database (retrieved 2022-10-27) using Comet v.2020_01_rev3[58]. The fasta database was further appended with contaminant proteins using Philosopher v4.4.0 (total 42,402 forward entries). The search settings were as follows: peptide mass tolerance: 10 ppm; isotope error: 0/1/2/3; number of enzyme termini: 1; allowed missed cleavages: 2; fragment bin tolerance: 0.02; fragment bin offset: 0; variable modifications: TMT-10plex tag +229.1629 for TMT experiments, and lysine + 8.0142, arginine + 10.0083 for all SILAC experiments; fixed modifications: cysteine + 57.0214. The search results were further reranked and filtered using Percolator v3.0[59] at a 1% FDR; proteins identified by 2 peptides are included.

To facilitate data analysis, we provide a Snakemake pipeline "splat-pipeline" that executes Riana and pyTMT to extract and collate MS1 (SILAC) and MS2 (TMT) information. Converted mzML files are input to the splat-pipeline software pipeline to perform the database search step above using Comet and Percolator, after which the mzML files and search result Percolator files are analyzed by Riana and pyTMT. The dynamic SILAC data are analyzed using Riana v0.7.1[11] to integrate the peak intensity within a 25 ppm error of the light (+0), heavy (+8, +10), and double K/R ( + 16, +18, +20) peptide peaks over a 20 second retention time window encompassing the first and last

MS2 scan where the peptide is confidently identified. We then calculated the fractional synthesis of all K/R containing peptides as the intensity of the 0th isotopomer peak (m0) over the sum of applicable light and heavy isotopomers (e.g., m0/m0 + m8 for a peptide with one lysine). Riana then performs intensity-weighted least-square curve-fitting using the scipy optimize function to a first-order exponential rise model to find the best-fit peptide turnover rate. Protein turnover rates are calculated as the median of peptide turnover rates +/- median absolute deviation. To compare protein turnover rates across treatment groups, up to top 10 most intense peptides for each protein in each sample are considered, and the best-fit turnover rates of each peptide are averaged over replicate experiments. A wilcoxon rank sum test is used to assess difference in isotope incorporation across peptides, followed by multiple-testing correction using the Benjamini-Hochberg procedure.

### TMT labeling assignment, isotope correction, and purity assessment

To extract the ultracentrifugation fraction quantification information for spatial analysis, we developed a new version of the pyTMT tool which we previously described[21], and used it to perform TMT label quantification for the Comet/Percolator workflow. First, to account for specific challenges related to spatial proteomics data features, we made two new modifications. First, we account for isotope impurities in TMT tags. Because the TMT data are row normalized in the LOPIT-DC design, we incorporated correction of isotope contamination of TMT channels based on the batch contamination data sheet (Supplementary Table 1) to account for isotope impurity in fractional abundance calculation from randomized channels across experiments. TMT-10 plex lots were #WF309595 for control AC16 replicate 1 and 2; thapsigargin treated AC16 replicate 1, and tunicamycin treated AC16 replicate 1; and #XB318561 for other samples. Isotope impurities in TMT tags can lead to up to 10% spillover to neighbor channels and decrease quantitative accuracy. We used the non-negative least square algorithm in scipy[60] to solve for the true channel matrix from the observed channel intensity and impurity matrix for downstream quantification.

Second, in order to compare protein TMT profiles (i.e., their distributions across ultracentrifugation fractions), the measured TMT channel intensities of each peptide were summed into protein groups. Different proteins or protein isoforms may have different subcellular localization but share peptide sequences. To prevent peptides shared by multiple proteins or protein isoforms from affecting subcellular compartment assignment, we implemented an isoform-aware quantitative rollup of peptide channel intensities into the protein level for the downstream spatial proteomics analysis. Standard protein inference invokes parsimony rules that assign peptides to the protein within a protein group with the highest level of evidence, but razor peptides can conflate spatial information from different proteins with different localization. Here a more conservative aggregate method where identified peptides that are assigned to two or more top-level UniProt protein accessions are discarded to avoid confounding of spatial information in the TMT channels. Moreover, protein groups containing two or more proteins belonging to the same top-level UniProt accession are removed from consideration if one of the non-canonical isoforms contain a unique peptide. Peptides that are shared between a canoncail and an isoform protein of the same top-level UniProt protein accession, but where the non-canonical isoform is not associated with any unique peptide, are assigned to the canonical protein. Protein isoforms are only included in downstream analysis through quantified isoform-unique peptides. False positive proteins are further controlled by requiring two peptides for inclusion in spatial analysis. Next, the TMT-quantified peptide intensities for each channel within each light or heavy protein are then grouped and summed based on whether the peptides contained heavy SILAC modifications. Following Riana and

pyTMT processing, the splat-pipeline then combines the dynamic SILAC and TMT information by peptides and appends a heavy ("_H") tag to the UniProt accession of all peptides containing dynamic SILAC modifications for separate localization analyses. The protein TMT intensities of each channel for each light or heavy protein are then column-normalized and row-normalized for input into pRoloc.

To assess TMT quantification quality (Supplementary Fig. 3), spectral purity was calculated with Philosopher freequant[61] within FragPipe using the MSFragger[62] built-in TMT10 workflow modified to include heavy K (8.0142) and R (10.00827) within the MSFragger variable modifications. mzML files were searched against the UniProt Swiss-Prot human canonical and isoform protein sequence database (retrieved 2023.08.17) appended with decoys and common contaminants.

### Subcellular localization classification

Subcellular localization classification and differential localization predictions were performed using the pRoloc[63] and the BANDLE[64] packages in R/Bioconductor. Three replicate batches of AC16 cells per condition each were individually labeled and treated, fractionated and analyzed by mass spectrometry, and biological replicate data were used for pRoloc and BANDLE analysis. Briefly, the subcellular localization markers were selected from the intersecting proteins with a prior data set generated from human U-2 OS osteosarcoma cells[17] with further curation to account for cell type-specific marker expression (Supplementary Data 10). A random walk algorithm is used to prune the markers to maximize normalized between-class separation. For differential localization analyses we used the Markov-chain Monte-Carlo (MCMC) and non-parametric model in BANDLE to find unknown protein classification and evaluate differential localization probability. MCMC parameters are 9 chains, 10,000 iterations, and 5000 burn-in, 20 thinning, seed 42; convergence of the Markov chains is assessed visually by rank plots as recommended[65]. For additional analysis to describe baseline protein classification and compare MS2 and MS3 performance, a T-augmented Gaussian mixture model with a maximum a posteriori (MAP) method in pRoloc was used.

### Calculation of euclidean distance between fraction profiles

Following column- and row-wise normalization, the distance between light and heavy protein profiles was calculated as their Euclidean distance between an array containing relative abundance across 3 replicate profiles. This distance was divided by the number of replicates.

To compare, an equal sample of light – light pairs were randomly selected and calculated as described. For ranked changes, the difference in heavy-light distances in thapsigargin is adjusted by the average changes in the spatial distance of the light protein with 250 other sampled light proteins to calculate the normalized difference.

### Human iPSC-derived cardiomyocytes and proteasome inhibition

Human AICS-0052-003 induced pluripotent stem cell (iPSC) (monoallelic C-terminus mEGFP-tagged MYL7 WTC; Allen Institute Cell Collection) line was acquired from Coriell Institute and seeded onto Geltrex (Gibco) coated 6 well plates and maintained in StemFlex (Thermo Scientific) media at 37 °C, 5% CO2 with daily media changes. At 80% confluency, cells were passaged using 0.5 mM EDTA before resuspension in StemFlex supplemented with 10 μM Y-27632 (Selleck). Cells were replated into Geltrex coated 12 well plates at a density of $3 \times 10^5$ cells/well and daily media changes of StemFlex continued until the cells reached 80% confluency, day 0 of cardiac differentiation. The iPSCs were differentiated into cardiomyocytes using a small molecule based GSK-3 inhibition-Wnt inhibition protocol[66]. Briefly, on day 0, cell media was replaced with 2 mL/well RPMI supplemented with B-27 minus insulin (Gibco) and 6 μM CHIR99021 (STEMCELL); on day 2, the media was changed to 2 mL/well RPMI + B-27 minus insulin. On day 3,

the media was changed to 2 mL/well RPMI + B27 minus insulin supplemented with 5 µM IWR-1-Endo (STEMCELL). On day 7, the media was changed to 2 mL RPMI + B27 with insulin. Differentiation was confirmed via visualization of morphology, spontaneous contraction of cells, and imaging of the GFP tagged MYL7/MLC-2a. On day 9, the media was changed to RPMI + B27 with insulin without glucose to select for cardiomyocytes. The cardiomyocytes were then passaged at low density with 2 µM CHIR-99021[67]. At approximately 75% confluency on passage 2, CHIR supplemented media was removed and replaced with RPMI B-27 with insulin, and used for experiments on day 25–30 post differentiation. For isotopic labeling, RPMI (Thermo Scientific) deficient in both L-lysine and L-arginine was supplemented with B27 with insulin and heavy amino acids $^{13}C_6^{15}N_2$ L-Lysine-2HCl and $^{13}C_6^{15}N_4$ L-Arginine-HCl (Thermo Scientific) at concentrations 0.219 mM and 1.149 mM, respectively. 48 h after CHIR removal, light media was replaced with this heavy media and cells were labeled for 48 h prior to harvest. 0.5 µM carfilzomib (Selleck) was added with heavy media in the treatment group. Harvesting and ultracentrifugation proceeded as above with the following exception. Due to the diffuse nature of pellets generated in the iPSC-derived cardiomyocytes (iPSC-CM) control experiment, the MLA-50 (Beckman) rotor was switched to the TLA-55 rotor after the generation of pellet. The consistent force (g) of each spin was maintained by increasing the smaller rotor's RPM on subsequent spins. This change was repeated during the iPSC-CM treatment experiment. Proteins from each cellular fraction were digested and analyzed with mass spectrometry as above.

## Data availability

The mass spectrometry data generated in this study have been deposited in the ProteomeXchange database under accession numbers PXD038054, PXD041386, PXD046669, PXD046670, and PXD046671. Source data are provided with this paper. Mass spectrometry raw data were searched against the UniProt Swiss-Prot human canonical and isoform protein sequence database (retrieved 2022-10-27) and appended with contaminant proteins using Philosopher v4.4.0 (total 42,402 forward entries). Known stress granule proteins were retrieved from the RNA granule database (https://rnagranuledb.lunenfeld.ca/). Known subcellular localization were retrieved from UniProt Gene Ontology Cellular Component (CC) terms using UniProt.ws with the following terms: CYTOSOL – cytosol [GO:0005829]; ER – endoplasmic reticulum [GO:0005783] OR endoplasmic reticulum membrane [GO:0005789] OR endoplasmic reticulum lumen [GO:0005788]; GA – Golgi apparatus [GO:0005794] OR Golgi lumen [GO:0005796] OR Golgi membrane [GO:0000139]; LYSOSOME – lysosome [GO:0005764] OR lysosomal membrane [GO:0005765] OR lysosomal lumen [GO:0043202]; MITOCHONDRION – mitochondrion [GO:0005739] OR mitochondrial inner membrane [GO:0005743] OR mitochondrial outer membrane [GO:0005741] OR mitochondrial matrix [GO:0005759] OR mitochondrial respirasome [GO:0005746]; NUCLEUS – nucleus [GO:0005634] OR chromatin [GO:0000785] OR nucleoplasm [GO:0005654] OR nucleolus [GO:0005730]; PEROXISOME – peroxisome [GO:0005777] OR peroxisomal matrix [GO:0005782] OR peroxisomal membrane [GO:0005778]; PM – plasma membrane [GO:0005886] OR cell surface [GO:0009986]; PROTEASOME – proteasome complex [GO:0000502] OR proteasome accessory complex [GO:0022624] OR proteasome regulatory particle [GO:0005838]; RIBOSOME – ribosome [GO:0005840] OR cytosolic ribosome [GO:0022626]; CHROMATIN – chromosome [GO:0005694] OR chromatin [GO:0000785] OR nucleosome [GO:0000786] OR euchromatin [GO:0000791] OR heterochromatin [GO:0000792]"; CYTOPLASM – cytoplasm [GO:0005737] OR cytoskeleton [GO:0005856] OR actin cytoskeleton [GO:0015629] OR microtubule [GO:0005874] OR microtubule cytoskeleton [GO:0015630] OR cortical actin cytoskeleton [GO:0030864] OR actin filament [GO:0005884] OR cortical cytoskeleton [GO:0030863] OR intermediate filament cytoskeleton [GO:0045111]. For iPSC-CM, the RIBOSOME (40 S) compartment was matched against ribosome [GO:0005840] OR cytosolic ribosome [GO:0022626] OR eukaryotic 43 S preinitiation complex [GO:0016282] OR eukaryotic 48 S pre-initiation complex [GO:0033290]; the RIBOSOME (60 S) compartment was matched against ribosome [GO:0005840] OR cytosolic ribosome [GO:0022626] OR polysomal ribosome [GO:0042788]. The LYSOSOME/JUNCTION compartment was additionally matched against cell-cell junction [GO:0005911] OR adherens junction [GO:0005912] OR catenin complex [GO:0016342] in addition to the LYSOSOME terms above. The CHROMATIN/SARCOMERE compartment was additionally matched against sarcomere [GO:0030017] OR Z disc [GO:0030018] OR muscle myosin complex [GO:0005859] in addition to the CHROMATIN terms above.

## Code availability

Software code is available on GitHub or the splat-pipeline at https://github.com/lau-lab/splat (https://doi.org/10.5281/zenodo.10614236), Riana at https://github.com/ed-lau/riana (https://doi.org/10.5281/zenodo.10614234) and pyTMT at https://github.com/ed-lau/pytmt (https://doi.org/10.5281/zenodo.10614229).

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

## Acknowledgements

The authors thank Dr. Christopher Ebmeier (Director, Proteomics and Mass Spectrometry Core Facility, University of Colorado Boulder) for his assistance with TMT-MS3 experiments. This work was supported in part by NIH award K08-HL148540 to J-W.R., NIH awards R01-HL141278 and R01-GM144456 to M.P.L. and NIH award R35-GM146815 to E.L; and the University of Colorado SOM Translational Research Scholars Program (TSRP) award to E.L.

## Author contributions

J.C. and E.L. conceptualized the study. J.C., V.M., S.K.R. V.H., C.L., V.A., R.W.L., K.Z., J.P., J-W.R. and M.P.L. performed experiments. J.C., V.M., M.P.L., and E.L. processed the data and interpreted the results. J.C. and E.L. wrote software code and performed analysis. J.C., S.K.R., V.M., and E.L. drafted and revised the manuscript. J-W.R., Z.V.W., M.P.L., and E.L. managed funding. All authors read and approved the final version of the manuscript.

## Competing interests

The authors declare no competing interests.
