## [Peer Review File · Nature Communications]

REVIEWER COMMENTS

Reviewer #2 (Remarks to the Author):

The manuscript 'Simultaneous proteome localization and turnover analysis reveals spatiotemporal features of protein homeostasis disruptions' by Currie et al. establishes SPLAT, a mass spectrometry based proteomics approach that links the temporal component of protein turnover with spatial information of protein localization by combining heavy isotope labeling by SILAC with subcellular fractionation and tandem mass tag (TMT) labeling. The authors use this approach to decipher organelle-dependent differences in protein turnover rates upon ER stress, and to identify stress-induced localization changes of pre-existing and newly synthesized proteins during stress. Lastly, the authors apply SPLAT to a cellular cardiotoxicity model to investigate temporal and spatial proteome changes upon treatment with the proteasome inhibitor carfilzomib at cardiotoxic concentrations.

The manuscript covers a very interesting topic and the authors combine their proteomics analyses with a wide range of computational approaches. However, the overall novelty remains limited. While the SPLAT technique is a useful tool to monitor dynamics in cellular stress responses, the data interpretation in this manuscript does not provide sufficient new mechanistic insights on proteomic changes in reaction to protein folding stress. More importantly, validation of the method or findings is missing. Perhaps the collected data could be of some use if made available.

Major comments

1) The manuscript relies on several assumptions, which are not convincing to me and validation is lacking. E.g. how do the authors account for possible (and reported) changes in ER morphology upon induction of the UPR? These could explain the observed changes in proteins localizing to different fractions without them being in different subcellular compartments. ER protein import is co-translational. How would the observed differences be achieved, in a SRP-independent manner?

2) There needs to be some form of validation that the method indeed identifies protein in the ER (or other compartments). How reliable is the fractionation method? Some digestion method (e.g. proteinase K) needs to be applied to distinguish proteins inside the organelle from proteins bound to the organelle.

3) TMT experiments shot by MS2 have inherent issue with ratio compression. Consequently, the field has moved to MS3 measurements, including the cited paper (Dephoure 2012) describing SILAC plus TMT approaches. Why were the samples not shot with MS3? How was ration compression controlled?

4) To my understanding (a clear description in the figures or legends is missing), the SILAC-TMT data are based on duplicates. Considering that the manuscript is entirely based on proteomics data, duplicates are not sufficient. P-values must not be calculated from duplicates. Whether p-values have been corrected, as required for such experiments, is not apparent from the manuscript.

5) It is difficult to judge the validity of the SPLAT method without any experimental validation.

6) Relocalization studies require more validation in order to validate the accuracy of the assay. E.g. do known UPR mediators show expected relocalization? Biochemical validation of mentioned translocator candidates that show prominent relocalization is required.

7) Half-lives and turnover rates are calculated based on one time point (in duplicate) that is much earlier than the calculated half-lives. This is not sufficient. Proper time-courses covering beyond 10 days are required to calculate half-lives. This is particularly important since the authors claim that the single time points used are based on protein half-lives, which consequently need to be accurate.

8) ER stress is a transient response that typically shuts itself off via feedback loops since prolonged

activation induces apoptosis. Why were such a long treatment time chosen (16h)? Does it lead to cells death? Are UPR markers of the integrated stress response (i.e. eIF2alpha phosphorylation, detectable ATF4 and CHOP protein levels) still detectable by Western blot? This issue is even more pronounced for the conditions of inhibiting the proteasome for 48h. Is there any induction of cell death? Considering the surprising lack of changes in protein degradation, experiments are required to show that the proteasomal function is actually inhibited over the time span of 48h. Is there a concurrent induction of autophagy as the other major protein degradation pathway in cells?

9) UPR activation leads to an extensive shutdown in protein translation via the integrated stress response. How do the observed changes in translation compare to published observation and how is this difference accounted for in the analyses?

Minor and specific comments

Figure legends have to explain abbreviations and give number of replicates.

Supplemental tables should have a cover page with a title, figure cross-reference and explanation of the columns.

Labeling in figures should be consistent (e.g. Fig. 2A, B).

Panels in figures do not appear in the order as they are discussed in the results section, please reorder.

137: add reference for LOPIT approach, revise sentence

187-192: some fold changes do not match the median values shown in Figure S1

235-236: unclear what the authors are trying to say

256-259: include references for suppression of protein synthesis through ribosome remodeling

268-271: where do the numbers (FC, p-values) come from, please make accessible in a table.

Proteins mentioned here are not shown in the figure.

280-285: discussed data is not shown

315-326: discussed data is not shown in a figure or provided as table

355-356: tunicamycin data are not shown. Either show data or remove from the results section.

356: refer to Fig. 4A

360: typo 'sediment'

367-368: add references for 'previously known translocators'

367-371: proteins mentioned here are not the ones highlighted in the figure

380: refer to Fig. S3A

380-383: where can this data be found?

408-410: discussed data is not shown

412-414: refer to Fig. 4A

412-417: most mentioned proteins are not shown in the figure. What is the interpretation, could this be explained by autophagy activation?

419: refer to Fig. S3B

429-430: give examples of stress response proteins that are found in this group

462-466: where can this data be found?

468-470: where can this data be found?

477-479: tunicamycin data is not shown

493-503: an alternative explanation would be folding stress induced stalling of protein trafficking along the secretory pathway, which is even stated in the abstract (line 37-40)

519-520: title is somewhat misleading because the utilized PI concentration did not cause sufficient inhibition of proteasomal degradation, as shown in Fig. 6B

536-537: add references for prior reports

538: typo 'hands'

564-567: refer to Fig. 6D

567-569: are the protein half-lives provided anywhere?

573-574: how many of these proteins showed differential localization upon stress?

642-644: immunofluorescence cannot distinguish between old/new protein pool; hence, these results do not support the finding that only the young protein pool shifts location. Please rephrase.

644-649: please be careful with interpretation of the results (as stated for lines 493-503).

Otherwise provide additional data that support the 'ligand-independent EGFR internalization' hypothesis.

675-676: show data or give reference

Figure 1B: think about emphasizing temporal information gained in MS1 and spatial information gained in MS2

Figure legend 1B: not clear that 'SILAC light' (used in figure) is the same as 'unlabeled' in the legend. Not clear that '+R[10.0083]' (used in figure) refers to heavy Arg.

Figure 2B: what was the reason to test against ribosomal proteins?

Figure 2C: it is unclear how one time point is sufficient for such analyses

Is the data shown in Figure 2 provided as a table?

Figure legend 2B: which statistical test was used?

Figure 3C is not discussed in the results section

Figure 4: ER stress is known to cause ER membrane expansion and remodeling. This might alter the sedimentation behavior of the ER fraction in the LOPIT-DC protocol and provide an explanation for the appearance of the peroxisome/ERV fraction.

Figure 4B: include LOPIT color code legend

Figure 5A: include LOPIT color code legend

Figure 5C: scale bar missing

Figure legend 5D: Figure does NOT include 'associated subcellular localization changes' as stated in the legend. Please remove.

Figure 6C: what was the reason to test against ribosomal proteins?

Supplemental table S1 does not include data with tunicamycin

Reviewer #3 (Remarks to the Author):

The present study utilized a combination of in vivo dynamic SILAC metabolic labeling and TMT labeling of spatially separated fractions, known as the SPLAT strategy. Initially, these techniques were applied to investigate human AC16 cells, a transformed cardiomyocyte line, as well as induced pluripotent stem cell (iPS)-derived cardiomyocytes. The authors focused on studying the effects of two common unfolded protein response (UPR) models, thapsigargin and tunicamycin, on the proteome in AC16 cells. They observed a more severe slowdown in lysosome protein turnover compared to the endoplasmic reticulum. Additionally, the SPLAT strategy was employed to study a cardiotoxicity model using human iPS-derived cardiomyocytes exposed to the proteasome inhibitor carfilzomib. This analysis revealed potential disruptions in carbohydrate metabolism and contraction proteins as mechanisms underlying carfilzomib-induced cardiotoxicity. An advantage of the SPLAT strategy is its ability to simultaneously encode temporal and spatial protein information through isotope labels in the MS1 and MS2 levels.

1) Figure 1, the number of independent experiments and the observed variation between independent replicates were not explicitly mentioned. It would be beneficial for the authors to provide this information to better understand the reliability and reproducibility of the results.

2) In Figure 2B, it is highlighted that the protein turnover change is highest in the nucleus/chromatin compartment. However, there may be concerns about potential misannotation since the heavy labels in Figure 1E show significant overlap between nuclei/chromatin and ribosomes. The authors should address this issue and provide further clarification to ensure accurate annotation of the compartments.

3) The "t" in the legend of Figure 2B likely refers to the "t-test." It is important for the authors to mention whether they performed adjustments for multiple comparisons?

4) To address the potential for cross-contamination during fractionation of subcellular compartments, it would be valuable to confirm more translocation events using immunostaining

techniques. For example, in addition to EGFR, the authors should consider examining the translocation of newly synthesized cathepsin A from the lysosome to the peroxisome (Suppl Figure 4) with tunicamycin and the translocation event of newly synthesized ITGAV to the ER fraction in the presence of thapsigargin (Suppl Figure 5).

5) Since the fractionation of subcellular compartments is achieved through ultracentrifugation, it is necessary to demonstrate whether the isolation process preserves the integrity of cellular organelles under protein misfolding stress or cardiotoxicity, particularly in comparison to untreated cells.

6) Considering the authors' previous publication on cardiac remodeling (Nat Commun 2018) and the reported similarity of turnover among interacting partners, it would be valuable to compare the protein turnover in AC16 cells to that in mouse hearts. This comparison would provide insights into the generalizability of the results from AC16 cells. Alternatively, similar experiments should be conducted in other cardiomyocyte-like cell types, such as iPS cells.

7) The phenomenon of "end-membrane stalling" is intriguing, but it is important to explore the functional consequences associated with it. Additionally, the finding that newly synthesized proteins are more likely to translocate could be explained by their lack of incorporation into protein complexes. The authors should explore these potential functional consequences and provide further insights into the implications of their findings.

8) The findings regarding the mechanism of carfilzomib cardiotoxicity would benefit from in vivo validation. It is important to investigate whether the disruptions of carbohydrate metabolism and contraction proteins can be reversed or attenuated to mitigate cardiotoxicity. In vivo experiments would provide a more comprehensive understanding of the therapeutic implications.

Minor:

- When mentioning the potential impurities in TMT tags leading to spillover to neighboring channels, it would be helpful for the authors to include a reference or provide supporting evidence for this statement.
- There is a typo in the phrase "proteins turnover rate." It should be "protein turnover rate."
- In Figure 5C, the confocal images of EGFR lack isotype controls. The authors should consider including isotype controls to ensure proper interpretation of the results.

Reviewer #1 (Remarks to the Author):

Currie and co-authors present SPLAT, an integrated spatial proteomics and turnover analysis approach. The idea of SPLAT has been desired for many years amongst the spatial proteomics community but has remained elusive because of some of the technical difficulties in the processing of the data and the care needed to ensure reproducibility. The method and data presented are very interesting and are well show-cased in the manuscript, which I enjoyed reading. There are a few limitations that are worth addressing but I believe the manuscript is a sufficient advance that additional experiments would add value but are not needed.

General comments

The results on SCL3A2 are interesting - did the authors look at microscopy analysis of this translocation?

Did the authors perform any analysis to check that pulsed SILAC strategy did not introduce any confounding effects? For example, control with and without pulsed SILAC?

I must confess that I am not an expert on ER stress, why was 16 hours chosen as the relevant time point? Was this decided through some initial experimentation?

Are the light and heavy samples fractionated together, or just the MS is at the same time?

The authors mention that TMT impurities may cause issues. I did not fully understand what was done to the data when the authors mention corrections in lines 159-162. Perhaps the authors could expand in the methods section? On a related note, the authors can randomize the TMTs across the spin speeds to avoid some of this confounding so that neighboring TMTs are not neighboring spin speeds. Could the authors clarify which TMTs went with which spin speeds?

The narrow isolation window and extensive offline fractions was sensible if SPS-MS3 was not used. Could the authors clarify the pooling strategy used for the 20 peptide fractions?

Bioinformatics analysis

The authors mention that an 0.99 threshold was used, but it is not clear what this means for an FDR. This can be calculated as the average error rate above the chosen threshold.

There are quite some number changes which may be difficult to prioritize. Some strategies that can help:

- 1) Bandle does not explicitly take into account the paired design, so you can run the method pairwise for each replicate across conditions.
- 2) Bandle is better calibrated when thresholding also on the outlier probability
- 3) Did the authors check that for differential localisation that the same (and enough) peptides were measured for both samples. Heuristically, having at least two peptides that were quantified in both is desirable.

- 4) The authors can run bundle between replicates and filter any hits out of these out of differential localisation results.= - this is especially useful when there are only two replicates and so the results are poorly averaged over replicate-replicate variability.
- 5) One must be careful about unannotated compartments in these analyses. Did the authors check whether exosome or endosome markers are enriched in their list of differential localisations?

Minor Comments:

- 1) The reference to the LOPIT-DC paper is missing on page and I think this acronym should be spelled out on first use.
- 2) \log is a symbol and so is always lower case, some figures have uppercase L. In fact, Log is something else mathematically
- 3) Figure 2 the p-values are difficult to read
- 4) Figure 3A Replicate 2 Tunicamycin the PCA plot is mirrored, this happens because PCA is invariant to reflection. You should be able to specific `mirrorY = TRUE` to fix this
- 5) Figure B and C, I think using the same diverging color scale is confusing and C would definitely benefit from a gradient rather than have 0 as white.

Oliver Crook (University of Oxford)

Response to Reviewers

Reviewer #1 (Remarks to the Author):

“Currie and co-authors present SPLAT, an integrated spatial proteomics and turnover analysis approach. The idea of SPLAT has been desired for many years amongst the spatial proteomics community but has remained elusive because of some of the technical difficulties in the processing of the data and the care needed to ensure reproducibility. The method and data presented are very interesting and are well show-cased in the manuscript, which I enjoyed reading. There are a few limitations that are worth addressing but I believe the manuscript is a sufficient advance that additional experiments would add value but are not needed.”

Response: We thank the Reviewer for the supportive comments and for sharing our enthusiasm on the study. In response to the comments, we have incorporated major improvements to the manuscript, including:

- We now include a BUNDLE thresholding strategy for outliers, estimated FDR, bootstrapping estimate, to prioritize differential localization events
- We now use a two-peptide rule for TMT data for localization analysis as recommended.
- We have performed microscopy analysis of SLC3A2 translocation as recommended.

In our view the Reviewer’s helpful suggestions have led to a much-improved manuscript. Please see below for our itemized responses.

General comments

Reviewer 1 Comment 1: *“The results on SCL3A2 are interesting - did the authors look at microscopy analysis of this translocation?”*

Response: In response to the Reviewer’s comments, we have now performed a new microscopy analysis using anti-SLC3A2 antibodies in untreated and thapsigargin-treated human AC16 cells. The microscopy analysis corroborates a change in localization as reflected by a decrease in co-localization index with anti-LAMP2 signals (lysosome marker). The data are now included in the **new Figure panels 3D–E** in the revised manuscript.

Figure 3D. Immunofluorescence of SLC3A2 (red) against the lysosome marker LAMP2 (green) and DAPI (blue). Numbers in cell boundary: colocalization score per cell. Scale bar: 90 μm . **E.** Colocalization score (Mander's correlation coefficient) between SLC3A2 and LAMP2 decreases significantly (t-test P value: $3e-8$) following thapsigargin treatment, consistent with movement away from lysosomal fraction ($n = 205$ normal cells, $n = 32$ thapsigargin treated cells).

In addition, the main text has been modified to refer to the new figure panels:

Main text, Results, line 310 “The change in localization of SLC3A2 is corroborated by immunostaining (**Figure 3D**), which shows a decrease in co-localization between SLC3A2 and lysosome marker LAMP2 upon thapsigargin treatment (**Figure 3E**).”

Reviewer 1 Comment 2: “Did the authors perform any analysis to check that pulsed SILAC strategy did not introduce any confounding effects? For example, control with and without pulsed SILAC?”

Response: We have not observed any difference that pulsed SILAC has introduced to the AC16 cells in terms of proliferation rate or viability. Since the SILAC labeled K/R amino acids are added back to K/R depleted DMEM/F12 media to match the normal concentration of lysines and arginines found in normal DMEM/F12 media precisely (i.e., no total change in amino acid concentrations), we do not expect there to be severe confounding effects introduced by pulsed SILAC.

To explore this issue further in response to the Reviewer's comments, we have performed a new analysis using an orthogonal method to measure the turnover rates of proteins in human AC16 cells and assess whether the SILAC labeled K/R led to differences in overall label utilization. Heavy water (D_2O) labeling presents an orthogonal method to pulsed SILAC to measure the turnover rates of proteins while avoiding potential issues such as stimulation of protein synthesis from free amino acids (e.g., see our prior publications (Hammond et al., 2022; Lau et al., 2018). In our hands from replicate analysis of human AC16 cells labeled with 6% D_2O ($n=5$), we find very similar protein half-life distributions between D_2O and SILAC, with the median protein half-life in D_2O labeling being 19.1 – 19.9 hours (compared to 19.7 hours among common protein groups in SILAC). Likewise, we observed very similar half-life distributions in thapsigargin-treated samples with the median protein half-life in D_2O labeling being 37.2 – 39.6 hours (compared to 37.9 hours among

common proteins in SILAC) (see **Reviewer Figure R1**). These results are consistent with minimal confounding effects from pulsed SILAC on cellular protein dynamics.

Reviewer Figure R1: Histograms of \log_{10} protein turnover rates measured from heavy water labeling, which measures protein isotope incorporation kinetics without introducing free amino acids unlike pulsed SILAC. Rows: control vs. thapsigargin treated AC16 cells; columns: replicates ($n=5$); black dashed line: median half-life in D_2O across the replicates; red: median half-life in SILAC among shared proteins.

Reviewer 1 Comment 3: “I must confess that I am not an expert on ER stress, why was 16 hours chosen as the relevant time point? Was this decided through some initial experimentation?”

Response: The dosages and durations were determined based on accepted range in the literature to model the effect of ER stress on cellular pathologies in cardiac and cardiac-like cells.

For thapsigargin, a treatment of 12 to 18 hours in the $\sim 1 \mu\text{M}$ range was used with minimal loss in cell viability (Ghosh et al., 2023; Stoner et al., 2020); whereas treatment of 24 hours or beyond have been observed to induce apoptosis (Chen et al., 2019; Stoner et al., 2020). Specifically, Stoner et al. used $0.5 \mu\text{M}$ thapsigargin for 18 hours in human AC16 cardiac cells to induce ER stress and observed increased XBP1 and ATG6 expression; whereas Ghosh et al. treated AC16 cells for $0.5 \mu\text{M}$ thapsigargin for 16 hours in human AC16 cardiac cells to induce UPR and show upregulated PERK and XBP1 (Ghosh et al., 2023). Chen et al. used $1 \mu\text{M}$ thapsigargin in rat H9c2 cardiac myotubes and primary neonatal rat cardiomyocytes for 12 hours to induce ER stress as evident by BiP induction (Chen et al., 2019). On the other hand, $1 \mu\text{M}$ thapsigargin treatment for 24 hours in induced apoptosis in two studies (Chen et al., 2019; Stoner et al., 2020).

For tunicamycin, a range of timepoints from 4 to 18 hours have been used in the literature, and we have opted to stay within this range while maintaining a 16 hour treatment window to allow drug treatment and heavy SILAC amino acids to be applied concurrently, and for ease of comparison with thapsigargin. For instance, Palomar et al. used a higher dose of $5 \mu\text{g}/\text{mL}$ for 4 hours in human AC16 cardiomyocytes to induce ER stress as evident by BiP and ATF3 (Palomar et al., 2014). Toro et al. used $2 \mu\text{M}$ ($\sim 1.6 \mu\text{g}/\text{mL}$) for 6 hours and 18 hours in human AC16 cardiomyocytes to induce ER stress as evidenced by BiP (GRP78) increase (Toro et al., 2022). Siltanen et al. used up to $0.3 \mu\text{g}/\text{mL}$ tunicamycin for 24 hours in primary fetal rat cardiac cells to activate protective UPR (Siltanen et al., 2016). On the other hand, Liu et al. used $1 \mu\text{g}/\text{mL}$ tunicamycin

for 36 hours to induce apoptosis in cultured neonatal rat cardiomyocytes with about 30% drop in viability (Liu et al., 2012).

The doses and duration we used are therefore within the range of cardiac cells in the literature. We observed a loss of proliferation rate but not viability. In addition, in the revised manuscript, we now show that the thapsigargin and tunicamycin led to statistically significant induction (limma FDR adjusted $P < 0.01$) of established ER stress markers (Glembotski, 2007), including the major canonical marker BiP (GRP78/HSPA5), ASNS, HERPUD1, HSP90B1, PDIA4, and PTX3 in thapsigargin; and BiP, ASNS, CALR, CANX, DNAJB11, DNAJC3, HERPUD1, HSP90B1, HSPA13, HSPA5, and PDIA4 in tunicamycin. These results support the ER stress induction dosage and time point and are now presented in **Figure 2A** and **Figure 5A** in the revised manuscript.

Figure 2A. Bar charts showing activation of known ER stress markers upon thapsigargin treatment for 8 hours and 16 hours. X-axis: ER stress markers; y-axis: expression ratio ($n=6$ normal AC16; $n=3$ thapsigargin). *: limma adjusted $P < 0.01$; **: limma adjusted $P < 0.001$; ***: limma adjusted $P < 0.0001$.

Figure 5A. Bar charts showing activation of known ER stress markers upon tunicamycin treatment for 8 hours and 16 hours. X-axis: ER stress markers; y-axis: expression ratio ($n=6$ normal AC16; $n=3$ tunicamycin). *: limma adjusted $P < 0.01$; **: limma adjusted $P < 0.001$; ***: limma adjusted $P < 0.0001$.

Reviewer 1 Comment 4: “Are the light and heavy samples fractionated together, or just the MS is at the same time?”

Response: With the dynamic SILAC labeling strategy we have employed, the cells were cultured in light media and then swapped to heavy media for 16 hours (for AC16 cells) or 48 hours (in for iPSC-CMs). Since the majority of the protein pools are not labeled to saturation, we see heavy and light peaks for a given peptide species from the same sample. In this way the heavy and light proteins that came from the same cell culture are digested, tagged, and fractionated together.

Reviewer 1 Comment 5: *“The authors mention that TMT impurities may cause issues. I did not fully understand what was done to the data when the authors mention corrections in lines 159-162. Perhaps the authors could expand in the methods section? On a related note, the authors can randomize the TMTs across the spin speeds to avoid some of this confounding so that neighboring TMTs are not neighboring spin speeds. Could the authors clarify which TMTs went with which spin speeds?”*

Response: Because the TMT isobaric tags are not 100% isotopically pure, up to ~8% of the channel intensity from a tag can be found to spill over to neighboring channels. The isotopic contamination matrix is provided by Thermo for each lot of TMT, which can be used to calculate the true channel intensities from the observed channel intensity (Searle and Yergey, 2020). This is done in this study using `scipy.optimize.nnls` in the `pytmt` module of the SPLAT pipeline. Following the Reviewer’s comment, the contaminant matrices have now been reproduced in the Supplemental Information (new **Supplemental Table S1**)

TMT tags were randomized in our study using a random number generator. The channel assignment metadata was given in the shared data on ProteomeXchange, but is now reproduced in the manuscript as well following the Reviewer’s comment (new **Supplemental Table S3** and **Supplemental Table S4**). The impurity correction helps improve reproducibility when the tag assignment in each replicate is randomized differently, as is the case in this study, because the different degrees of confounding from neighboring channels that represent different ultracentrifugation fractions in each replicate is corrected. We now clarify this with the following modification in the main text:

Main text, Results, line 150 *“Because the TMT data are row normalized in the LOPIT-DC design, we incorporated correction of isotope contamination of TMT channels based on the batch contamination data sheet (**Supplemental Table S1**) to account for isotope impurity in fractional abundance calculation from randomized channels across experiments (**Supplemental Methods**).”*

Supplementary Information, Supplementary Methods, Line 26 *“TMT-10 plex lots were #WF309595 for control AC16 replicate 1 and 2; thapsigargin treated AC16 replicate 1, and tunicamycin treated AC16 replicate 1; and #XB318561 for other samples. Isotope impurities in TMT tags can lead to up to 10% spillover to neighbor channels and decrease quantitative accuracy (**Supplemental Table S1**). We used the non-negative least square algorithm in `scipy` (Virtanen et al., 2020) to solve for the true channel matrix from the observed channel intensity and impurity matrix for downstream quantification.”*

Reviewer 1 Comment 6: *“The narrow isolation window and extensive offline fractions was sensible if SPS-MS3 was not used. Could the authors clarify the pooling strategy used for the 20 peptide fractions?”*

Response: Although high-pH reversed phase fractions were collected every minute, the UV chromatogram (see **Reviewer Figure R2**) generated during offline fractionation suggested some fractions contained less peptide content than others. When this was the case, adjacent fractions were pooled for injection into the second-dimension low-pH reversed-phase separation prior to mass spectrometry. Specifically, the following high-pH reversed phase fractions were combined, where each number corresponds to the minute in the HPLC gradient: 1-3; 4-6; 7-9; 10-12; 30-31; 32-33; 34-35; 36-37; 38-39; 40-41; 42-46. All other fractions were subjected to LC-MS individually. This scheme was maintained for each experiment.

Reviewer Figure R2: Representative high-pH reversed phase UV chromatogram for two-dimensional peptide fractionation. *x axis: retention time in minutes. y axis: UV 220 nm (AU). Dashed line: gradient of %B. Fraction was collected every minute as in the x-axis with negligible post-detector delay.*

Bioinformatics analysis

Reviewer 1 Comment 7: *“The authors mention that an 0.99 threshold was used, but it is not clear what this means for an FDR. This can be calculated as the average error rate above the chosen threshold. There are quite some number changes which may be difficult to prioritize. Some strategies that can help: Bandle does not explicitly take into account the paired design, so you can run the method pairwise for each replicate across conditions.”*

Response: We thank the Reviewer for the helpful suggestions. We now calculate the estimated FDR for the chosen threshold as the Reviewer suggested. Based on the estimated FDR, we now use a differential localization probability threshold of 0.95, and also a bootstrap differential localization probability threshold of 0.95.

In the thapsigargin comparison, 1,306 protein features are differential localization candidates, including 687 light proteins and 619 heavy proteins, and the estimated false discovery rate is approximately 0.0018 (0.18%). In the tunicamycin comparison, 588 protein features (including 263 light proteins and 325 heavy labeled proteins) are differential localization candidates, and the estimated false discovery rate is approximately 0.0036 (0.36%). In the carfilzomib comparison, 1,625 protein features (including 767 light proteins and 858 proteins) are differential localization candidates, and the estimated false discovery rate is approximately 0.0038 (0.38%).

To further prioritize the targets, we have further highlighted a subset of high-confidence proteins where both the light and heavy protein features show clear translocation based on the thresholds above, which result in 330 protein pairs (i.e., 330 light + 330 heavy counterpart) in the thapsigargin comparison, 109 pairs in the tunicamycin comparison, and 339 pairs in the carfilzomib comparison.

Main text, Results, Line 289

*“In total, we identified 1,306 translocating protein features (687 light and 619 heavy) in thapsigargin under a stringent filter of BUNDLE differential localization probability > 0.95 with an estimated FDR of 0.0018 (0.18%), and further filtered using a bootstrap differential localization probability of > 0.95. We then further prioritized 330 pairs of differentially localized proteins where the light and heavy features both show confident differential localization (**Supplemental Data S2, Supplemental Data S4**).”*

Main text, Results, Line 481

“Parallel to the less prominent changes in vesicle transport, tunicamycin treatment also led to fewer translocating proteins than thapsigargin, with 620 translocating features (including 282 light proteins and 338 heavy proteins) at BUNDLE differential localization probability > 0.95, corresponding to an estimated FDR of 0.35%, and thresholded by bootstrapping differential localization probability > 0.95; from which we highlighted 109 proteins where the heavy and light versions both showed translocation”

Reviewer 1 Comment 8: *“Bandle is better calibrated when thresholding also on the outlier probability”*

Response: We thank the Reviewer for the helpful suggestion. We now perform thresholding also on the outlier probability as recommended by the Reviewer, which is applied at 0.95 localization threshold and 0.95 outlier threshold for all locations in untreated cells. To gain more insight into potential partial localizations, we used a 0.9/0.5 threshold for stimulated cells. This did not change the differential localization estimates but had an effect on whether a protein is shown to be differentially localized to an assigned compartment vs. the unclassified subset. In the figures we now give the probability of allocation in select cases after manual inspection.

Reviewer 1 Comment 9: *“Did the authors check that for differential localisation that the same (and enough) peptides were measured for both samples. Heuristically, having at least two peptides that were quantified in both is desirable.”*

Response: We thank the Reviewer for the suggestion and have now included a two-peptide rule for admitting proteins for analysis. Expectedly this led to a substantial decrease in the number of common features (by ~33% from 5630–6444 protein features to 3617–4407 protein features in each replicate) that can be analyzed. While we have followed the Reviewer’s suggestion here, we also believe that the two-peptide rule has been debated in other proteomics applications and in some instances may be considered overly conservative (Gupta and Pevzner, 2009). While the main conclusion of the study is not affected by the filter, some of the highlighted examples have been updated as a result. Future work to re-analyze the data generated here may be able to recover further changes from the dataset.

Reviewer 1 Comment 10: *“The authors can run bandle between replicates and filter any hits out of these out of differential localisation results. This especially useful when there are only two replicates and so the results are poorly averaged over replicate-replicate variability.”*

Response: We thank the Reviewer for the helpful suggestion. Since we have now re-created all the analysis using three replicates, this particular measure was not pursued in this instance.

Reviewer 1 Comment 11: “One must be careful about unannotated compartments in these analyses. Did the authors check whether exosome or endosome markers are enriched in their list of differential localisations?”

Response: We fully agree and thank the Reviewer for an important suggestion. We compared proteins in each assigned compartment in each sample against prior annotations, in order to verify the general performance of the classification as well as identify potentially unannotated compartments in AC16 cells and in human iPSC-cardiomyocytes.

Figure 2C. Distribution of light (unlabeled) protein features in each of the 12 subcellular compartments ($n=3$); fill color represents whether the protein is also annotated to the same subcellular compartment in UniProt Gene Ontology Cellular Component terms.

In normal AC16 cells, approximately 70% of proteins are assigned to localizations that match their known prior annotations. We believe this is a reasonable number given GO CC annotation is incomplete and lacks cell type specific contexts. Most compartments contain proteins with matching annotations, with the exceptions that many ER and GA annotated proteins are interspersed across the two compartments, perhaps indicative of the interconnectedness of ER and Golgi in the cells; and the lysosome compartment. A closer inspection shows that in lysosome compartment 93 out of 140 proteins (66.4%) have the “extracellular exosome [GO:0070062]” cellular component term. In addition, proteins annotated with the “extracellular exosome [GO:0070062]” cellular component term can also be found in the cytosol, where 151 out of 189 (44.4%) of the classified light (i.e., without SILAC heavy label) proteins have exosome annotation. Although it is likely that the lysosome fraction contains multiple unannotated endomembrane compartments, because exosomal proteins would be expected to also reside in other cellular locations when they are not packaged for secretion, we refrain from renaming the lysosome compartment or making claims about whether these annotations represent true unannotated compartment within the fraction.

In parallel, we find that 30 out of 49 (61.2%) of proteins in the peroxisome compartment contained endosomal annotations (early endosome [GO:0005769]; early endosome membrane [GO:0031901]; endosome [GO:0005768]; endosome membrane [GO:0010008]; recycling endosome [GO:0055037]), even though the fraction was trained with only peroxisome makers. We refer to this compartment as the peroxisome/endosome in the results section (new **Supplemental Figure S5**). However, the proteins migrating to this compartment themselves are not enriched in exosome/endosome terms.

Supplemental Figure S5. Correspondence of spatial classification with prior annotations in stress cells. As related to main Figure 2C, the bar charts show the number of light (i.e., non-heavy-SILAC labeled) proteins (y-axis) classified to each of 12 subcellular locations (x-axis) in thapsigargin (left) and tunicamycin (right) treated AC16 cells (n=3). The colors represent whether proteins classified to each subcellular location are also known to reside in the subcellular component of question in Gene Ontology Cellular Component terms retrieved from UniProt. In normal, thapsigargin, and tunicamycin treated AC16 cells, 69.5%, 71.9%, and 63.0% of classified proteins are consistent with known annotations, respectively; hence the classified subcellular localization match the expected assignments from prior knowledge and are not substantially affected by cellular stressors. The expanded peroxisome compartment in stressed AC16 cells primarily contained non-peroxisome annotated proteins that co-sedimented with the trained peroxisome compartment, and are referred to as the peroxisome/endosome compartment in the manuscript.

Lastly, in response to the Reviewer's comments above, we have made every effort to refine the bioinformatics analysis including compartment classification. The CYTOSOL compartment is now divided to create a new small fraction that shows a distinct sedimentation profile (new **Supplemental Figure S2A**) and that is separable in the first several principal components. Inspection of the proteins assigned to this compartment show that it contains a majority of proteins with "cytoplasm [GO:0005737]" terms (63 out of 87), proteins with cytoskeleton and actin filament terms, as well as potentially dually localized proteins without both cytosolic and nuclear annotations. We refer to this compartment as the CYTOPLASM in the manuscript to distinguish from the cytosolic compartment. Similarly, a new CHROMATIN fraction is defined using markers from the Mulvey et al. 2021 LOPIT data (Mulvey et al., 2021) followed by manual curation to select for nuclear/chromatin proteins that show a distinct sedimentation behavior (primarily sedimented at the first ultracentrifugation step). The new map containing 12 fractions led to better separation of clusters and higher confidence of localization assignment.

Supplemental Figure S2. Ultracentrifugation fraction distributions of cellular component markers. Replicate one of each experimental condition is shown. The line plots show the normalized abundance (y-axis) of marker proteins for each

subcellular localization experiment across ultracentrifugation fractions (x-axis) as measured by the TMT channel intensities. The fractions correspond to the ultracentrifugation steps in Supplemental Table S3. Colors correspond to spatial maps for AC16 cells throughout the manuscript. Black lines show average trend line and standard deviation, showing consistent sedimentation profiles of subcellular localization in **A**. normal cells (n=3).

Likewise, we have taken extra effort to revisit the marker assignment of the iPSC-CM data set by manual curation with the aid of pRoloc methods. We have identified multiple cell junction and desmosome proteins including JUP, DMD, FLOPT1, and PDLIM5 from a “hidden compartment” that are co-localized with similar ultracentrifugation profiles as lysosome markers. These proteins have now been added as markers for a LYOSOME/JUNCTION compartment. Likewise, multiple sarcomeric proteins can be identified in the iPSC-CM spatial maps including MYH4, MYH6, and TNNT2. These sarcomeric proteins share similar sedimentation profiles as chromatin proteins, with highest relative abundance in the first centrifugation step. They have been added as a marker for a combined CHROMATIN/SARCOMERE compartment. Lastly, we have separated the 40S and 60S ribosomes into two clusters based on their distinct sedimentation behaviors. The new markers can be found in **Supplemental Data S10**, and the sedimentation profiles in **Supplemental Figure S13**.

Supplemental Figure S13. Ultracentrifugation fraction distributions of cellular component markers in human iPSC-CMs. Additional iPSC-CM specific compartments and markers were curated manually, including a sarcomere and a cell junction compartment. The compartments were merged with the chromatin and the lysosome compartments due to similarity in sedimentation profile under the present ultracentrifugation scheme. Replicate one of each experimental condition is shown. **A**. Control iPSC-CM. The line plots show the normalized abundance (y-axis) of marker proteins for each subcellular localization experiment across ultracentrifugation fractions (x-axis) as measured by the TMT channel intensities. The fractions correspond to the ultracentrifugation steps in Supplemental Table S3. Colors correspond to spatial maps for iPSC-CMs cells throughout the manuscript. Black lines show average trend line and standard deviation, showing consistent sedimentation profiles upon carfilzomib treatment.

Minor Comments

Reviewer 1 Comment 12: “The reference to the LOPIT-DC paper is missing on page and I think this acronym should be spelled out on first use.”

Response: Thank you. This has now been updated.

Main Text, Results, Line 127 *“In particular, the LOPIT-DC (Localisation of Organelle Proteins by Isotope Tagging after Differential ultraCentrifugation) method (Geladaki et al., 2019) allows the advantageous use of sequential ultracentrifugation to enrich different subcellular fractions from the same samples...”*

Reviewer 1 Comment 13: *“\log is a symbol and so is always lower case, some figures have uppercase L. In fact, Log is something else mathematically.”*

Response: Thank you. We have now checked the manuscript to ensure all the axis titles read log rather than Log where applicable (e.g., **Figure 2E**, **Figure 5C**, **Figure 6M**, and **Supplemental Figure S6**).

Reviewer 1 Comment 15: *“Figure 2 the p-values are difficult to read”*

Response: We agree with the Reviewer and have remade this figure for clarity. Each new facet in the new panel **Figure 2E** now highlights only the comparison between a single compartment and their complement rather than multiple cross-compartment comparison. The values should hopefully be now more readable.

Reviewer 1 Comment 16: *“Figure 3A Replicate 2 Tunicamycin the PCA plot is mirrored, this happens because PCA is invariant to reflection. You should be able to specific mirrorY = TRUE to fix this”*

Response: We thank the Reviewer for the helpful recommendation and now use mirrorY/mirrorX where applicable for consistent spatial map orientation.

Reviewer 1 Comment 17: *“Figure 3B and C, I think using the same diverging color scale is confusing and C would definitely benefit from a gradient rather than have 0 as white.”*

Response: We agree with the Reviewer that the colors and these figure panels are confusing. In the revision, we have given more consideration to color choices. The specific figure panels in question have been updated as the new **Figure panels 4A–C** to more clearly present the consistent localization of heavy and light proteins

Oliver Crook (University of Oxford)

We thank Dr. Crook again for the many helpful suggestions.

Reviewer #2 (Remarks to the Author):

“The manuscript ‘Simultaneous proteome localization and turnover analysis reveals spatiotemporal features of protein homeostasis disruptions’ by Currie et al. establishes SPLAT, a mass spectrometry based proteomics approach that links the temporal component of protein turnover with spatial information of protein localization by combining heavy isotope labeling by SILAC with subcellular fractionation and tandem mass tag (TMT) labeling. The authors use this approach to decipher organelle-dependent differences in protein turnover rates upon ER stress, and to identify stress-induced localization changes of pre-existing and newly synthesized proteins during stress. Lastly, the authors apply SPLAT to a cellular cardiotoxicity model to investigate temporal and spatial proteome changes upon treatment with the proteasome inhibitor carfilzomib at cardiotoxic concentrations.

The manuscript covers a very interesting topic and the authors combine their proteomics analyses with a wide range of computational approaches. However, the overall novelty remains limited. While the SPLAT technique is a useful tool to monitor dynamics in cellular stress responses, the data interpretation in this manuscript does not provide sufficient new mechanistic insights on proteomic changes in reaction to protein folding stress. More importantly, validation of the method or findings is missing. Perhaps the collected data could be of some use if made available.”

Response: We thank the Reviewer for their evaluation of our manuscript and for the constructive critiques. We believe the Reviewer’s comments have led to a much improved study. In response to the Reviewer, we have incorporated major improvements to the manuscript, including but not limited to the following:

- We have performed new replicate mass spectrometer experiments for normal, thapsigargin, and tunicamycin treatment to bring the total number of replicate SPLAT/LOPIT experiments to three.
- We have performed a new set of experiments to directly compare MS2 and MS3 based TMT quantification for spatial proteomics data acquisition.
- We have performed extensive additional analysis to ensure reliability and reproducibility of the method, including spatial distance calculations of light/heavy protein pairs, comparison of spatial predictions to Gene Ontology annotations, new immunofluorescence validation, and analysis of expected translocation events (e.g. EIF3 subunits and CAV1 in AC16, PSME4/PA200 and BAG3 in iPSC-CM).
- We have performed new validation experiments of the cell stress models, showing the expected activation of known ER stress markers upon thapsigargin and tunicamycin; and acquired new data on cell viability and mitochondrial function assays at multiple doses of carfilzomib in the human iPSC-CM; and proteasome activity and autophagy assays to demonstrate proteasome inhibition and cardiotoxicity.
- We provide new analysis to demonstrate the reliability and reproducibility of cellular compartment fractionation in normal and stressed cells.

In our view the Reviewer’s helpful suggestions have led to a much improved manuscript. Please see our itemized responses below.

Major comments

Reviewer 2 Comment 1: *“The manuscript relies on several assumptions, which are not convincing to me and validation is lacking. E.g. how do the authors account for possible (and reported) changes in ER morphology upon induction of the UPR? These could explain the observed changes in proteins localizing to different*

fractions without them being in different subcellular compartments. ER protein import is co-translational. How would the observed differences be achieved, in a SRP-independent manner?"

Response: We would like to clarify that morphological changes of ER would not be expected to affect localization assignment in the LOPIT method, because a spatial map is generated for each replicate and each condition, as opposed to applying a baseline map to predict localizations in treatment conditions. In other words, the localization classification is trained separately for each experiment. This gives a marker sedimentation profile for each experiment, which is visually confirmed to match within an experiment to ensure that the markers remain well separated in each treatment condition and replicate. In response to the Reviewer's comment, we have now included additional analysis to show clear separation of canonical compartment markers in both control and UPR experiments (new **Supplemental Figure S1**). Moreover, the new **Supplemental Figure S2** shows the sedimentation profiles in control and UPR conditions, respectively, which shows that ER localization is based on marker behavior and independent of specific shape of the sedimentation curve.

New Supplemental Figure S1. Separation of subcellular component markers in the spatial proteomics data. Scatter plots of the first (x-axis) and second (y-axis) principal components of ultracentrifugation fraction profiles are shown for each experimental condition: **A.** normal, **B.** thapsigargin, and **C.** tunicamycin treated AC16 cells, $n=3$ each. Each data point is a protein species. The colored data points correspond to marker proteins known to reside in each of 12 subcellular locations used to train the classification models, showing clear and consistent separation across the experimental conditions. Colors correspond to other spatial maps for AC16 cells throughout the manuscript.

New Supplemental Figure S2. Ultracentrifugation fraction distributions of cellular component markers. Replicate one of each experimental condition is shown. The line plots show the normalized abundance (y-axis) of marker proteins for each subcellular localization experiment across ultracentrifugation fractions (x-axis) as measured by the TMT channel intensities. The fractions correspond to the ultracentrifugation steps in Supplemental Table S3. Colors correspond to spatial maps for AC16 cells throughout the manuscript. Black lines show average trend line and standard deviation, showing consistent sedimentation profiles of subcellular localization in the normal, thapsigargin, and tunicamycin treated AC16 cells ($n=3$).

We have now extended the Results section of the main text to clarify this point.

Main text, Results, Line 202

“Three biological replicate SPLAT experiments were carried out for each condition (untreated, thapsigargin, and tunicamycin). [...] A spatial classification model is trained separately for each condition using a basket of canonical organelle markers which showed clear separation in PC1 and PC2 in each condition (**Supplemental Figure S1**). The ultracentrifugation profiles are largely consistent across treatments and replicates (**Supplemental Figure S2**).”

Furthermore, in response to the Reviewer’s comment, we have extended the Results section to note that there is no observed decrease in the proportion of localized proteins in thapsigargin-treated cells with prior annotations. In normal cells, 69.5% (1353 out of 1946) of proteins confidently assigned to a localization following localization and outlier probability filters match with UniProt Gene Ontology Cellular Component (CC) annotation terms of the same compartments, with 194 proteins assigned to the ER compartment matching UniProt annotations. In thapsigargin treated cells, 1312 out of 1833 proteins with confident allocations (71.%) match to UniProt annotation. Therefore, ER stress treatment did not lead to a substantial impact on localization assignment confidence and accuracy (See **Figure 2C**, **Supplemental Figure S5**, **Supplemental Data S2** and **Supplemental Data S4**). While the number of proteins assigned to ER compartment matching UniProt annotations is lower in thapsigargin cells (149 vs. 194), this can be attributable to the expansion of other compartments peroxisome/endosome co-sedimenting fraction that contain also ER vesicles and stress granule proteins (90 total assigned proteins in thapsigargin vs 49).

Figure 2C. Distribution of light (unlabeled) protein features in each of the 12 subcellular compartments (n=3); fill color represents whether the protein is also annotated to the same subcellular compartment in UniProt Gene Ontology Cellular Component terms.

New Supplemental Figure S5: Number of proteins localized (5% FDR) to one of 10 cellular compartments in the thapsigargin-treated (left) and tunicamycin-treated (right) AC16 cells. Proteins with unallocated localization are not included.

This point is now also mentioned in the main text:

Main Text, Results, Line 222 “The accuracy of the spatial classification is supported by the observation that 69.5% assigned proteins in normal AC16 cells contain matching cellular component annotation in Gene Ontology despite the current incompleteness of annotations (**Figure 2C**) and 71.6% of proteins match their localization annotation in thapsigargin-treated cells (**Supplemental Figure S5**).”

Reviewer 2 Comment 2: “There needs to be some form of validation that the method indeed identifies protein in the ER (or other compartments). How reliable is the fractionation method? Some digestion method (e.g. proteinase K) needs to be applied to distinguish proteins inside the organelle from proteins bound to the organelle.”

Response: We thank the Reviewer for the comment. With regard to the reliability of the fractionation method, we note that the spatial component of SPLAT is based directly on the ultracentrifugation fractionation scheme in the LOPIT-DC protocol which has now been applied to multiple studies (e.g., Geladaki et al., 2019; Shin et al., 2020). The LOPIT-DC approach is in turn built on well established protocols in spatial proteomics (Christoforou et al., 2016; Dunkley et al., 2004; Foster et al., 2006; Lundberg and Borner, 2019) which have been used extensively, including to validate immunofluorescence microscopy determination of subcellular localization of thousands of proteins (Thul et al., 2017). As the Reviewer will be aware, the principles and reliability of linking subcellular localization to ultracentrifugation sedimentation patterns can be traced back in turn to a long tradition of biochemistry experiments including de Duve’s discoveries of the lysosome and the peroxisome. Assurance that the fractionation protocols were properly carried out in our hands can be seen from the marker separation comparisons used for supervised learning as noted above (e.g., new **Supplemental Figure S2**), which allowed cross referencing to previous spatial maps and annotated localizations. We have now included new analysis that compares the predicted protein localizations to Gene Ontology Cellular Component (CC) terms in UniProt retrieved via uniprot.ws.

As stated in our response to **Reviewer 2 Comment 1** above, in baseline AC16 cells, 69.5% of predicted location matches UniProt GO_CC annotations, including 67.8% of ER localized proteins which are annotated with one of the following terms: endoplasmic reticulum [GO:0005783], endoplasmic reticulum membrane [GO:0005789], and endoplasmic reticulum lumen [GO:0005788]. This substantial overlap is found despite the incomplete nature of UniProt annotations (including especially non-canonical protein isoforms) and the non-exhaustive search terms we used, as well as using less than 10% of identified proteins as canonical markers, altogether suggesting the fractionation and prediction largely recapitulate known compartment assignments. The major exceptions are Golgi, which contains a number of proteins annotated to be in the ER in UniProt, which likely reflects the gradual connection between these two compartments in vivo, and the lysosomal compartment which contains a number of exosomal and endosomal labels due to unannotated compartments that are not trained with canonical markers in the spatial model (please see also our response to **Reviewer 1 Comment 11**).

We thank the Reviewer for the insightful suggestion for proteinase K treatment of organelles. However, as proteins bound to organelles are also important parts of the organelle-associated subproteome, and our goal is not to determine intra- vs. extra- organelle composition, we respectfully suggest that the experiments are beyond the scope of the current study.

Reviewer 2 Comment 3: “TMT experiments shot by MS2 have inherent issue with ratio compression. Consequently, the field has moved to MS3 measurements, including the cited paper (Dephoure 2012) describing SILAC plus TMT approaches. Why were the samples not shot with MS3? How was ratio compression controlled?”

Response: We thank the Reviewer for the comment. While we agree with the Reviewer that MS3-based TMT quantification can alleviate (though not eliminate) some ratio compression effect from co-isolated precursors, it also comes with the significant drawbacks of lowering profiling depth and thus DDA data completeness. These two analytical attributes are important for this project as we aim to find the spatial distributions of thousands of proteins and need to compare across independent injections. Ratio compression in MS2-based quantification can be effectively mitigated using methods, including the following that we have taken:

(1) Ratio compression can be alleviated by performing extensive offline two-dimensional peptide fractionation to reduce sample complexity. We have performed 20-fraction high-pH fractionation using an HPLC column, which reduces chromatographic crowding and hence the chance of co-isolated isobaric species from interfering with the measured TMT ratios.

(2) Isobaric peptide contamination can be further controlled by narrowing the quadrupole isolation window to increase precursor selectivity for fragmentation. In this study we used a narrow 0.7 m/z isolation, reduced from more commonly used 1.4 m/z for non TMT studies in comparable instruments. We are heartened to learn that Reviewer 1 who has extensive expertise in spatial proteomics methods also deems both measures 1 and 2 to be sensible in his comments (see **Reviewer 1 Comment 6** above).

(3) In addition, correction of channel crosstalk from isotopic impurities in isobaric tags is known to reduce the effect of ratio compression, by minimizing the unwanted contributions of neighboring channels in co-isolated isobaric species (Searle and Yergey, 2020). This correction was performed for all TMT data in this study using non-negative least square matrix factorization in the pytmt component of the pipeline (see also **Response to Reviewer 1 Comment 5** above).

(4) Another measure that is sometimes taken to further control for isobaric contamination is to calculate the degree of co-isolation as the fraction of precursor ion signals in the isolation window in the MS1 scan. This is calculated as the precursor ion fraction (PIF) value in MaxQuant or spectral purity in FragPipe, and can be used to filter out low-purity MS2 spectra with PIF/purity < 0.5 in some work flows. We did not deem this measure to be necessary here because the distribution of spectral purity in our experiment was already very high, with median PIF/purity ~ 0.93 (see new **Supplementary Figure S3**). Overall, we find that over 75% of the MS2 spectra had PIF/purity over 0.8 (i.e., estimated 80% isolation purity) and 57% had PIF/purity over 0.9, which corroborates the high measurement accuracy of the MS2-based TMT quantification.

To further explore this issue, we collaborated with Dr. Christopher Ebmeier (Director of Mass Spectrometry, University of Colorado Boulder) to perform extensive new experiments and acquire a new data set using MS3-based TMT quantification on a separate aliquot of an identical SPLAT sample in AC16 cells (untreated, replicate 2). The MS3-based quantification led to fewer quantified protein features than MS2-based quantification (6028 vs. 7453). To measure whether the subcellular fractions are better resolved in MS3-based TMT, we used the QSep metrics in the pRoloc package to calculate the normalized ratios of average between-cluster distance and the within-cluster distance, using the identical set of canonical protein markers for 10 subcellular fractions used in this study as the clusters. We find that the MS3-based TMT improved on the resolution of some clusters (e.g., separation of ribosome and cytosol), which is reflected by a moderate increase in inter-cluster vs. intra-cluster separation (non-diagonal median QSep 3.51 in MS2 vs.

3.98 in MS3) (New **Supplementary Figure S4A**). When we performed TAGM-MAP classification of proteins to the 10 clusters, we observed that the MS3-based TMT likewise led to overall improved appearance of cluster resolution in the first two principal components (New **Supplementary Figure S4B**). However From this experiment, we conclude that while MS3 may lead to better resolution of subcellular fractions in spatial proteomics studies, this improvement is not substantial for the purpose of subcellular location classification. MS2 quantification preserves effectively the same group relationships (New **Supplementary Figure S4B**), and has similar ability to distinguish subcellular fractions with high confidence.

This observation is consistent with the LOPIT-DC study by the Munro and Lilley groups, who also used MS2 quantification for their LOPIT-DC spatial proteomics study, and likewise performed MS2/MS3 comparisons and noted that although the SPS-MS3 resolution was somewhat better, MS2 showed effective separation of organelles with good resolution (Shin et al., 2020). In our view, while there is yet to be a consensus on the extent to which MS3 quantification may benefit spatial proteomics experiments, given the main usage of the TMT channel intensity information is pattern recognition rather than numerical ratio quantification, ratio compression per se may not present serious issues.

Lastly, in our view the notion that the field has “moved to MS3” for TMT quantification is not well supported by current evidence. Recent landmark proteomics studies, including MoTrPAC (Sanford et al., 2020) and NCI-CPTAC (Cao et al., 2021; Clark et al., 2019; Dou et al., 2020; Gillette et al., 2020; Krug et al., 2020; Wang et al., 2021), which are among the deepest and best controlled large-scale studies the proteomics field has produced, have been performed using MS2-based TMT quantification. The continued widespread acceptance of MS2-based TMT quantification is also evidenced by a comparison of the number of data sets on ProteomeXchange associated with publications using MS2 vs. MS3 TMT techniques. In 2015, the year when the Gygi lab’s multi-notch MS3 TMT data set was deposited to ProteomeXchange, the number of deposited data sets generated using Q-Exactive and Exploris instruments (which utilize MS2 TMT) was 5, compared to 4 using Tribrid Fusion/Lumos/Eclipse instruments (44%). Within the last two years (2021 to 2023 July), there were 284 data sets deposited using MS2-only instruments, and 295 using MS3-capable instruments. Hence both MS2 and MS3 TMT measurements continue to be widely employed in about 50/50 ratios (we note that this is likely an underestimate of MS2 usage since many Fusion Lumos data sets, e.g., CPTAC studies, use MS2 for TMT). As the latest Thermo Scientific flagship instrument (Orbitrap Astral) does not possess MS3 capacity, we expect MS2-based TMT quantification will continue to be widely employed.

New **Supplemental Figure S4**. Comparison of subcellular localization assignment in MS2 and MS3-based TMT measurements. An identical sample (replicate 2 of normal AC16 cells) was analyzed by MS2 and MS3 based quantification. **A**. The QSep index in the pRoloc package reflects the between-cluster distance vs. within-cluster distances of the 12 subcellular locations. MS3 achieved a modest increase in median QSep (3.98 vs. 3.51) suggesting the subcellular component clusters were slightly better separated. **B**. Spatial maps of proteins in MS2 vs. MS3 quantification. Colors correspond to other spatial maps in AC16 cells throughout the manuscript. Data point size reflects the confidence of TAGM-MAP classification.

In response to the Reviewer's Comment, we have now added the following to the revised manuscript:

Main text, Results, line 210 *"To minimize the potential ratio compression that can result from MS2-based TMT quantification, we employed extensive two-dimensional fractionation and narrow isolation window, and verified that identified MS2 spectra had high precursor ion purity (median purity 92–93%) (Supplemental Figure S3). We further performed a direct comparison of MS2 and MS3 based quantification on an identical sample (control replicate 2) (Supplemental Figure S4), which confirmed that MS2-based quantification produced acceptable spatial resolution, consistent with previous observations (Shin et al., 2020)."*

Finally, the newly acquired MS3 SILAC-TMT data has been made publicly available on ProteomeXchange under the accession number PXD046669. We thank the Reviewer for the opportunity to address this important point.

Reviewer 2 Comment 4: *“To my understanding (a clear description in the figures or legends is missing), the SILAC-TMT data are based on duplicates. Considering that the manuscript is entirely based on proteomics data, duplicates are not sufficient. P-values must not be calculated from duplicates. Whether p-values have been corrected, as required for such experiments, is not apparent from the manuscript.”*

Response: Thank you for the comment. We have now gone through the manuscript again to ensure all P values and sample numbers are listed where applicable. Likewise, other P values (e.g., ANOVA for functional assays) and multiple testing correction are clearly stated. All protein identification, protein turnover, and protein abundance comparison experiments are multiple-testing corrected using conventional Benjamini-Hochberg procedures. As stated below, the number of SPLAT/LOPIT experiments has been increased to three following the Reviewer’s comments. This has now been made apparent throughout the manuscript.

With regard to the Reviewer’s statement that P values “must not be calculated from duplicates”, we respectfully offer a different perspective. Setting aside that P values are not an inherent feature of Bayesian frameworks, we note that the Bayesian statistical method (BUNDLE) used in this study for spatial comparisons determines the posterior probability of protein localization using Gaussian processes of all proteins and is specifically compatible with any number of replicates including one or two (e.g., see discussion in (Crook et al., 2022)). Landmark spatial proteomics studies have found reliable results with one (Jean Beltran et al., 2016) or two (Thul et al., 2017) replicates per condition. As spatial proteomics involves large experiments that require many millions of cells and dozens of hours of bench work and data acquisition per replicate, we believe two replicates were a reasonable starting point. An estimated error calculation shows that the estimated FDR of translocation events are below 1% across all compared conditions, which is now highlighted in the text. On a related note, we note that P values of two-group comparisons are also routinely calculated with as few as two replicates in transcriptomics and proteomics experiments using common statistical packages that fit for mean-variance trends, such as limma, DESeq2, and edgeR.

Nevertheless, the Reviewer’s suggestion is well taken. Following the Reviewer’s comment, we have now performed extensive additional experiments to perform a third biological replicate experiment for normal, thapsigargin, and tunicamycin treated AC16 cells in the revised manuscript. All relevant spatial and temporal proteomics analyses have been redone to incorporate the third biological replicate and all relevant figures and panels (**Figure 2B–H, Figure 3A–C, Figure 3F–G, Figure 4A–G, Figure 5B–F, Supplemental Figures S1–S3, S5–S12**) have been remade. Overall, the third biological replicate shows a high degree of reproducibility with the first two and preserves previously presented findings while increasing confidence. The new set of SILAC-TMT data have been made publicly available on ProteomeXchange under the accession PXD046671.

In response to the Reviewer’s comments, the following changes have been made in the manuscript

Main Text, Results, Line 455 *“Three biological replicate SPLAT experiments were performed in tunicamycin-treated cells to resolve the temporal kinetics and subcellular localisation of proteins (**Supplemental Figures S1, S2C, S5B, S10**).”*

Reviewer 2 Comment 5: *“It is difficult to judge the validity of the SPLAT method without any experimental validation.”*

Response: The major goal of the study is to establish an in vitro method to encode spatial and temporal information within one mass spectrometry experiment. Both the spatial and temporal components are built on well established principles and prior methods. We have applied the approach to well established models of ER stress and show that it behaves as expected, for instance, nominating known ER stress proteins from temporal kinetics, while allowing new insights where both the spatial and temporal distributions of a protein change. All presented results are based on direct experimental data. However, the Reviewer's point is well taken and in the revision we have taken efforts to provide further considerations on the internal and external validity of the results through both new comparative analysis with expected changes and experimental validation, most prominently in the new/updated **Figure 2C, Figure 3D–3E, Figure 4A–4C, Figure 4H–4K, Supplemental Figures S5, S8, S11A–S11C, S14, and S16**). As this comment shares similarities with other comments from the Reviewer (e.g., **Reviewer 2 Comment 2** above and **Reviewer 2 Comment 6** below), please see our consolidated responses under the other comments.

Reviewer 2 Comment 6: *“Relocalization studies require more validation in order to validate the accuracy of the assay. E.g. do known UPR mediators show expected relocalization? Biochemical validation of mentioned translocator candidates that show prominent relocalization is required.”*

Response: We thank the Reviewer for the comment. In response, we have now expanded on the text to evaluate the accuracy of the assay, both in terms of whether known translocators show expected relocalization, as well as experimental validation.

Regarding expected relocalization of UPR mediators, we note that because of the sample complexity of bottom-up proteomics experiments, current methods are unable to monitor every protein in the proteome, especially proteins with low molecular weight or abundance such as many transcription factors. This challenge is compounded by our requirement of data completeness across three independent sets of mass spectrometry experiments, and the two-peptide filtering rule in place to ensure rigorous identification and quantification. The two best known early translocating UPR mediators, XBP1 and ATF6, are not within the coverage of the spatial proteomics data, so we are unable to comment on their spatial distributions. Moreover, as the Reviewer pointed out, the translocation of XBP1 and ATF6 are transient events that happen early in initial ER stress response signaling. Here, we designed the experimental time point with the goal of capturing the sequelae of unfolded protein response. Therefore, it is not certain whether they would be expected to show relocalization even if they were within the coverage of the experiment.

On the other hand, the spatial proteomics data in the study clearly recapitulate known hallmark translocators in stress response, which we now highlight in the text in response to the Reviewer's comment. Considering only the 330 differential localization candidates in the thapsigargin treatment after stringent filtering criteria, we identified multiple established translocation events. First, PERK-induced shift from EIF2-dependent to EIF3-dependent translation initiation is a hallmark of translational reprogramming under integrated stress response (Guan et al., 2017; Lamper et al., 2020). In the spatial proteomics data, we observe a clear shift of the constitutive EIF3 subunits EIF3A, EIF3H and EIF3L relocalized towards the ribosome compartment (BUNDLE translocation probability > 0.999), consistent with EIF3 engagement with the ribosome and partition from ribosome-free fractions as previously reported (Guan et al., 2017). Secondly, caveolae are known to migrate toward the mitochondria as a protective response to various cell stresses including in the heart (Fridolfsson et al., 2012; Volonte et al., 2016). In the spatial proteomics data, we observe a shift of caveolin 1 (CAV1) and caveolae-associated protein 1 (CAVIN1) toward the mitochondrial compartment (BUNDLE translocation probability > 0.999). Thirdly, other translocation events highlighted in the text show relocalization that take place at expected compartments (e.g., relocalization of PA28 and PA200 toward the proteasome) even if their role in cardiac cell stress has not been established. Taken

together, we believe these expected translocation events lend strong evidence to the external validity of the spatial proteomics data presented in the study. These results are now presented in the new **Supplemental Figure S8**.

Supplemental Figure S8. Known translocation events under cellular stress captured in the spatial proteomics data. **A.** Caveolae migration toward the mitochondrion during cellular stress is reflected in the differential localization of CAV1 and CAVIN1 (BANDLE probability > 0.999) toward the mitochondrion compartment. **B.** A switch to EIF3-dependent translation initiation, a hallmark of prolonged ER stress and integrated stress response, is evident in the differential

localization toward the ribosome compartment of three independent EIF3 subunits EIF3A (BUNDLE probability: 0.996), EIF3A (BUNDLE probability: 0.996), EIF3H (EIF3A (BUNDLE probability: >0.999), and EIF3L (EIF3A (BUNDLE probability: 0.992). The nucleus localization probability of these proteins in normal AC16 cells is accompanied by a high outlier probability (**Supplemental Data S2**) and may reflect partial ribosome localization. Left: spatial maps of PC1 vs PC2; colors correspond to other spatial maps for AC16 cells throughout the manuscript. Open circles: location of light and heavy protein in each condition. Only the map of one of three replicates is shown for simplicity. Right: ultracentrifugation profiles showing relative abundance (y-axis) across fraction (x-axis). Numbers inside the fraction profile correspond to BUNDLE localization probability.

The main text has now been updated to refer to this figure.

Main Text, Results, Line 294 “The differential localization of these 330 proteins recapitulate previously established relocalization events in cellular stress response, capturing the migration of caveolae toward the mitochondrion under cellular stress (Fridolfsson et al., 2012) (**Supplemental Figure S8A**), and the engagement of EIF3 to ribosomes in EIF3-dependent translation initiation in integrated stress response (Guan et al., 2017) (**Supplemental Figure S8B**), thus supporting the confidence of the spatial translocation assignment.”

Secondly, the technical reliability of the spatial proteomics data can also be seen from the highly concordant profiles of light and heavy proteins, which are independently measured in a label-agnostic manner in the mass spectrometer. We now present more clearly the significant difference in spatial distribution dissimilarity between light-heavy pair of the same protein species compared to between two different protein species (Mann-Whitney $P < 2.2e-16$) (**Figure 4A–B, Supplemental Figure S11A–C, Supplemental Figure S14B–C**). In the majority of the cases except for the special partition cases we highlighted (e.g., EGFR) we find the expected translocation of the new proteins when the old protein pool relocalizes, which speaks to the internal validity of the spatial proteomics data.

Figure 4A. Histogram showing the similarity in light and heavy proteins in normalized fraction abundance profiles in (left) normal and (right) thapsigargin-treated AC16 cells. X-axis: the spatial distribution distance of two proteins is measured as the average euclidean distance of all TMT channel relative abundance in the ultracentrifugation fraction profiles across 3 replicates; y-axis: count. Blue: distance for 1,614 quantified light-heavy protein pairs (e.g., unlabeled EGFR, heavy SILAC-labeled EGFR). Grey: distribution of each corresponding light protein with another random light protein. P value: Mann-Whitney test. **B.** Proportion of heavy-light protein pairs with confidently assigned localization that are assigned to the same location (purple) in normal (left; 93%) and thapsigargin-treated (right; 89%) cells.

Lastly, regarding experimental validation, in the study we have performed new sets of spatial proteomics experiments to increase the number of replicates for normal, thapsigargin, and tunicamycin to three, and show that the major results are reproduced. We have also pursued orthogonal observations using immunostaining of the preferential localization of EGFR to the ER in the thapsigargin comparison. In response in part to the Reviewers’ input, we have included additional orthogonal experiments in the revised manuscript to corroborate the observed relocalization events, including the differential localization of SLC3A2, which shows localization away from lysosome upon thapsigargin treatment (see response to **Reviewer 3 Comment**

4). We believe these experimental validations serve to corroborate the credibility of our findings on some newly discovered protein translocation upon stressors.

Reviewer 2 Comment 7: *“Half-lives and turnover rates are calculated based on one time point (in duplicate) that is much earlier than the calculated half-lives. This is not sufficient. Proper time-courses covering beyond 10 days are required to calculate half-lives. This is particularly important since the authors claim that the single time points used are based on protein half-lives, which consequently need to be accurate.”*

Response: We thank the Reviewer for the opportunity to clarify an important point. The kinetic model depends on only one variable (k) in the optimization procedure, therefore data from a single time point is fully sufficient to define the kinetic curve without the use of a time course, provided that the sampling time point is within the informative region of the turnover function. In practice, the kinetic model is heteroscedastic and a model sensitivity analysis (e.g., as in our prior studies using analytical solutions to dk/dA under a custom kinetic model (Lam et al., 2014) or two-compartment models (Hammond et al., 2022)) is seldom pursued. In our experience, it is typical for studies in the protein turnover literature to report turnover rates where the sampling time points range have reached 0.25 to 4 half-lives. In our case, in the AC16 experiment sampled at 16 hours, that would translate to being able to reliably measure peptides with half-lives between ~4 and 96 hours, where the proportions of the heavy SILAC peak to the heavy plus light peptides would range from ~11% to 94%. These ratios are well within the dynamic range of MS1 quantification for SILAC experiments, and covered virtually all reported protein turnover rates in this study. A more detailed discussion on half-life range based on sampling time points can be found in our recent publication (Hammond et al., 2022).

Although a time course design can aid in ensuring that the sampling time points fall within the informative regions of kinetic curves and hence support a wider range of measurable half-life, there are also many instances where a time course design is not necessary or practical. For instance, both we (Lam et al., 2014) and others (Naylor et al., 2022) have considered the accuracy of single-time-point measurements in cases where opportunities for biopsies are limited following stable isotope enrichment, such as when biopsy collection opportunities are limited. In the case of this study, although the SPLAT method can in principle be applied to any arbitrary time course, additional time points were not pursued because we already had reasonable expectation of the range of half-lives in the studied cell type and aimed to synchronize the experimental end point with the perturbation duration.

To further illustrate how protein turnover can be reliably measured from single time points taken earlier than the protein half-life, we downsampled a full time series in one of our prior publications (Hammond et al., 2022), which performed stable isotope labeling in C57BL/6J mice to measure protein turnover rates in the heart, liver, kidney, and skeletal muscle. In the original study design, samples were taken at 12 time points following day 0, 1, 2, 3, 6, 7, 9, 13, 16, 21, 24, 31 of labeling, with the measured median protein half-life being approximately 11 days as is typical in the adult mouse heart under normal feeding conditions. **Reviewer Figure R3** shows the measured \log_{10} protein turnover rates and standard deviations between taking cardiac data from only a single time point on day 6 (left) (thus much earlier than the half-life of most proteins). The scatterplots show comparable protein turnover rates ($r: 0.94$) with the full time course among common proteins, quantifiable over 2 orders of magnitude, and the single-time-point sampling at day 6 returned reliable turnover rate values to at least $\log_{10}(k)$ of -1.5 (i.e., half life of 21.9 days or over 3x the sampling time point). The correlation with the full time course is also comparable to a single time point chosen around the median half-life (day 13) on the right. Hence a judiciously placed single time point is sufficient for protein half-life calculation, as is consistent with reports in the protein turnover literature.

Reviewer Figure R3. Single time points are sufficient for protein turnover measurements. Each data point is a single protein measured by mass spectrometry in the mouse heart. x-axis: the \log_{10} turnover rate (k) of the protein, calculated as the harmonic mean of constituent peptides, in the full time course experiment. y-axis: the \log_{10} turnover rate (k) of a protein, calculated as the harmonic mean of constituent peptides, from data of only a single time point (day 6 on the left, day 13 on the right). Error bars: standard deviation in the single time point sampling and the full time course. Proteins quantified with 3 peptides in the single time point sampling are included.

Reviewer 2 Comment 8: “ER stress is a transient response that typically shuts itself off via feedback loops since prolonged activation induces apoptosis. Why were such a long treatment time chosen (16h)? Does it lead to cells death? Are UPR markers of the integrated stress response (i.e. eIF2alpha phosphorylation, detectable ATF4 and CHOP protein levels) still detectable by Western blot? This issue is even more pronounced for the conditions of inhibiting the proteasome for 48h. Is there any induction of cell death? Considering the surprising lack of changes in protein degradation, experiments are required to show that the proteasomal function is actually inhibited over the time span of 48h. Is there a concurrent induction of autophagy as the other major protein degradation pathway in cells?”

Response: We thank the Reviewer for the comment. As the Reviewer is aware, ER stress is distinct from the transient ER stress response pathways. Our goal here is to model proteostatic stress on protein spatiotemporal dynamics, using established models with dosages and time points well within the literature range. For thapsigargin, a treatment of 12 to 18 hours in the $\sim 1 \mu\text{M}$ range was used with minimal loss in cell viability (Ghosh et al., 2023; Stoner et al., 2020); whereas treatment of 24 hours or beyond have been observed to induce apoptosis (Chen et al., 2019; Stoner et al., 2020). Specifically, Stoner et al. used $0.5 \mu\text{M}$ thapsigargin for 18 hours in human AC16 cardiac cells to induce ER stress and observed increased XBP1 and ATG6 expression; whereas Ghosh et al. treated AC16 cells for $0.5 \mu\text{M}$ thapsigargin for 16 hours in human AC16 cardiac cells to induce UPR and show upregulated PERK and XBP1 (Ghosh et al., 2023). Chen et al. used $1 \mu\text{M}$ thapsigargin in rat H9c2 cardiac myotubes and primary neonatal rat cardiomyocytes for 12 hours to induce ER stress as evident by BiP induction (Chen et al., 2019). On the other hand, $1 \mu\text{M}$ thapsigargin treatment for 24 hours in induced apoptosis in two studies (Chen et al., 2019; Stoner et al., 2020).

For tunicamycin, a range of timepoints from 4 to 18 hours have been used for cardiac and cardiac-like cells in the literature, and we have opted to stay within this range while maintaining a 16 hour treatment window to allow drug treatment and heavy SILAC amino acids to be applied concurrently, and for ease of

comparison with thapsigargin. For instance, Palomar et al used a higher dose of 5 $\mu\text{g}/\text{mL}$ for 4 hours in human AC16 cardiomyocytes to induce ER stress as evident by BiP and ATF3 (Palomer et al., 2014). Toro et al. used 2 μM ($\sim 1.6 \mu\text{g}/\text{mL}$) for 6 hours and 18 hours in human AC16 cardiomyocytes to induce ER stress as evident by BiP (GRP78) increase (Toro et al., 2022). Siltanen et al. used up to 0.3 $\mu\text{g}/\text{mL}$ tunicamycin for 24 hours in primary fetal rat cardiac cells to activate protective UPR (Siltanen et al., 2016). On the other hand, Liu et al. used 1 $\mu\text{g}/\text{mL}$ tunicamycin for 36 hours to induce apoptosis in cultured neonatal rat cardiomyocytes with about 30% drop in viability (Liu et al., 2012).

Based on the Reviewer's comments, we have now expanded the main text to show known mammalian ER stress induced markers in the literature (Glembotski, 2007). Following thapsigargin treatment for 16 hours, we observed robust induction of multiple established ER stress induced genes, including PDIA4, HSPA5/GRP78/BIP, GRP94/HSP90B1/TRA1/ DNAJB11, HERPUD1, ASNS, and PTX3. Likewise, following tunicamycin treatment for 16 hours, we observed a robust induction of a multitude of ER stress inducible markers including PDIA4, CALR, CANX, HSPA5/GRP78/BIP, HSPA13/STCH, GRP94/HSP90B1/TRA1, DNAJB11, DNAJC3, HERPUD1, and ASNS. While one might not necessarily expect every ER stress marker to be induced due to the incomplete profiling depth and potential cell type differences, the preponderance of evidence points strongly to robust UPR induction in a well established model. Relevant to the discussion on treatment time point, a number of markers were more elevated following 16 hours than 8 hours. We did not observe loss of cell viability under 16 hours treatment, which is consistent with the literature cited above. The marker expression data are now presented in the main text as the new **Figure 2A** and new **Figure 5A**.

New Figure 2A: Induction of known ER stress response markers following 16 hours of thapsigargin treatment in AC16 cells. *: limma adjusted P (multiple-testing corrected) < 0.01 ; **: limma adjusted $P < 0.001$, ***: limma adjusted $P < 0.0001$; $n=6$ for untreated; $n=3$ for thapsigargin 8 hours; $n=3$ for thapsigargin 16 hours.

New Figure Panel 5A: Induction of known ER stress response markers following 16 hours of tunicamycin treatment in AC16 cells. *: limma adjusted P (multiple-testing corrected) < 0.01 ; **: limma adjusted $P < 0.001$, ***: limma adjusted $P < 0.0001$; $n=6$ for untreated; $n=3$ for tunicamycin 8 hours; $n=3$ for tunicamycin 16 hours.

For proteasome inhibition of 48 hours, this dosage and duration was chosen based on prior work in the literature for a clinically relevant dose that models cardiotoxicity in human iPSC-derived cardiomyocytes (Forghani et al., 2021). To explore this issue further, we have performed additional experiments on iPSC-CM viability and phenotypes under 0 to 5 μM carfilzomib for 24 and 48 hours (new figure panels **Figure 6B-6H**). Under the chosen dose (0.5 μg for 48 hours), iPSC-CMs showed sarcomeric disarray consistent with prior observations on the cardiotoxic effects of carfilzomib (**Figure 6B**) but maintained viability (82%) (**Figure 6C**) while showing significant decreases in oxygen consumption (**Figure 6D**) basal respiration (**Figure 6E**), maximal respiration (**Figure 6F**), and ATP production (**Figure 6H**), whereas higher doses are accompanied with drops in viability at 48 hours. These data are consistent with the prior literature that show the effect of modeled cardiotoxicity in iPSC-CMs and corroborate the chosen dose.

Figure 6C-H. Cell viability (%), normalized Seahorse oxygen consumption rate (OCR; pmol/min), basal respiration, maximal respiration, proton leak, and ATP production upon 0 – 5 μM carfilzomib for 24 or 48 hrs; \therefore adjusted $P < 0.1$; *: adjusted $P < 0.05$; **: adjusted $P < 0.01$, ANOVA with Tukey's HSD post-hoc at 95% confidence level; $n=5$. Error bars: s.d. for bar charts in panels C, E, F, G, H; s.e.m. for the OCR graph in panel D. Colors in panel D: dosage, same as panel C. O: Oligomycin; AA/R: Antimycin A/Rotenone. **I.** Spatial map with 13 assigned subcellular localizations in iPSC-CMs at the baseline (top) and upon 0.5 μM carfilzomib treatment ($n=2$). **J.** Histogram of \log_{10} protein turnover rates (k), with median half-life of 97.4 hours and 100.0 hours in normal and carfilzomib-treated iPSC-CM.

Further in response to the Reviewer's comment, we have now performed proteasome activity assays with iPSC-CMs treated with 0.5 μM carfilzomib for 48 hours. Briefly, we used the Proteasome Glo Chymotrypsin-Like Cell-Based kit to measure the chymotrypsin-like activity of proteasomes, which are specifically targeted by carfilzomib. We find that at 48 hours after carfilzomib treatment, proteasome activity remains suppressed in iPSC-CMs, to about 25% of normal levels (new **Figure 6K**, see below). It is also clear that some residual activity persists when compared to the in vitro MG-132 inhibition negative control. We note also that because the substrate of the assay is an unstructured tetrapeptide linked to a luciferase substrate, the proteasome assay only measures the catalytic step but not the commitment step of protein degradation, the latter of which is rate limiting in living cells (Claydon and Beynon, 2012). We observed a suggestive induction of autophagy at $P = 0.053$ in iPSC-CMs treated with 0.5 μM carfilzomib for 48 hours, which may compensate partially for protein degradation (new **Figure 6L**, see below). Hence, we show that although carfilzomib continues to inhibit PSMB5 catalytic activity to a significant degree at 48 hours, the temporal proteomics data paint a more nuanced picture where global per-protein average half-life is similar but particular proteins are affected. Finally, we would like to emphasize that we have used an established and clinically relevant dose that has led to verifiable cellular pathologies, and the degree of proteasome activity per se may not be the driving cause of cardiotoxicity.

Figure 6K. Proteasome activity in iPSC-CMs treated with 0 (Ctrl) vs. 0.5 μM carfilzomib (Cfbz) for 48 hrs. P value: t -test; $n = 3$. **L.** Autophagy assay for iPSC-CMs treated with 0 (Ctrl) vs. 0.5 μM carfilzomib (Cfbz) for 48 hrs, and positive control (Pos); data were normalized to DAPI and normal cells. P value: t -test; $n = 10$.

Finally, the main text has been updated to reflect these changes.

Main Text, Results, Line 548 “To verify toxicity modeling, we measured iPSC-CM viability and phenotypes under 0 to 5 μM carfilzomib for 24 and 48 hours. Under the chosen treatment (0.5 μg for 48 hours), iPSC-CMs showed sarcomeric disarray consistent with prior observations on the cardiotoxic effects of carfilzomib (**Figure 6B**) but maintained viability (82%) (**Figure 6C**), while showing significant decreases in oxygen consumption (**Figure 6D**), basal respiration (**Figure 6E**), and maximal respiration (**Figure 6F**), whereas higher doses are accompanied with drops in viability at 48 hours and an increase in proton leak (**Figure 6G**). ATP production at the 0.5 μg dose was significantly lower than untreated cells at both 24 and 48 hours (**Figure 6H**). Therefore cardiotoxicity due to carfilzomib can be recapitulated in a human iPSC-CM model, consistent with prior work in the literature.”

Main Text, Results, Line 602 “At 48 hours of carfilzomib treatment in iPSC-CMs, proteasome chymotrypsin-like activities are partially suppressed but significant partial activities are also observable (**Figure 6K**); whereas other proteolysis mechanisms may also compensate for proteasome inhibition, including a suggestive increase in autophagy ($P: 0.053$) (**Figure 6L**).”

Reviewer 2 Comment 9: “UPR activation leads to an extensive shutdown in protein translation via the

integrated stress response. How do the observed changes in translation compare to published observation and how is this difference accounted for in the analyses?”

Response: We observed a decrease in global isotope incorporation consistent with decreased translation, as we described in the manuscript. The spatial profiles of newly synthesized proteins are resolved separately from pre-existing proteins in the method.

Main Text, Results, Line 232 *“This slowdown is consistent with the extensive shutdown in protein translation under integrated stress response, which decreases the rate of SILAC incorporation into proteins.”*

Minor and specific comments

Reviewer 2 Comment 10: *“Figure legends have to explain abbreviations and give number of replicates.”*

Response: We thank the Reviewer for the reminder. We have now gone through the figure legends again to make sure we explain abbreviations and list the number of replicates.

Reviewer 2 Comment 11: *“Supplemental tables should have a cover page with a title, figure cross-reference and explanation of the columns.”*

Response: We fully agree with the Reviewer. We have now moved the supplemental material to a Supplemental Information document with a cover page and title as suggested. All Supplemental Figures, Tables, and Data should now be referenced in the main text. We have reorganized the Supplemental Tables to include relevant information and into 10 Supplemental Data tables as listed in the Supplemental Information:

- Supplemental Data S1 - Abundance changes in ER stress vs. Normal AC16 cells
- Supplemental Data S2 - Protein localization and assignment in Normal AC16 cell
- Supplemental Data S3 - Turnover rate ratios in Thapsigargin vs. Normal AC16 cells
- Supplemental Data S4 - Protein localization and assignment in Thapsigargin AC16
- Supplemental Data S5 - Turnover rate ratios in Tunicamycin vs. Normal AC16 cells
- Supplemental Data S6 - Protein localization and assignment in Tunicamycin AC16
- Supplemental Data S7 - Protein localization and assignment in Normal iPSC-CM
- Supplemental Data S8 - Turnover rate ratios in carfilzomib vs. Normal iPSC-CMs
- Supplemental Data S9 - Protein localization and assignment in Carfilzomib iPSC-CMs
- Supplemental Data S10 - List of canonical compartment markers in spatial experiments

Each Supplemental Data is now formatted as an Excel spreadsheet, and the columns are now clearly explained in a Column Description sheet within each Excel file. We thank the Reviewer for the helpful guidance.

Reviewer 2 Comment 12: *“Labeling in figures should be consistent (e.g. Fig. 2A, B).”*

Response: We thank the Reviewer for the reminder. We have made every effort to go through all the figures to make sure the letter labels are consistent, and the figures are referenced consistently in the main text. Since the changes to the figures are substantial due to new experiments and analyses, they are not reproduced here. Please refer to the main text and supplemental information for details.

Reviewer 2 Comment 13: *“Panels in figures do not appear in the order as they are discussed in the results section, please reorder.”*

Response: We thank the Reviewer for the reminder. The figure panels should now be referenced in order in the Results section of the main text.

Reviewer 2 Comment 14: “137: add reference for LOPIT approach, revise sentence”

Response: We have now added the reference to the LOPIT-DC paper and revised the sentence for clarity.

Main Text, Results, Line 127 *“In particular, the LOPIT-DC (Localisation of Organelle Proteins by Isotope Tagging after Differential ultraCentrifugation) method (Geladaki et al., 2019) allows the advantageous use of sequential ultracentrifugation to enrich different subcellular fractions from the same samples...”*

Reviewer 2 Comment 15: “187-192: some fold changes do not match the median values shown in Figure S1”

Response: We thank the Reviewer for the comment. We have examined the figure in question (now moved to **Figure 2A** and **Figure 5A**) for any discrepancy. Some of the ratio differences might have come from unclear labeling between 8 hours and 16 hours of thapsigargin and tunicamycin treatment. We now refer the readers to Supplemental Data S1 for numerical fold-changes for accuracy.

Reviewer 2 Comment 16: “235-236: unclear what the authors are trying to say”

Response: We apologize for the confusion. The intended meaning of the text was that there is evidence of regulated protein kinetics changes, vs. uniform slowdown due to reduced proliferation. We have now reworded for simplicity.

Main Text, Results, Line 235 *“Notwithstanding the overall slowdown, we also observed a wide range of protein turnover rates in both the untreated and thapsigargin treated conditions that differ by the assigned subcellular compartment (**Supplemental Figure S6**).”*

Reviewer 2 Comment 17: “256-259: include references for suppression of protein synthesis through ribosome remodeling.”

Response: We have updated the main text to include two references.

Main Text, Results, Line 232 *“This slowdown is consistent with the extensive shutdown in protein translation due to ribosome remodeling under integrated stress response (Bresson et al., 2020; Pakos-Zebrucka et al., 2016), shown here by the decreased rate of SILAC incorporation into proteins.”*

Reviewer 2 Comment 18: “268-271: where do the numbers (FC, p-values) come from, please make accessible in a table. Proteins mentioned here are not shown in the figure.”

Response: We apologize for the omission. We now refer to **Supplemental Data S3** for the fold-change and adjusted P values.

Main Text, Results, Line 241 *“On an individual protein level, out of the 2516 proteins measured, 1542 showed significant changes in temporal kinetics (Mann-Whitney test, FDR adjusted P value < 0.1), but the vast majority of these proteins show decreases in turnover as expected, with only 12 proteins showing significant increased temporal kinetics. Among these are the induced ER stress markers BiP/HSPA5, HSP90B1, and PDIA4 (**Figure 2F; Supplemental Data S3**) but also other ER and Golgi proteins that may be involved in*

stress response (**Figure 2G**). SDF2L1 (stromal-cell derived factor 2 like 1) is recently described to form a complex with the ER chaperone DNAJB11 to retain it in the ER (Hanafusa et al., 2019). In control cells, we found that SDF2L1 has a basal turnover rate of 0.027/hr. Upon thapsigargin treatment, its turnover rate increased to 0.048 /hr (adjusted P : 0.07). DNAJB11 also experienced accelerated kinetics (1.28-fold in thapsigargin, adjusted P 0.029) hence both proteins may be preferentially synthesized during UPR.”

In addition, where practical, we have now made every effort to match the proteins discussed throughout the main text to a figure panel, for instance, the proteins above are shown in **Figure 2F–2G**.

Figure 2F. Example of best fit curves in the first-order kinetic model at the protein level between normal (gray), and thapsigargin treated (red) AC16 cells showing four known ER stress markers with elevated turnover (HSPA5, RCN3, HSP90B1, and PDIA4); Asterisks: FDR adjusted $P < 0.1$. Because the sampling time point is known, the measured relative isotope abundance of a peptide (prior to reaching the asymptote) is sufficient to define the kinetic curve and the parameter of interest (k). **G.** Turnover rate ratio (thapsigargin vs. normal) of the top proteins with elevated temporal kinetics in UPR within the ER (blue) and Golgi (green); \therefore Mann-Whitney test FDR adjusted P value < 0.1 ; *: < 0.05 ; ** < 0.01 ; red dashed line: 1:1 ratio; bars: standard error.

Reviewer 2 Comment 19: 280-285: discussed data is not shown

Response: We apologize for the omission. We have made every effort to match the proteins discussed throughout the main text to a figure panel. For the sentences in question, the results of the tunicamycin comparison has been moved to its own section in the main text under the section “Spatiotemporal proteomics highlights similarities and differences of ER stress induction protocols” in response to the Reviewer’s comment that the tunicamycin results are not sufficiently highlighted.

Main Text, Results, Line 471 “Compared to thapsigargin treatment however, no significant enrichment of glycosylation and vesicle transport related terms were found in tunicamycin. Inspection of individual protein kinetics changes likewise revealed both similar induction of the ER stress response markers HSPA5, HSP90B1 and PDIA4 as in thapsigargin treatment, but other stress response genes PDIA3 and NIBAN1 are not induced in thapsigargin (**Figure 5E**). On the other hand, RCN3 (reticulocalbin 3) is an ER lumen calcium binding protein that regulates collagen production (Martínez-Martínez et al., 2017) and shows increased temporal kinetics in thapsigargin (**Figure 2F**) but not in tunicamycin (ratio 0.76 over normal; **Supplemental Data S5**), altogether reflecting potential differences in stress response modality to a different ER stress inducer.”

The discussed data are now shown in **Figure 5E** and **Supplemental Data S5**.

Figure 5E. Example of best fit curves in the first-order kinetic model at the protein level between normal (gray), tunicamycin (blue) and thapsigargin (red) treated AC16 cells showing known ER stress markers with elevated turnover in both ER stress inducers (HSPA5, HSP90B1, and PDIA4) as well as stress response proteins with elevated turnover only in tunicamycin (PDIA3, DNAJB11, NIBAN1).

Reviewer 2 Comment 20: “315-326: discussed data is not shown in a figure or provided as table”

Response: We apologize for the poor presentation. The data in question (whether light and heavy proteins have similar localization patterns or are localized to identical cellular locales) is now presented as a figure in the new **Figure 4A–4B**.

Figure 4A. Histogram showing the similarity in light and heavy proteins in normalized fraction abundance profiles in (left) normal and (right) thapsigargin-treated AC16 cells. X-axis: the spatial distribution distance of two proteins is measured as the average euclidean distance of all TMT channel relative abundance in the ultracentrifugation fraction profiles across 3 replicates; y-axis: count. Blue: distance for 1,614 quantified light-heavy protein pairs (e.g., unlabeled EGFR, heavy SILAC-labeled EGFR). Grey: distribution of each corresponding light protein with another random light protein. P value: Mann-Whitney test. **B.** Proportion of heavy-light protein pairs with confidently assigned localization that are assigned to the same location (purple) in normal (left; 93%) and thapsigargin-treated (right; 89%) cells.

The data for the tunicamycin treated AC16 cells is now shown in **Supplemental Figure S11A–A11B**

Supplemental Figure S11A. Histogram showing the similarity in light and heavy proteins in normalized spatial distribution distances in (tunicamycin-treated AC16 cells. X-axis: euclidean distance of fraction profiles across 3 replicates; y-axis: count. Blue: distance for quantified light-heavy protein pairs. Grey: distribution of each corresponding light protein with another random light protein. P value: Mann-Whitney test. **B.** Proportion of

heavy-light protein pairs with confidently assigned localization that are assigned to the same location (purple) in normal (left; 93.0%) and tunicamycin-treated (right; 85.2%) cells.

The data for iPSC-CM is now presented in **Supplemental Figure S14B–S14C**.

Supplemental Figure S14B. Histogram showing the similarity in light and heavy proteins in normalized fraction abundance profiles in (left) normal and (right) carfilzomib-treated iPSC-CMs. X-axis: euclidean distance of fraction profiles across 2 replicates; y-axis: count. Blue: euclidean distance for quantified light-heavy protein pairs. Grey: distance of each corresponding light protein with a random sampled light protein. P value: Mann-Whitney test. **C.** In baseline and stressed iPSC-CMs, 87.6% and 80.2% of light and heavy protein pairs are assigned to the same subcellular localization.

Reviewer 2 Comment 21: “355-356: tunicamycin data are not shown. Either show data or remove from the results section.”

Response: We apologize for the omission. The results of the tunicamycin comparison has been moved to its own section in the main text under the section “Spatiotemporal proteomics highlights similarities and differences of ER stress induction protocols” in response to the Reviewer’s comment. In addition, we have made every effort in the revised manuscript to include in the figure panels all proteins discussed throughout the main text. For the sentence in question, namely the migration of proteins toward the peroxisome/endosome co-sedimenting compartment, the accompanying figures are now presented in **Supplemental Figure S12**.

Supplemental Figure S12. Examples of proteins translocating toward the peroxisome-endosome cosidementing compartment in tunicamycin treatment. (Left) Alluvial plot of significant protein translocation ($P_r > 0.95$) from the ER, Golgi apparatus (GA), PM, and lysosome toward the peroxisome/endosome. Colors correspond to spatial maps for AC16 cells throughout the manuscript. (Right) Ultracentrifugation fraction profile of DNAJB11, DNAJC3, DNAJC10, PDIA6, EMC4, EMC8, VAPA, and VAPB showing the localization of the proteins to the ER and to the peroxisome/endosome fraction in normal and thapsigargin-treated AC16 cells, respectively. X-axis: fraction 1 to 10 of ultracentrifugation. Y-axis: relative channel abundance. Bold lines represent the protein of interest; light lines represent ultracentrifugation profiles of all proteins classified to a respective localization. Colors correspond to subcellular localization in panel B and for all AC16 data throughout the manuscript. Numbers in the box represent BANDLE localization probability to the compartment. RNA Granule Score: score from RNA Granule Database (<https://rnagranuledb.lunenfeld.ca/>). A score of 7 or above is considered a known stress granule protein. Phi: predicted phase separation participation. Circle denotes a prediction of True within the database, X denotes a prediction of False. RBP: Annotated RNA binding protein on the RNA Granule Database. One circle denotes known RNA binding proteins (RBP) in at least one data set; two circles denote known RBP in multiple datasets. Dashes indicate proteins not found within the RNA Granule Database.

In addition, the main text has been updated to expand on the tunicamycin result as suggested by the Reviewers at multiple comments, and to reference **Supplemental Figure S12**.

Main Text, Results, line 502 “Notably, although tunicamycin also induced the translocation of proteins toward the peroxisome/endosome fraction, different proteins are involved, including the stress response proteins DNAJB11, DNAJC3, DNAJC10, and PDIA6 as well as other proteins EMC4, EMC8, VAPA, and VAPB (**Supplemental Figure S12**) which further outlines different modalities of cellular response toward two different ER stress inducers.”

Reviewer 2 Comment 22: “356: refer to Fig. 4A”

Response: Thank you for the suggestion. In response to the Reviewer’s comments, the discussion in question has been reorganized to refer to the peroxisome/endosome migration of proteins upon tunicamycin, presented individually in **Supplemental Figure S12** (see our response to Reviewer 2 Comment 21 above).

Reviewer 2 Comment 23: “360: typo ‘sediment”

Response: Thank you. This is now fixed and rewritten.

Main Text, Results, Line 315 “In mammalian cells, ER and peroxisomes are spatially adjacent; the peroxisome associated fractions sediment prominently at 5000–9000 × g (F3 and F4) in the LOPIT-DC protocol (**Supplemental Figure S2**), marked by canonical peroxisome markers PEX14 and ACOX1 (**Supplemental Data S2**).”

Reviewer 2 Comment 24: “367-368: add references for ‘previously known translocators”

Response: Thank you. We have reorganized the Results section for clarity in response to the Reviewer’s comments. Expected translocation events are now presented in the main text and referenced.

Main Text, Results, Line 294 “The differential localization of these 330 proteins recapitulate previously established relocalization events in cellular stress response, capturing the migration of caveolae toward the mitochondrion under cellular stress (Fridolfsson et al., 2012) (**Supplemental Figure S8A**), and the engagement of EIF3 to ribosomes in EIF3-dependent translation initiation in integrated stress response (Guan et al., 2017) (**Supplemental Figure S8B**), thus supporting the confidence of the spatial translocation assignment.”

Reviewer 2 Comment 25: “367-371: proteins mentioned here are not the ones highlighted in the figure.”

Response: We apologize for the poor presentation. In the revision, we have endeavored to include an individual figure presentation for every protein mentioned, in addition to pointing to the corresponding Supplemental Data table for numerical values. For the discussion in question (ER vesicle related proteins migrating toward the peroxisome/endosome co-sedimenting compartment), we now present their individual fraction profiles and localization probability in the new **Supplemental Figure S9**.

Supplemental Figure S9. Additional examples of proteins translocating toward the peroxisome-endosome cosidementing compartment upon thapsigargin treatment. (Left) Alluvial plot of significant protein translocation ($Pr > 0.95$) from the ER, Golgi apparatus (GA), and nucleus toward the peroxisome. Colors correspond to spatial maps for AC16 cells throughout the manuscript. (Right) Ultracentrifugation fraction profile of CNOT3, CNOT7, CNOT10, LMAN1, LMAN2, SCYL2, SNX1, GOLT1B, GOSR2, RER1, and NAPA showing the localization of the proteins to the ER and to the peroxisome/endosome fraction in normal and thapsigargin-treated AC16 cells, respectively. X-axis: fraction 1 to 10 of ultracentrifugation. Y-axis: relative channel abundance. Bold lines represent the protein of interest; light lines represent ultracentrifugation profiles of all proteins classified to a respective localization. Colors correspond to subcellular localization for all AC16 data throughout the manuscript. Numbers in the box represent BUNDLE localization probability to the compartment. RNA Granule Score: score from RNA Granule Database (<https://rnagranuledb.lunenfeld.ca/>). A score of 7 or above is considered a known stress granule protein. Phi: predicted phase separation participation. RBP: Annotated RNA binding protein on the RNA Granule Database. One circle denotes known RNA binding proteins (RBP) in at least one data set; two circles denote known RBP in multiple datasets.

Reviewer 2 Comment 26: “380: refer to Fig. S3A”

Response: Thank you. The text now refers to **Supplemental Figure S12** for the discussion in question.

Main Text, Results, line 502 “Notably, although tunicamycin also induced the translocation of proteins toward the peroxisome/endosome fraction, different proteins are involved, including the stress response proteins DNAJB11, DNAJC3, DNAJC10, and PDIA6 as well as other proteins EMC4, EMC8, VAPA, and VAPB (**Supplemental Figure S12**) which further outlines different modalities of cellular response toward two different ER stress inducers.”

Reviewer 2 Comment 27: 380-383: where can this data be found?

Response: We apologize for the poor presentation. In the revision, we have endeavored to include an individual figure presentation for every protein mentioned. The discussion in question (same as **Reviewer 2 Comment 26** above) is now in **Supplemental Figure S12**.

Main Text, Results, line 502 *“Notably, although tunicamycin also induced the translocation of proteins toward the peroxisome/endosome fraction, different proteins are involved, including the stress response proteins DNAJB11, DNAJC3, DNAJC10, and PDIA6 as well as other proteins EMC4, EMC8, VAPA, and VAPB (Supplemental Figure S12) which further outlines different modalities of cellular response toward two different ER stress inducers.”*

Supplemental Figure S12 is reproduced under our response to **Reviewer 2 Comment 21**.

Reviewer 2 Comment 28: *“408-410: discussed data is not shown”*

Response: We thank the Reviewer for the comment. In response, we have opted to remove the discussion in question (proteins that move from the plasma membrane toward internal membrane) from the main text in order to maintain scope and focus. Due to the large-scale nature of the spatial proteomics experiments, we feel unable to highlight each protein individually despite already adding over 100 graphical elements (e.g., individual fractionation profiles) in the revision to ensure graphical representation of the discussed proteins. The behavior of these differential localization candidates can be found in **Supplemental Data S4** and **Supplemental Data S6**.

Reviewer 2 Comment 29: *“412-414: refer to Fig. 4A”*

Response: We apologize for the omission. The results of the tunicamycin comparison has been moved to its own section in the main text under the section “Spatiotemporal proteomics highlights similarities and differences of ER stress induction protocols” in response to the Reviewer’s comment. In addition, we have made every effort to include in the figure panels all proteins discussed throughout the main text. The discussion in question (lysosome targeting in thapsigargin and tunicamycin) is now associated with individual protein graphs in **Figure 5F**.

Main Text, Results, Line 487 *“We found that in both tunicamycin and thapsigargin treatment, there was evidence of lysosome targeting from other endomembrane compartments, including: RRBP1, a ribosome-binding protein of the ER, GANAB, a glucosidase II alpha subunit integral to the proper folding of proteins in the ER, FKBP11, a peptidyl-prolyl cis/trans isomerase important to the folding of proline-containing peptides, IKBIP, an interacting protein to the IKBKB nuclear kinase, and MANF, a neurotrophic factor which has relations to ER stress-related cell death when its expression is lowered (BUNDLE differential localization probability > 0.999) (Sayers et al., 2022) (Figure 5F).”*

Figure 5F. Alluvial plot showing the migration of ER, GA, and peroxisome/endosome proteins toward the lysosome (left). On the right, the ultracentrifugation fraction profiles of translocating proteins RRBP1, FKBP11, GANAB, MANF, IKBP, and P3H1 are shown that are targeted toward the lysosome in both tunicamycin and thapsigargin treatment (BUNDLE differential localization probability > 0.95). Numbers in boxes are the BUNDLE allocation probability in each condition ($n=3$).

Reviewer 2 Comment 30: “412-417: most mentioned proteins are not shown in the figure. What is the interpretation, could this be explained by autophagy activation?”

Response: We apologize for the omission. We have made every effort to match the proteins discussed throughout the main text to a figure panel. In this instance, the mentioned lysosome-targeted proteins are now presented in **Figure 5F**, which is reproduced immediately above in our response to **Reviewer 2 Comment 29**.

We agree with the Reviewer that autophagy activation could be an intriguing explanation for the observed lysosome targeting events. One of the translocating proteins MANF is a known ER stress response factor and in other models including mouse kidney and *C. elegans* has been found to localize to lysosome and to activate autophagy (Kim et al., 2023; Taylor and Gupta, 2023). However, since we have not investigated autophagy activation in this study, with the Reviewer’s understanding we prefer to refrain from further interpretation in the main text.

Reviewer 2 Comment 31: “419: refer to Fig. S3B”

Response: We thank the Reviewer for the suggestion. The text now refers to **Figure 5F**, as noted in our response to Reviewer Comment 29 above.

Reviewer 2 Comment 32: “429-430: give examples of stress response proteins that are found in this group”

Response: We thank the Reviewer for the suggestion. We now give specific examples in the main text on the stress response proteins that alter in kinetics and localization. The identity and associated fold-change and statistics can be found in **Supplemental Data S6**.

Main Text, Results, Line 506 “The translocating stress response proteins DNAJB11, DNAJC10, and PDIA6 also showed significant acceleration in temporal kinetics in tunicamycin (Mann-Whitney test, FDR adjusted $P < 0.10$; **Supplemental Data S6**)”

Reviewer 2 Comment 33: “462–466: where can this data be found?”

Response: In the revised manuscript, we now present Figure 4A–4B that more clearly highlights the similarity in spatial distance between light and heavy SILAC labeled versions of the same protein species, which is detailed in our response to Reviewer 2 Comment 20 above and is now reproduced here. Moreover, regarding the translocation of heavy and light proteins, we now present data in **Figure 4C** and **Supplemental Figure S11C** that show that the majority of proteins do not deviate significantly in light-heavy spatial distance upon thapsigargin or tunicamycin treatment, with the highlighted proteins EGFR and ITGAV being among unusual cases (Z score $> \sim 2$). The light-heavy profiles (i.e., new and old EGFR, and ITGAV) of these proteins are further presented in the updated **Figure 4D–4G** and the new **Supplemental Figure S11D–E**

Figure 4C. Ranked changes in heavy-light pair euclidean distance upon thapsigargin treatment. The difference in heavy-light distances in thapsigargin is adjusted by the average changes in the spatial distance of the light protein with 250 other sampled light proteins to calculate the normalized difference. The majority of proteins show no change (± 0.02 in euclidean distance). The positions of EGFR and ITGAV are highlighted. Inset: Z score distribution of all changes.

Reviewer 2 Comment 34: “468–470: where can this data be found?”

Response: The new **Supplemental Figure S11** has been added to show the partition of new and old proteins and the ER retention of new proteins in tunicamycin treated AC16 cells.

Supplemental Figure S11: Partition of new and old EGFR and ITGAV in tunicamycin treatment. **A.** Histogram showing the similarity in light and heavy proteins in normalized spatial distribution distances in (tunicamycin-treated AC16 cells. X-axis: euclidean distance of fraction profiles across 3 replicates; y-axis: count. Blue: distance for quantified light-heavy protein pairs. Grey: distribution of each corresponding light protein with another random light protein. P value: Mann-Whitney test. **B.** Proportion of heavy-light protein pairs with confidently assigned localization that are assigned to the same location (purple) in normal (left; 93.0%) and tunicamycin-treated (right; 85.2%) cells. **C.** Ranked changes in heavy-light pair euclidean distance upon tunicamycin treatment. The majority of proteins show no change (± 0.02 in euclidean distance). The positions of EGFR and ITGAV are highlighted. Inset: Z score distribution of all changes. The spatial maps for **D.** EGFR and **E.** ITGAV showing a translocation of newly synthesized (heavy; H) but not old (light; L) proteins from the plasma membrane (PM) to the ER fraction in tunicamycin treated AC16 cells. Open circles show the location of the proteins in the map. Numbers denote BUNDLE allocation probability. (Right) Ultracentrifugation profiles showing different sedimentation behaviors of the light and heavy proteins.

Reviewer 2 Comment 35: “477-479: tunicamycin data is not shown”

Response: We apologize for the omission. The new **Supplemental Figure S11** has been added to show the data in tunicamycin treated AC16 cells. Please see our response to Reviewer 2 Comment 34 immediately above.

Reviewer 2 Comment 36: “493-503: an alternative explanation would be folding stress induced stalling of protein trafficking along the secretory pathway, which is even stated in the abstract (line 37-40)”

Response: We apologize for the unclear writing, but this was in fact our preferred explanation as well rather than an alternative explanation, so the Reviewer and we are in agreement. We have now rewritten this for clarity.

Main text, Results, line 402 “With the function of integrins as cell surface receptors that function in intracellular-to-extracellular and retrograde communication, the internalization of a newly synthesized V class integrin subunit, such as due to stress-induced stalling of protein trafficking along the secretory pathway, would be indicative of a decrease in integrin signaling function through spatial regulation rather than protein abundance ...”

Reviewer 2 Comment 37: “519-520: title is somewhat misleading because the utilized PI concentration did not cause sufficient inhibition of proteasomal degradation, as shown in Fig. 6B”

Response: We agree with the Reviewer that the title is less than ideal. In the revised manuscript, we have revised the heading of the iPSC-CM section to “Application of SPLAT to the mechanism of cardiotoxicity in iPSC-CM models”. We have also included new proteasome assay data to show that there is reduced chymotrypsin-like proteasome activity at 48 hours after 0.5 μ M carfilzomib treatment to approximately 25% of normal iPSC-CM, which is now presented in the new **Figure 6K**. The figure is detailed in our response to **Reviewer 2 Comment 8** and is not further reproduced here.

Reviewer 2 Comment 38: “536-537: add references for prior reports”

Response: We apologize for the omission. The literature reference has now been included in the main text.

Main Text, Results, Line 543 “This cardiotoxicity has been modeled in vitro by exposure of 0.01 – 10 μ M carfilzomib to human iPSC-CMs (Forghani et al., 2021) [...]”

In addition, in response to the Reviewer’s comments (e.g., see our response to **Reviewer 2 Comment 8** above), we have included substantial new experiments to show cell viability, metabolic dysfunction, proteasome degradation and autophagy activity under the used dose.

Reviewer 2 Comment 39: “538: typo ‘hands’”

Response: This has now been fixed.

Reviewer 2 Comment 40: “564-567: refer to Fig. 6D”

Response: The text has been updated to refer to figure panels more consistently. Protein turnover changes in carfilzomib-mediated cardiotoxicity is now presented in the new **Supplemental Data S8** as well as the new **Figure 7A** (please see our response to **Reviewer 2 Comment 41** immediately below)

Reviewer 2 Comment 41: “567-569: are the protein half-lives provided anywhere?”

Response: We apologize for the omission on our part. The protein turnover changes in carfilzomib-mediated cardiotoxicity are now presented in tabular formats in the new **Supplemental Data S8**. The turnover kinetic changes of the discussed proteins are now also individually presented in the new **Figure 7A** and **Figure 7B**.

Figure 7A. Changes in protein turnover rates between carfilzomib vs. normal iPSC-CMs across selected cellular compartments; **: $P < 0.01$; *: $P < 0.05$; .: $P < 0.1$; Mann-Whitney test FDR adjusted P values. error bars: standard error. **B.** Kinetic curve representations of proteins with accelerated temporal kinetics in carfilzomib including PSMC2 which corresponds to the ratio in panel A, as well as additional ERAD proteins and chaperones; gray: normal iPSC-CM; green: carfilzomib.

Reviewer 2 Comment 42: “573-574: how many of these proteins showed differential localization upon stress?”

Response: We thank the Reviewer for the prompt. Upon carfilzomib treatment, we find 339 heavy/light pairs of proteins that translocate, among which 23 also change in temporal kinetics. In response to the Reviewer’s comment to focus on biological insights, we now also highlight 2 specific examples in the main text.

Main Text, Results, Line 621 “We observed an interconnectivity of spatial and temporal changes, with 23 out of 339 pairs of confident translocators also showing significant kinetic changes. BAG3, a muscle chaperone important for sarcomere turnover (Martin et al., 2021a), shows elevated kinetics (Figure 7B) and a partition away from the soluble cytosol compartment (Figure 7D) toward an expanded compartment in carfilzomib that co-sediments with Golgi markers. Inspection of existing annotations show that the majority of categorized proteins are not canonical Golgi proteins but contain cytoplasm and endosome terms; hence it likely represents a less soluble cytoplasmic fraction consistent with a lower abundance in the last ultracentrifugation step (Supplemental Figure S13, Supplemental Figure S14, Supplemental Data S9). This is consistent with the known dynamic partitioning of BAG3 between the cytosol and myofibril fractions for its function (Martin et al., 2021b). Secondly, we find that accelerated temporal kinetics of PA200/PSME4 proteasome inhibitor

(Figure 7B) in conjunction with a change in localization from the nuclear compartment in baseline toward the proteasome compartment upon carfilzomib (Figure 7E).”

The kinetic changes of BAG3 and PSME4 are presented in Figure 7B (see our response to Reviewer 2 Comment 41 immediately above). The translocation results are also presented in the new Figure 7D and Figure 7E. Moreover, the localization and turnover data can be found in Supplemental Data S7, Supplemental Data S8, and Supplemental Data S9).

Figure 7D-E. Spatial map (PC1 vs. PC2) and ultracentrifugation fraction profiles of **D.** BAG3 and **E.** PSME4 in normal and carfilzomib-treated human iPSC-CM, showing a likely differential localisation in conjunction with kinetics changes. White-filled circles: light and heavy BAG3 or PSME4 in each plot. The kinetic curves of BAG3 and PSME4 are in panel B. Numbers at arrows correspond to BANDLE differential localization probability (Diff. Loc. Pr.).

Reviewer 2 Comment 43: “642-644: immunofluorescence cannot distinguish between old/new protein pool; hence, these results do not support the finding that only the young protein pool shifts location. Please rephrase.”

Response: We agree with the Reviewer and thank the Reviewer for the recommendation to rephrase. This has now been rephrased for clarity.

Main Text, Discussion, Line 718 “Upon ER stress induction, EGFR immunofluorescence showed a partial translocation away from the plasma membrane. Although immunofluorescence cannot distinguish between old and new protein pools, this partial translocation is consistent with the mass spectrometry data showing partial translocation, involving the newly synthesized heavy protein pool.”

Reviewer 2 Comment 44: “644-649: please be careful with interpretation of the results (as stated for lines 493-503). Otherwise provide additional data that support the ‘ligand-independent EGFR internalization’ hypothesis.”

Response: We thank the Reviewer for the suggestion and have now rephrased the interpretation with better care to emphasize the hypothesis generated.

Main Text, Discussion, Line 722 “We hypothesize that this partial translocation is suggestive of a ligand-independent trafficking of newly synthesized protein, rather than ligand dependent activation and internalization that is agnostic to protein lifetime. Of note, ligand-independent activation and internalization of

EGFR has been previously induced via both starvation and tyrosine kinase inhibitor treatment, leading to cellular autophagy (Tan et al., 2015), hence this partial translocation may carry functional significance to protective cellular response.”

Reviewer 2 Comment 45: “675-676: show data or give reference”

Response: We now point to relevant data generated in this study (**Figure 1D** and **Figure 6B**), and have further reworded the sentence for clarity.

Main Text, Discussion, Line 746 “...in our experiments we have used only a single time point per treatment (16 hours post thapsigargin or tunicamycin; 48 hours post carfilzomib), which was selected based on the drug treatment models but also needed to be sufficient to capture the median half-life of proteins in the cell types studied (**Figure 1D**; **Figure 6B**). Hence the collection time points need to be optimized for different cell types with distinct intrinsic protein turnover rates.”

Reviewer 2 Comment 46: “Figure 1B: think about emphasizing temporal information gained in MS1 and spatial information gained in MS2.”

Response: We thank the Reviewer for the suggestion. We have revised the **Figure 1** illustrations and legends to emphasize the temporal and spatial information gained in MS1 and MS2, respectively.

Figure 1D. Temporal information and spatial information is resolved in MS1 and MS2 levels, respectively. SPLAT allows the subcellular spatial information of the heavy (new) and light (old) subpools of thousands of proteins to be quantified simultaneously in normal and perturbed cells.

Reviewer 2 Comment 47: “Figure legend 1B: not clear that ‘SILAC light’ (used in figure) is the same as ‘unlabeled’ in the legend. Not clear that ‘+R[10.0083]’ (used in figure) refers to heavy Arg.”

Response: We apologize for the unclear presentation. We have now modified the legends of Figure 1B for clarity. We have also made it clearer in **Figure 1B** that +R[10.0083] refers to heavy SILAC.

Figure 1B. Dynamic SILAC labeling allowed differentiation of pre-existing (unlabeled, i.e., SILAC light) and post-labeling (heavy lysine or arginine, i.e., +R[10.0083]) synthesized peptides at 16 hours. The light and heavy peptides were isolated for fragmentation separately to allow the protein sedimentation profiles containing spatial information to be discerned from TMT channel intensities.

Reviewer 2 Comment 48: “Figure 2B: what was the reason to test against ribosomal proteins?”

Response: We apologize for the confusion. We did not present all comparisons in the figure in order to avoid having too many pairwise comparisons and P value in the figure. However, we now realize this is less than ideal. In the revised manuscript, we now compare each compartment with its complement (i.e., proteasome vs. non-proteasome, ER vs. non-ER, etc.) in the new **Figure 2E**, which offers a clearer comparison.

Figure 2E. Boxplot showing the log₂ turnover rate ratios in thapsigargin over normal AC16 cells for proteins that are localized to the ER (blue) (T) or not (F); or the Golgi (GA; green). P values: Mann-Whitney test. A Bonferroni corrected threshold of 0.05/13 is considered significant.

Reviewer 2 Comment 49: “Figure 2C: it is unclear how one time point is sufficient for such analyses.”

Response: We have now modified the text for clarity. This panel has been moved to **Figure 2F** in the new manuscript and we now explain how a single time point is sufficient for turnover determination. Please also see our response to the similar **Reviewer 2 Comment 7**).

Main Text, Results, Line 277 “Because the sampling time point is known, the measured relative isotope abundance of a peptide (prior to reaching the asymptote) is sufficient to define the kinetic curve and the parameter of interest (k).”

Reviewer 2 Comment 50: “Is the data shown in Figure 2 provided as a table?”

Response: The data shown in **Figure 2** (protein turnover rate changes upon UPR) is provided as a table containing turnover rates, ratios, P values, and adjusted P values in **Supplemental Data S3** for thapsigargin vs. normal AC16 cells, **Supplemental Data S5** for tunicamycin vs. normal AC16 cells. In addition, the previous **Figure 2A** (turnover rates vs. abundance changes) is now presented in **Supplemental Figure S7**, which is expanded to highlight the turnover rate vs. protein abundance changes in thapsigargin and tunicamycin treatment, with proteins in three compartments with particular kinetics changes (ER, Golgi, and Lysosome) highlighted. The protein abundance change information is also provided as a table in **Supplemental Data S1**.

Supplemental Figure S7. Protein abundance and turnover changes following thapsigargin and tunicamycin treatment. Scatterplots showing the relationship between \log_2 of protein abundance fold changes (x-axis) and \log_2 of turnover ratios (y-axis) in **A.** thapsigargin and **B.** tunicamycin treatment. In each of the series of scatterplots from left to right, proteins assigned to the ER (blue), GA (green), and lysosome (orange) are labeled. Overall protein kinetic changes are only modestly correlated with protein abundance changes.

Reviewer 2 Comment 51: “Figure legend 2B: which statistical test was used?”

Response: We apologize for the lack of clarity. The figure legends for the former Figure 2B (now labeled **Figure 2F**) now clearly state that a Mann-Whitney test with Bonferroni corrected threshold is used.

Main text, Results, line 271 “Boxplot showing the \log_2 turnover rate ratios in thapsigargin over normal AC16 cells for proteins that are localized to the ER (blue) (T) or not (F); or the Golgi (GA; green). P values: Mann-Whitney test. A Bonferroni corrected threshold of 0.05/13 is considered significant.”

Reviewer 2 Comment 52: “Figure 3C is not discussed in the results section.”

Response: We apologize for the omission. Figure 3 in the original submission has been reorganized substantially in the revision based on Reviewer comments. A more quantitative comparison is shown in the new **Figure 4A** to highlight the concordance of the spatial distribution of light and heavy proteins, which is now discussed in the main text:

Main text, Results, line 370 “*In both normal and thapsigargin-treated cells, we found that the independently measured spatial profiles of light (pre-existing) proteins and their corresponding heavy SILAC (newly synthesized) counterparts are highly concordant, with a normalized spatial distribution distance (see Supplemental Methods) of 0.020 [0.015 – 0.030], compared to 0.117 [0.080 – 0.155] in random pairs of pre-existing proteins (1,614 light-heavy pairs, Mann-Whitney $P < 2.2e-16$) in normal cells, and 0.028 [0.019 – 0.041] and 0.122 [0.081 – 0.161] in thapsigargin-treated cells (1,614 light-heavy pairs, Mann-Whitney $P < 2.2e-16$) (Figure 4A).*”

Figure 4. SPLAT reveals protein-lifetime dependent translocation. **A.** Histogram showing the similarity in light and heavy proteins in normalized fraction abundance profiles in (left) normal and (right) thapsigargin-treated AC16 cells. X-axis: the spatial distribution distance of two proteins is measured as the average euclidean distance of all TMT channel relative abundance in the ultracentrifugation fraction profiles across 3 replicates; y-axis: count. Blue: distance for 1,614 quantified light-heavy protein pairs (e.g., unlabeled EGFR, heavy SILAC-labeled EGFR). Grey: distribution of each corresponding light protein with another random light protein. P value: Mann-Whitney test.

Reviewer 2 Comment 53: “Figure 4: ER stress is known to cause ER membrane expansion and remodeling. This might alter the sedimentation behavior of the ER fraction in the LOPIT-DC protocol and provide an explanation for the appearance of the peroxisome/ERV fraction.”

Response: We believe this is unlikely, as in the analysis workflow, subcellular localization classification is performed for each sample separately in a supervised manner using the sedimentation behavior of canonical markers (see also our response to **Reviewer 2 Comment 1**). Similarity of sedimentation behavior can also be seen from the TMT profiles across ultracentrifugation fractions between normal and ER stress samples. The peroxisome fraction has distinct sedimentation profiles that are reflected in the PC maps.

Reviewer 2 Comment 54: “Figure 4B: include LOPIT color code legend”

Response: We thank the Reviewer for the suggestion. The color code is now added to the figure (labeled as **Figure 3B–C** in the order of the revised manuscript).

Figure 3B. Protein spatial map for SLC3A2 (open black circle) in normal (left) and thapsigargin-treated (right) AC16 cells, showing its colocalization with lysosomal proteins in normal cells and in PM proteins in thapsigargin-treated cells. Colors represent allocated subcellular localization. **C.** Ultracentrifugation fraction profile of SLC3A2 and other amino acid transporters SLC7A5, SLC1A4, SLC1A5 and ion channel proteins SLC30A1, ATP1B1, ATP1B3, and ATP2B1 with similar migration patterns. X-axis: fraction 1 to 10 of ultracentrifugation. Y-axis: relative channel abundance. Bold lines represent the protein in question; light lines represent ultracentrifugation profiles of all proteins classified to a respective localization. Colors correspond to subcellular localization in panel B and for all AC16 data throughout the manuscript; numbers within boxes correspond to BANDLE allocation probability to compartment

Reviewer 2 Comment 55: "Figure 5A: include LOPIT color code legend"

Response: We thank the Reviewer for the suggestion. The color code is now added to the figure (labeled as Figure 4D–G in the order of the revised manuscript).

Reviewer 2 Comment 56: “Figure 5C: scale bar missing”

Response: We thank the Reviewer for the reminder. A scale bar representing 90 μm is now added to the figure in question (labeled as **Figure 4H** in the order of the revised manuscript).

Figure 4H. Confocal imaging of EGFR immunofluorescence supports a partial relocation of EGFR from the cell surface toward internal membranes following thapsigargin treatment. Numbers: The mean intensity of the labeled EGFR channel of a 3 pixel border at cell boundaries was divided by mean intensity of the whole cell to estimate the ratio of EGFR at the plasma membrane to the cell interior. Blue: DAPI; Green: EGFR; scale bar: 90 μm .

Reviewer 2 Comment 57: “Figure legend 5D: Figure does NOT include ‘c’ as stated in the legend. Please remove.”

Response: We are unable to find the legend to 5D in the original submission as Figure 5 only had three panels to our knowledge. However, the figure on new-old protein has been rebuilt as Figure 4 in the current revision. We apologize for the omission and hope the issue has now been resolved.

Reviewer 2 Comment 58: “Figure 6C: what was the reason to test against ribosomal proteins?”

Response: Since most compartments show no changes in log₂ turnover ratios between carfilzomib and normal iPSC-CMs, the elevated turnover kinetics in the proteasome is contrasted with other compartments including the ribosome. We did not present all comparisons in the figure in order to avoid having too many pairwise comparisons and P value in the figure. However, we now realize this is less than ideal. In the revised manuscript, we now compare each compartment with its complement (i.e., proteasome vs. non-proteasome, ER vs. non-ER, etc.) which offers a clearer comparison in the new **Figure 6M**. Thank you for the comment.

Figure 6M. log₂ Turnover rate ratios between carfilzomib-treated and untreated iPSC-CM from the spatiotemporal proteomics data. Proteins assigned the proteasome compartment have significantly increased temporal kinetics; proteins in the lysosome/junction and chromatin/sarcomere compartments have significantly reduced temporal kinetics. P values: Mann-Whitney; with a threshold of 0.05/14 considered significant.

Reviewer 2 Comment 59: “Supplemental table S1 does not include data with tunicamycin”

Response: Thank you for the comment. We now present the tunicamycin data more clearly in a dedicated section in the main text (**Figure 5**) to highlight similarities and differences between thapsigargin and tunicamycin treatment. The tunicamycin data are now presented in separate Supplemental Data spreadsheets, in **Supplemental Data S5** (turnover rates) and **Supplemental Data S6** (localization).

We thank Reviewer 2 again for their thorough evaluation and the many helpful suggestions.

Reviewer #3 (Remarks to the Author):

“The present study utilized a combination of in vivo dynamic SILAC metabolic labeling and TMT labeling of spatially separated fractions, known as the SPLAT strategy. Initially, these techniques were applied to investigate human AC16 cells, a transformed cardiomyocyte line, as well as induced pluripotent stem cell (iPS)-derived cardiomyocytes. The authors focused on studying the effects of two common unfolded protein response (UPR) models, thapsigargin and tunicamycin, on the proteome in AC16 cells. They observed a more severe slowdown in lysosome protein turnover compared to the endoplasmic reticulum. Additionally, the SPLAT strategy was employed to study a cardiotoxicity model using human iPS-derived cardiomyocytes exposed to the proteasome inhibitor carfilzomib. This analysis revealed potential disruptions in carbohydrate metabolism and contraction proteins as mechanisms underlying carfilzomib-induced cardiotoxicity. An advantage of the SPLAT strategy is its ability to simultaneously encode temporal and spatial protein information through isotope labels in the MS1 and MS2 levels.”

Response: We thank the Reviewer for a thorough review of our manuscript and for the many helpful comments. In response to the Reviewer, we have incorporated major improvements to the manuscript, including the following:

- We have performed new immunofluorescence microscopy experiments to corroborate the translocation of SLC3A2
- As recommended, we have performed an in vivo validation experiment in C57BL/6N mice treated with cardiotoxic doses of carfilzomib to support the in vitro experiment results.
- We have included new analyses to demonstrate consistent fractionation and organelle integrity in stressed cells, and to show that the measured protein localizations agree closely with expected subcellular compartments in prior annotations.

In our view the Reviewer’s helpful suggestions have led to a much improved manuscript. Please see below for our itemized responses.

Reviewer 3 Comment 1: *“Figure 1, the number of independent experiments and the observed variation between independent replicates were not explicitly mentioned. It would be beneficial for the authors to provide this information to better understand the reliability and reproducibility of the results.”*

Response: We thank the Reviewer for the suggestion and now clarify the number of replicates in Figure 1.

Main text, Results, Line 177: *“For each condition, 3 biological replicate SPLAT experiments were performed (n=3).”*

Reviewer 3 Comment 2: *“In Figure 2B, it is highlighted that the protein turnover change is highest in the nucleus/chromatin compartment. However, there may be concerns about potential misannotation since the heavy labels in Figure 1E show significant overlap between nuclei/chromatin and ribosomes. The authors should address this issue and provide further clarification to ensure accurate annotation of the compartments.”*

Response: We thank the Reviewer for the comment. The original Figure 2B has been redrawn for clarity. In any case, we would like to address the comment about potential misannotations, and note that in the spatial maps, only PC1 and PC2 are shown, whereas all TMT channels for the ultracentrifugation profiles are used for localization. The updated **Figure 2C** for example shows that the majority of the proteins classified to be in the nucleus or the ribosome are also annotated to be in such compartments in UniProt GO term annotations,

which corroborate the reliability of the compartment classification. The new **Supplemental Figure S2** also shows that the ultracentrifugation profiles between nucleus and ribosome markers are distinguishable despite some similarity.

New Figure 2C. Distribution of light (unlabeled) protein features in each of the 12 subcellular compartments ($n=3$); fill color represents whether the protein is also annotated to the same subcellular compartment in UniProt Gene Ontology Cellular Component terms

Reviewer 3 Comment 3: “The “t” in the legend of Figure 2B likely refers to the “t-test.” It is important for the authors to mention whether they performed adjustments for multiple comparisons?”

Response: Thank you for the comment. Where applicable, all P values are multiple testing adjusted, using Tukey’s post-hoc tests for ANOVA comparison (e.g., **Figure 6C–H**), or using Benjamini-Hochberg procedures for protein turnover and abundance calculations (e.g., **Figure 2A**, **Figure 5A**, other proteomics results). These are now more clearly presented in the legend. The original Figure 2B has been updated as the new **Figure 2E** to more clearly highlight the difference between ER/non-ER and GA/non-GA proteins in response to the comments of the Reviewers. The text now clearly notes that presented t-test P values are below the Bonferroni corrected family wise error rate (0.05/13 compartments).

Figure 2E. Boxplot showing the log₂ turnover rate ratios in thapsigargin over normal AC16 cells for proteins that are localized to the ER (blue) (T) or not (F); or the Golgi (GA; green). P values: Mann-Whitney test. A Bonferroni corrected threshold of 0.05/13 is considered significant.

Reviewer 3 Comment 4: “To address the potential for cross-contamination during fractionation of subcellular compartments, it would be valuable to confirm more translocation events using immunostaining techniques. For example, in addition to EGFR, the authors should consider examining the translocation of newly synthesized cathepsin A from the lysosome to the peroxisome (Suppl Figure 4) with tunicamycin and the translocation event of newly synthesized ITGAV to the ER fraction in the presence of thapsigargin (Suppl Figure 5).”

Response: We thank the Reviewer for this important comment. We would like to clarify that cross-contamination during fractionation is not an inherent issue because the LOPIT/PCP type of approaches do not require that pure compartments be isolated in any fractions, but simply that localization can be recognized by distribution patterns across ultracentrifugation fractions using supervised learning with canonical markers. Because a spatial map is constructed separately for each experiment, the classification is robust to a certain extent to differences in fractionation procedures, e.g., if one of the fractions were to be dropped, or swapped, or combined with another.

Nevertheless, the Reviewer's point is well taken. In the revised manuscript, we have taken both analytical and experimental approaches to further evaluate the translocation events. For example, we show that in each experiment, the ultracentrifugation profiles and separation of compartments are highly reproducible; the proportion of proteins in each compartment that map to the same compartments according to UniProt Gene Ontology Cellular Component (CC) annotation terms is also similar. Please see our response to **Reviewer 3 Comment 5** below for more details. In parallel, we have included additional experimental validation of SLC3A2 translocation. This target was chosen because it was also suggested by Reviewer 1 and is one of the prominently highlighted examples in the text (see our response to Reviewer 1 Comment 1). Briefly, the immunostaining results corroborate the translocation event of SLC3A2 away from the lysosome. This result is now presented in the new figure panels **Figure 3D–3E** in the main text.

Figure 3. SPLAT captures extensive protein translocation in AC16 cells under UPR. **D.** Immunofluorescence of SLC3A2 (red) against the lysosome marker LAMP2 (green) and DAPI (blue). Numbers in cell boundary: colocalization score per cell. Scale bar: 90 μm . **E.** Colocalization score (Mandle's correlation coefficient) between SLC3A2 and LAMP2 decreases significantly (t -test P value: $3e-8$) following thapsigargin treatment, consistent with movement away from lysosomal fraction ($n = 205$ normal cells, $n = 32$ thapsigargin treated cells).

In parallel, we have also attempted an immunostaining experiment of the translocation of ITGAV. However, although we may see potential signs of differential spatial distribution, the results were not conclusive. We reason this experiment is inherently challenging because the events in question are partial translocations due to the partition of new and existing proteins that cannot be directly distinguished by common immunostaining techniques; hence the microscopy analysis can only capture net partial changes in

localization. In some instances, immunobiological approaches may also suffer from antibody specificity issues that do not affect mass spectrometry results. The result of this experiment is presented below in **Reviewer Figure R4** for the Reviewer's reference.

Reviewer Figure R4. Immunofluorescence of ITGAV (red) against DAPI (blue) in normal (control) and thapsigargin-treated human AC16 cells. Scale bar: 50 μm . "2° only" represents a slide incubated without primary antibody as negative control.

Reviewer 3 Comment 5: "Since the fractionation of subcellular compartments is achieved through ultracentrifugation, it is necessary to demonstrate whether the isolation process preserves the integrity of cellular organelles under protein misfolding stress or cardiotoxicity, particularly in comparison to untreated cells."

Response: We thank the Reviewer for an important comment, and agree that we will need to demonstrate whether the cellular organelle integrity is preserved in the ultracentrifugation fractionation method across normal and stressed cells. The following analyses are now more clearly presented that strongly point to reproducible compartment integrity and fractionation.

- (1) Firstly, the new **Supplemental Figure S1** is presented to show comparable canonical marker separation in untreated vs. thapsigargin and tunicamycin treated cells. Therefore, the subcellular compartments remain well separable from the same markers after stress, which is consistent with preserved integrity and biophysical properties of the cellular organelles under protein misfolding stress. The new **Supplemental Figure S10** demonstrates the same applies in the case of iPSC-CM cardiotoxicity.

Supplemental Figure S1. Separation of subcellular component markers in the spatial proteomics data. Scatter plots of the first (x-axis) and second (y-axis) principal components of ultracentrifugation fraction profiles are shown for each experimental condition: **A.** normal, **B.** thapsigargin, and **C.** tunicamycin treated AC16 cells, $n=3$ each. Each data point is a protein species. The colored data points correspond to marker proteins known to reside in each of 12 subcellular locations used to train the classification models, showing clear and consistent separation across the experimental conditions. Colors correspond to other spatial maps for AC16 cells throughout the manuscript.

Supplemental Figure S10: Subcellular localization of proteins following tunicamycin treatment. PC1 and PC2 of proteins spatial map showing the localization of confidently allocated proteins in tunicamycin-treated AC16 cells. Each data point represents a protein; color represents classification of subcellular localization consistent with other AC16 cell data throughout the manuscript.

(2) Second, the new **Supplemental Figure S2** now shows highly reproducible fraction distributions across the same ultracentrifugation speeds/durations in the 10-step ultracentrifugation process between untreated and treated cells. This provides strong evidence of effective fractionation and preserved integrity of cellular organelles under protein misfolding stress, as the dissolution of organelles integrity would be expected to lead to substantial change in sedimentation profile due to change in density (e.g., a substantial portion of the organelle markers might then be found in the last ultracentrifugation step with the highest speed where normally soluble cytosolic proteins reside). The new **Supplemental Figure S13** demonstrates the same applies in the case of iPSC-CM cardiotoxicity.

Supplemental Figure S2. Ultracentrifugation fraction distributions of cellular component markers. Replicate one of each experimental condition is shown. The line plots show the normalized abundance (y-axis) of marker proteins for each subcellular localization experiment across ultracentrifugation fractions (x-axis) as measured by the TMT channel intensities. The fractions correspond to the ultracentrifugation steps in Supplemental Table S3. Colors correspond to spatial maps for AC16 cells throughout the manuscript. Black lines show average trend line and standard deviation, showing consistent sedimentation profiles of subcellular localization in the **A.** normal, **B.** thapsigargin, and **C.** tunicamycin treated AC16 cells ($n=3$).

New Supplemental Figure S13. Ultracentrifugation fraction distributions of cellular component markers in human iPSC-CMs. Additional iPSC-CM specific compartments and markers were curated manually, including a sarcomere and a cell junction compartment. The compartments were merged with the chromatin and the lysosome compartments due to similarity in sedimentation profile under the present ultracentrifugation scheme. Replicate one of each experimental condition is shown. **A.** Control iPSC-CM. **B.** iPSC-CM treated with 0.5 μM carfilzomib, 48 hours. The line plots show the normalized abundance (y-axis) of marker proteins for each subcellular localization experiment across ultracentrifugation fractions (x-axis) as measured by the TMT channel intensities. The fractions correspond to the ultracentrifugation steps in Supplemental Table S3. Colors correspond to spatial maps for iPSC-CMs cells throughout the manuscript. Black lines show average trend line and standard deviation, showing consistent sedimentation profiles upon carfilzomib treatment.

(3) Thirdly, the new **Figure 2C** and **Supplemental Figure S5** show similar numbers and proportions of proteins in each assigned compartment with known UniProt Gene Ontology Cellular Component (CC) term annotation to the same location in untreated, thapsigargin, and tunicamycin treated cells. This provides strong evidence that the precision and recall of protein localization classification is not

affected by protein misfolding stress. The new **Supplemental Figure S14A** shows the same applies in iPSC-CM cardiotoxicity.

New Figure 2C. Distribution of light (unlabeled) protein features in each of the 12 subcellular compartments ($n=3$); fill color represents whether the protein is also annotated to the same subcellular compartment in UniProt Gene Ontology Cellular Component terms

New Supplemental Figure S5. Correspondence of spatial classification with prior annotations in stress cells. As related to main Figure 2C, the bar charts show the number of light (i.e., non-heavy-SILAC labeled) proteins (y -axis) classified to each of 12 subcellular locations (x -axis) in thapsigargin (left) and tunicamycin (right) treated AC16 cells ($n=3$). The colors represent whether proteins classified to each subcellular location are also known to reside in the subcellular component of question in Gene Ontology Cellular Component terms retrieved from UniProt. In normal, thapsigargin, and tunicamycin treated AC16 cells, 69.5%, 71.9%, and 63.0% of classified proteins are consistent with known annotations, respectively; hence the classified subcellular localization match the expected assignments from prior knowledge and are not substantially affected by cellular stressors. The expanded peroxisome compartment in stressed AC16 cells primarily contained non-peroxisome annotated proteins that co-sedimented with the trained peroxisome compartment, and are referred to as the peroxisome/endosome compartment in the manuscript.

A
New Supplemental Figure S14. Correspondence of spatial classification with prior annotations in human iPSC-CMs. **A.** The bar charts show the number of light (i.e., non-heavy-SILAC labeled) proteins (y-axis) classified to each of 13 subcellular locations (x-axis) in normal (left) and carfilzomib-treated (right) treated iPSC-CMs. Colors represent whether proteins classified to each subcellular location are also known to reside in the subcellular component of question in Gene Ontology Cellular Component terms retrieved from UniProt. In carfilzomib treated cells, a number of proteins are classified as co-sedimenting with Golgi markers; the top associated GO Cellular Component terms of these proteins are shown on the right and suggest they contain cytoplasmic proteins and proteins with multiple locations. In normal and carfilzomib-stressed cells, 70.8% and 63.0% of classified proteins are consistent with known annotations, respectively.

Please see also our similar responses to **Reviewer 2 Comment 1** and **Reviewer 2 Comment 2**, which also prompted us to show more demonstration of reproducible ultracentrifugation fractionation between normal and stressed cells. We thank the Reviewer for the reminder to include this information in the manuscript.

Reviewer 3 Comment 6: “Considering the authors' previous publication on cardiac remodeling (*Nat Commun* 2018) and the reported similarity of turnover among interacting partners, it would be valuable to compare the protein turnover in AC16 cells to that in mouse hearts. This comparison would provide insights into the generalizability of the results from AC16 cells. Alternatively, similar experiments should be conducted in other cardiomyocyte-like cell types, such as iPS cells.”

Response: We thank the Reviewer for this suggestion and for the opportunity to engage with the Reviewer on a topic we have a long-standing interest in. When comparing the turnover rates measured in human iPSC-derived cardiomyocytes to the protein turnover rates in the hearts of male C57BL6/J mice in a recent study we published (Hammond et al., 2022), we see a very weak but nonetheless statistically significant correlation (Person's r in $\log_{10}(k)$: 0.20, P : 2.8×10^{-6}). However, we believe a direct comparison of in vitro vs. in vivo protein turnover rates would be challenging for a number of reasons. Cell culture grown in vitro is not under the nutrient availability and metabolic constraints of an adult animal, and hence not subject to allometric scaling laws (West et al., 2002). Secondly, the protein turnover rates measured from bulk heart tissues comprise proteins from multiple cell types, which can have different proliferation and turnover rates. Thirdly, this comparison is complicated by species differences (mouse vs. human). Fourthly, in vivo protein turnover rates are affected by animal age, which is not captured in cell culture in vitro. These represent general limitations of measuring protein turnover in vitro faced by the whole field, which we have also discussed in prior work (Hammond et al., 2022). Nevertheless, some general parallels can be drawn between the measured turnover rates here and those from in vivo studies. For instance, as the Reviewer suggested, protein complex subunits tend to share similar turnover, which has already been extensively reported by us and by others both in vivo and in vitro (e.g., (Deberneh et al., 2023; Fornasiero et al., 2018; Lam et al., 2014; Lau et al., 2018; Li et al., 2014; Mathieson et al., 2018)). This can be gleaned from **Supplemental Figure S6** through the observation

that the ribosome and proteasome compartments occupy a limited range over the entire proteome, as these compartments essentially represent one or few protein complexes. Secondly, proteins in the mitochondria tend to have longer half-life than those outside the mitochondria, consistent with what we noted previously in the mouse heart, liver, kidney, and skeletal muscle (Hammond et al., 2022; Kim et al., 2012; Lam et al., 2014; Lau et al., 2016), likely reflecting a specialized protein quality control mechanism for the mitochondria, and this can also be seen in the data summarized in **Supplemental Figure S6**.

Supplemental Figure S6. Subcellular localization differences in protein turnover rate. Boxplots showing the \log_{10} protein turnover rates (k) of proteins assigned to each of 12 subcellular localizations in normal (left), thapsigargin (middle), and tunicamycin (right) treated AC16 cells ($n=3$).

Reviewer 3 Comment 7: “The phenomenon of “end-membrane stalling” is intriguing, but it is important to explore the functional consequences associated with it. Additionally, the finding that newly synthesized proteins are more likely to translocate could be explained by their lack of incorporation into protein complexes. The authors should explore these potential functional consequences and provide further insights into the implications of their findings.”

Response: We thank the Reviewer for the comment. While we agree with the Reviewer that this is an intriguing result, and one that in our view showcases the rationale to combine temporal and spatial proteomics analyses, we respectfully suggest that the investigation of functional consequences would require orthogonal methods to distinguish and manipulate new vs. old proteins, and is therefore rather non-trivial. Although the temporal dynamics of proteins can in some cases be validated using fluorescence timer tags, these specialized techniques require extensive validation themselves for each targeted protein, and may also affect the degradation and localization of the proteins through the constructs. We therefore suggest that these experiments are better suited for future follow up studies.

Reviewer 3 Comment 8: “The findings regarding the mechanism of carfilzomib cardiotoxicity would benefit from *in vivo* validation. It is important to investigate whether the disruptions of carbohydrate metabolism and contraction proteins can be reversed or attenuated to mitigate cardiotoxicity. *In vivo* experiments would provide a more comprehensive understanding of the therapeutic implications.”

Response: The major goal of our study is to establish an *in vitro* method to encode spatial and temporal information within one mass spectrometry experiment. We have applied the method to the cardiotoxic mechanisms of carfilzomib to demonstrate its applicability to a different, non-proliferative cell type for generating biologically relevant findings, e.g., overall protein degradation rate is unchanged upon carfilzomib

but the turnover of proteins in specific compartments such as sarcomeric proteins are preferentially affected. We believe that extensive validation and follow-up of specific targets would be beyond the scope of this work.

Nevertheless, the Reviewer's point is well taken. In response to the Reviewer's comment, we have now included a new experiment from the hearts of mice exposed to carfilzomib. Briefly, wild-type C57BL/6N mice were treated with 8 mg/kg carfilzomib or vehicle (n=5 each) via i.p. Injection twice a week for up to two weeks following established literature protocol (Efentakis et al., 2021). Cardiotoxicity is confirmed using echocardiography to show significant reduction in fractional synthesis and ejection fraction as reported. A global proteomics profiling experiment comparing the expression of 3,379 proteins showed that among the most significantly regulated proteins (top 1 percentile) are myosin heavy chain beta (MYH7) and desmoplakin (DSP), (limma FDR adjusted P values < 0.05). Both proteins were shown in our iPSC-CM studies to show decreased turnover which is consistent with accumulation under proteasome inhibition. Moreover, pathway analysis shows a specific repression of metabolism-related terms including glyconeogenesis. Therefore in our view, this limited validation experiment corroborates the findings from iPSC-CMs. Moreover, relatively few proteins are changed in statistically significant manners on an individual protein level (25 out of 3,379 at 10% FDR); the results corroborate the potential utility of using protein dynamical parameters to interrogate molecular changes following proteostatic interruptions.

The relevant in vivo experiment data are now presented in the new **Supplemental Figure S16** (see next page). The new set of mouse heart carfilzomib treatment proteomics data has been deposited on ProteomeXchange under the accession PXD046670.

New Supplemental Figure S16. In vivo cardiac effect of carfilzomib. A. Ejection fraction (EF) and fractional shortening (FS) of carfilzomib (CFZ) treated mice ($n=5$ for week 0 and week 1, $n=2$ for week 2), and vehicle (Veh) treated mice ($n=5$), injected twice weekly. Two-way ANOVA with Tukey's correction, p -value 0.0001 to 0.001: ***, p -value 0.001 to 0.01: **, p -value 0.01 to 0.05: *. **B.** Gene set enrichment analysis of protein quantification (Carfilzomib vs. DMSO) showing a number of significantly enriched terms (y -axis) implicated in cardiac dysfunction and mitochondrial changes. Size denotes number of quantified proteins in the gene set, color: GSEA FDR adjusted P value; x -axis: GSEA normalized enrichment score (NES). **C.** Two weeks of carfilzomib treatment led to 11 differentially expressed proteins at 5% FDR (25 at 10% FDR) in the mouse heart ($n=2$ for carfilzomib treated mice, $n=5$ for vehicle) out of 3379 quantified proteins. The significant proteins are visualized in bar charts to show the normalized expression in vehicle and carfilzomib treatment, highlighting the accumulation of MYH7 and DSP in carfilzomib. Error bars: median absolute deviation. *: limma FDR adjusted $P < 0.05$; **: limma FDR adjusted $P < 0.01$.

In addition, the main text has been updated to reference this new result.

Main Text, Results, Line 646 *“Finally, we assessed the protein-level expression profiles in the hearts of mice treated with carfilzomib for 2 weeks to model cardiac dysfunction (Supplemental Methods). Notably, we find differential protein abundance analysis showed that MHC- β (MYH7) and desmoplakin (DSP) are the 1st and 5th most significantly up-regulated proteins among 3,379 quantified proteins in the hearts of mice treated with carfilzomib (**Supplemental Figure S16**), consistent with their accumulation following proteasome inhibition and suggesting the possibility that similar proteostatic lesions may underlie cardiotoxicity mechanism in vivo.”*

Minor

Reviewer 3 Comment 9: *“When mentioning the potential impurities in TMT tags leading to spillover to neighboring channels, it would be helpful for the authors to include a reference or provide supporting evidence for this statement.”*

Response: We now include the reference (Searle and Yergey, 2020) in the manuscript, which details some of the rationale and methodology behind using matrix factorization to correct for isotope spillover, and also provides evidence of a secondary benefit of impurity correction, namely reducing the TMT ratio compression effect. Moreover, we have updated the main text and Supplemental Information for clarity, and a new **Supplemental Table S1** to document the isotope impurities of the used TMT lots as provided by the manufacturers (Thermo Scientific) has been added:

Main text, Results, line 150 *“Because the TMT data are row normalized in the LOPIT-DC design, we incorporated correction of isotope contamination of TMT channels based on the batch contamination data sheet (**Supplemental Table S1**) to account for isotope impurity in fractional abundance calculation from randomized channels across experiments (**Supplemental Methods**).”*

Supplementary Information, Supplementary Methods, line 26 *“TMT-10 plex lots were #WF309595 for control AC16 replicate 1 and 2; thapsigargin treated AC16 replicate 1, and tunicamycin treated AC16 replicate 1; and #XB318561 for other samples. Isotope impurities in TMT tags can lead to up to 10% spillover to neighbor channels and decrease quantitative accuracy (**Supplemental Table S1**). We used the non-negative least square algorithm in scipy (Virtanen et al., 2020) to solve for the true channel matrix from the observed channel intensity and impurity matrix for downstream quantification.”*

Reviewer 3 Comment 10: *“There is a typo in the phrase “proteins turnover rate.” It should be “protein turnover rate.”*

Response: This has been amended in the revision. Thank you.

Reviewer 3 Comment 11: *“In Figure 5C, the confocal images of EGFR lack isotype controls. The authors should consider including isotype controls to ensure proper interpretation of the results.”*

Response: We thank the Reviewer for the comment. Where applicable, we have incorporated secondary antibody only negative control (e.g., see **Reviewer Figure R4** under our response to **Reviewer 3 Comment 4** above, and **Reviewer Figure R5** below). While we agree with the Reviewer that an isotype control can be valuable in some experiments, it is not commonly pursued for immunofluorescence microscopy for fixed cells. In the case of this EGFR immunostaining experiment, we did not suspect non-specific binding from the antibody isotype, because this is a commonly used mouse monoclonal IgG antibody (abcam ab30), the immunofluorescence patterns matched with expected subcellular localization, and the mass spectrometry

data already provided a high prior likelihood for the results.

Reviewer Figure R5. Negative control for SLC3A2 (red) against DAPI (blue) in human AC16 cells. Scale bar: 50 μm . “2° only” represents a slide incubated without primary antibody as negative control. Figures correspond to the SLC3A2 immunostaining data shown in Figure 3D–3E.

We thank the Reviewer again for the many helpful suggestions. It is our sincere belief that the manuscript has improved substantially as a result.

References (Response to Reviewer Document)

- Bresson, S., Shchepachev, V., Spanos, C., Turowski, T.W., Rappsilber, J., Tollervey, D., 2020. Stress-Induced Translation Inhibition through Rapid Displacement of Scanning Initiation Factors. *Mol. Cell* 80, 470-484.e8. <https://doi.org/10.1016/j.molcel.2020.09.021>
- Cao, L., Huang, C., Cui Zhou, D., Hu, Y., Lih, T.M., Savage, S.R., Krug, K., Clark, D.J., Schnaubelt, M., Chen, L., da Veiga Leprevost, F., Eguez, R.V., Yang, W., Pan, J., Wen, B., Dou, Y., Jiang, W., Liao, Y., Shi, Z., Terekhanova, N.V., Cao, S., Lu, R.J.-H., Li, Y., Liu, R., Zhu, H., Ronning, P., Wu, Y., Wyczalkowski, M.A., Easwaran, H., Danilova, L., Mer, A.S., Yoo, S., Wang, J.M., Liu, W., Haibe-Kains, B., Thiagarajan, M., Jewell, S.D., Hostetter, G., Newton, C.J., Li, Q.K., Roehrl, M.H., Fenyö, D., Wang, P., Nesvizhskii, A.I., Mani, D.R., Omenn, G.S., Boja, E.S., Mesri, M., Robles, A.I., Rodriguez, H., Bathe, O.F., Chan, D.W., Hruban, R.H., Ding, L., Zhang, B., Zhang, H., Clinical Proteomic Tumor Analysis Consortium, 2021. Proteogenomic characterization of pancreatic ductal adenocarcinoma. *Cell* 184, 5031-5052.e26. <https://doi.org/10.1016/j.cell.2021.08.023>
- Chen, J., Xue, R., Li, Li, Xiao, L.L., Shangguan, J., Zhang, W., Bai, X., Liu, G., Li, Ling, 2019. Panax Notoginseng Saponins Protect Cardiac Myocytes Against Endoplasmic Reticulum Stress and Associated Apoptosis Through Mediation of Intracellular Calcium Homeostasis. *Front. Pharmacol.* 10.
- Christoforou, A., Mulvey, C.M., Breckels, L.M., Geladaki, A., Hurrell, T., Hayward, P.C., Naake, T., Gatto, L., Viner, R., Arias, A.M., Lilley, K.S., 2016. A draft map of the mouse pluripotent stem cell spatial proteome. *Nat. Commun.* 7, 9992. <https://doi.org/10.1038/ncomms9992>
- Clark, D.J., Dhanasekaran, S.M., Petralia, F., Pan, J., Song, X., Hu, Y., da Veiga Leprevost, F., Reva, B., Lih, T.-S.M., Chang, H.-Y., Ma, W., Huang, C., Ricketts, C.J., Chen, L., Krek, A., Li, Y., Rykunov, D., Li, Q.K., Chen, L.S., Ozbek, U., Vasaiakar, S., Wu, Y., Yoo, S., Chowdhury, S., Wyczalkowski, M.A., Ji, J., Schnaubelt, M., Kong, A., Sethuraman, S., Avtonomov, D.M., Ao, M., Colaprico, A., Cao, S., Cho, K.-C., Kalayci, S., Ma, S., Liu, W., Ruggles, K., Calinawan, A., Gümüş, Z.H., Geiszler, D., Kawaler, E., Teo, G.C., Wen, B., Zhang, Y., Keegan, S., Li, K., Chen, F., Edwards, N., Pierorazio, P.M., Chen, X.S., Pavlovich, C.P., Hakimi, A.A., Brominski, G., Hsieh, J.J., Antczak, A., Omelchenko, T., Lubinski, J., Wiznerowicz, M., Linehan, W.M., Kinsinger, C.R., Thiagarajan, M., Boja, E.S., Mesri, M., Hiltke, T., Robles, A.I., Rodriguez, H., Qian, J., Fenyö, D., Zhang, B., Ding, L., Schadt, E., Chinnaiyan, A.M., Zhang, Z., Omenn, G.S., Cieslik, M., Chan, D.W., Nesvizhskii, A.I., Wang, P., Zhang, H., Clinical Proteomic Tumor Analysis Consortium, 2019. Integrated Proteogenomic Characterization of Clear Cell Renal Cell Carcinoma. *Cell* 179, 964-983.e31. <https://doi.org/10.1016/j.cell.2019.10.007>
- Claydon, A.J., Beynon, R., 2012. Proteome dynamics: revisiting turnover with a global perspective. *Mol. Cell. Proteomics MCP* 11, 1551-1565. <https://doi.org/10.1074/mcp.O112.022186>
- Crook, O.M., Davies, C.T.R., Breckels, L.M., Christopher, J.A., Gatto, L., Kirk, P.D.W., Lilley, K.S., 2022. Inferring differential subcellular localisation in comparative spatial proteomics using BUNDLE. *Nat. Commun.* 13, 5948. <https://doi.org/10.1038/s41467-022-33570-9>
- Deberneh, H.M., Abdelrahman, D.R., Verma, S.K., Linares, J.J., Murton, A.J., Russell, W.K., Kuyumcu-Martinez, M.N., Miller, B.F., Sadygov, R.G., 2023. Quantifying label enrichment from two mass isotopomers increases proteome coverage for in vivo protein turnover using heavy water metabolic labeling. *Commun. Chem.* 6, 72. <https://doi.org/10.1038/s42004-023-00873-x>
- Dou, Y., Kawaler, E.A., Cui Zhou, D., Gritsenko, M.A., Huang, C., Blumenberg, L., Karpova, A., Petyuk, V.A., Savage, S.R., Satpathy, S., Liu, W., Wu, Y., Tsai, C.-F., Wen, B., Li, Z., Cao, S., Moon, J., Shi, Z., Cornwell, M., Wyczalkowski, M.A., Chu, R.K., Vasaiakar, S., Zhou, H., Gao, Q., Moore, R.J., Li, K., Sethuraman, S., Monroe, M.E., Zhao, R., Heiman, D., Krug, K., Clauser, K., Kothadia, R., Maruvka, Y., Pico, A.R., Oliphant, A.E., Hoskins, E.L., Pugh, S.L., Beecroft, S.J.I., Adams, D.W., Jarman, J.C., Kong, A., Chang, H.-Y., Reva, B., Liao, Y., Rykunov, D., Colaprico, A., Chen, X.S., Czekański, A., Jędryka, M., Matkowski, R., Wiznerowicz, M., Hiltke, T., Boja, E., Kinsinger, C.R., Mesri, M., Robles, A.I., Rodriguez, H., Mutch, D., Fuh, K., Ellis, M.J., DeLair, D., Thiagarajan, M., Mani, D.R., Getz, G., Noble, M., Nesvizhskii, A.I., Wang, P., Anderson, M.L., Levine, D.A., Smith, R.D., Payne, S.H., Ruggles, K.V., Rodland, K.D., Ding, L., Zhang, B., Liu, T., Fenyö, D., Agarwal, A., Anurag, M., Avtonomov, D., Birger, C., Birrer, M.J., Boca, S.M., Bocik, W.E., Borate, U., Borucki, M., Burke, M.C., Cai, S., Calinawan, A., Carr, S.A., Carter, S., Castro, P., Cerda, S., Chaikin, M., Chan, D.W., Chan, D., Charamut, A., Chen, F., Chen, J., Chen, L., Chen, L.S., Chesla, D., Chheda, M.G., Chinnaiyan, A.M., Chowdhury, S., Cieslik, M.P., Clark, D.J., Cottingham, S., Culpepper, H., Day, J., De Young, S., Demir, E., Dhanasekaran, S.M., Dhir, R., Domagalski, M.J., Dottino, P., Druker, B., Duffy, E., Dyer, M., Edwards,

- N.J., Edwards, R., Elburn, K., Field, J.B., Francis, A., Gabriel, S., Geffen, Y., Geiszler, D., Gillette, M.A., Godwin, A.K., Grady, P., Hannick, L., Hariharan, P., Hilsenbeck, S., Hindenach, B., Hoadley, K.A., Hong, R., Hostetter, G., Hsieh, J.J., Hu, Y., Ittmann, M.M., Jaehnig, E., Jewell, S.D., Ji, J., Jones, C.D., Karabon, R., Ketchum, K.A., Khan, M., Kim, B.-J., Krek, A., Krubit, T., Kumar-Sinha, C., Leprevost, F.D., Lewis, M., Li, Q.K., Li, Y., Liu, H., Lubinski, J., Ma, W., Madan, R., Malc, E., Malovannaya, A., Mareedu, S., Markey, S.P., Marrero-Oliveras, A., Martignetti, J., McDermott, J., McGarvey, P.B., McGee, J., Mieczkowski, P., Modugno, F., Montgomery, R., Newton, C.J., Omenn, G.S., Paulovich, A.G., Perou, A.M., Petralia, F., Piehowski, P., Polonskaya, L., Qi, L., Richey, S., Robinson, K., Roche, N., Rohrer, D.C., Schadt, E.E., Schnaubelt, M., Shi, Y., Skelly, T., Sokoll, L.J., Song, X., Stein, S.E., Suh, J., Tan, D., Tansil, D., Teo, G.C., Thangudu, R.R., Tognon, C., Traer, E., Tyner, J., Um, K.S., Valley, D.R., Vatanian, N., Vats, P., Velvulou, U., Vernon, M., Wang, L.-B., Wang, Y., Webster, A., Westbrook, T., Wheeler, D., Whiteaker, J.R., Wilson, G.D., Zakhartsev, Y., Zelt, R., Zhang, H., Zhang, Y., Zhang, Z., Zhao, G., 2020. Proteogenomic Characterization of Endometrial Carcinoma. *Cell* 180, 729-748.e26. <https://doi.org/10.1016/j.cell.2020.01.026>
- Dunkley, T.P.J., Watson, R., Griffin, J.L., Dupree, P., Lilley, K.S., 2004. Localization of Organelle Proteins by Isotope Tagging (LOPIT). *Mol. Cell. Proteomics* 3, 1128–1134. <https://doi.org/10.1074/mcp.T400009-MCP200>
- Efentakis, P., Psarakou, G., Varela, A., Papanagnou, E.D., Chatzistefanou, M., Nikolaou, P.-E., Davos, C.H., Gavriatopoulou, M., Trougakos, I.P., Dimopoulos, M.A., Andreadou, I., Terpos, E., 2021. Elucidating Carfilzomib's Induced Cardiotoxicity in an In Vivo Model of Aging: Prophylactic Potential of Metformin. *Int. J. Mol. Sci.* 22, 10956. <https://doi.org/10.3390/ijms222010956>
- Forghani, P., Rashid, A., Sun, F., Liu, R., Li, D., Lee, M.R., Hwang, H., Maxwell, J.T., Mandawat, A., Wu, R., Salaita, K., Xu, C., 2021. Carfilzomib Treatment Causes Molecular and Functional Alterations of Human Induced Pluripotent Stem Cell-Derived Cardiomyocytes. *J. Am. Heart Assoc.* 10, e022247. <https://doi.org/10.1161/JAHA.121.022247>
- Fornasiero, E.F., Mandad, S., Wildhagen, H., Alevra, M., Rammner, B., Keihani, S., Opazo, F., Urban, I., Ischebeck, T., Sakib, M.S., Fard, M.K., Kirli, K., Centeno, T.P., Vidal, R.O., Rahman, R.-U., Benito, E., Fischer, A., Dennerlein, S., Rehling, P., Feussner, I., Bonn, S., Simons, M., Urlaub, H., Rizzoli, S.O., 2018. Precisely measured protein lifetimes in the mouse brain reveal differences across tissues and subcellular fractions. *Nat. Commun.* 9, 4230. <https://doi.org/10.1038/s41467-018-06519-0>
- Foster, L.J., de Hoog, C.L., Zhang, Yanling, Zhang, Yong, Xie, X., Mootha, V.K., Mann, M., 2006. A mammalian organelle map by protein correlation profiling. *Cell* 125, 187–199. <https://doi.org/10.1016/j.cell.2006.03.022>
- Fridolfsson, H.N., Kawaraguchi, Y., Ali, S.S., Panneerselvam, M., Niesman, I.R., Finley, J.C., Kellerhals, S.E., Migita, M.Y., Okada, H., Moreno, A.L., Jennings, M., Kidd, M.W., Bonds, J.A., Balijepalli, R.C., Ross, R.S., Patel, P.M., Miyanohara, A., Chen, Q., Lesnefsky, E.J., Head, B.P., Roth, D.M., Insel, P.A., Patel, H.H., 2012. Mitochondria-localized caveolin in adaptation to cellular stress and injury. *FASEB J. Off. Publ. Fed. Am. Soc. Exp. Biol.* 26, 4637–4649. <https://doi.org/10.1096/fj.12-215798>
- Geladaki, A., Kočevár Britovšek, N., Breckels, L.M., Smith, T.S., Vennard, O.L., Mulvey, C.M., Crook, O.M., Gatto, L., Lilley, K.S., 2019. Combining LOPIT with differential ultracentrifugation for high-resolution spatial proteomics. *Nat. Commun.* 10, 331. <https://doi.org/10.1038/s41467-018-08191-w>
- Ghosh, S., Villacorta-Martin, C., Lindstrom-Vautrin, J., Kenney, D., Golden, C.S., Edwards, C.V., Sanchowala, V., Connors, L.H., Giadone, R.M., Murphy, G.J., 2023. Mapping cellular response to destabilized transthyretin reveals cell- and amyloidogenic protein-specific signatures. *Amyloid* 0, 1–15. <https://doi.org/10.1080/13506129.2023.2224494>
- Gillette, M.A., Satpathy, S., Cao, S., Dhanasekaran, S.M., Vasaikar, S.V., Krug, K., Petralia, F., Li, Y., Liang, W.-W., Reva, B., Krek, A., Ji, J., Song, X., Liu, W., Hong, R., Yao, L., Blumenberg, L., Savage, S.R., Wendl, M.C., Wen, B., Li, K., Tang, L.C., MacMullan, M.A., Avanesian, S.C., Kane, M.H., Newton, C.J., Cornwell, M., Kothadia, R.B., Ma, W., Yoo, S., Mannan, R., Vats, P., Kumar-Sinha, C., Kawaler, E.A., Omelchenko, T., Colaprico, A., Geffen, Y., Maruvka, Y.E., da Veiga Leprevost, F., Wiznerowicz, M., Gümüş, Z.H., Veluswamy, R.R., Hostetter, G., Heiman, D.I., Wyczalkowski, M.A., Hiltke, T., Mesri, M., Kinsinger, C.R., Boja, E.S., Omenn, G.S., Chinnaiyan, A.M., Rodriguez, H., Li, Q.K., Jewell, S.D., Thiagarajan, M., Getz, G., Zhang, B., Fenyö, D., Ruggles, K.V., Cieslik, M.P., Robles, A.I., Clauser, K.R., Govindan, R., Wang, P., Nesvizhskii, A.I., Ding, L., Mani, D.R., Carr, S.A., Clinical Proteomic Tumor Analysis Consortium, 2020. Proteogenomic Characterization Reveals Therapeutic Vulnerabilities in Lung Adenocarcinoma. *Cell* 182, 200-225.e35.

- <https://doi.org/10.1016/j.cell.2020.06.013>
- Glembotski, C.C., 2007. Endoplasmic Reticulum Stress in the Heart. *Circ. Res.* 101, 975–984.
<https://doi.org/10.1161/CIRCRESAHA.107.161273>
- Guan, B.-J., van Hoef, V., Jobava, R., Elroy-Stein, O., Valasek, L.S., Cargnello, M., Gao, X.-H., Krokowski, D., Merrick, W.C., Kimball, S.R., Komar, A.A., Koromilas, A.E., Wynshaw-Boris, A., Topisirovic, I., Larsson, O., Hatzoglou, M., 2017. A Unique ISR Program Determines Cellular Responses to Chronic Stress. *Mol. Cell* 68, 885-900.e6. <https://doi.org/10.1016/j.molcel.2017.11.007>
- Gupta, N., Pevzner, P.A., 2009. False discovery rates of protein identifications: a strike against the two-peptide rule. *J. Proteome Res.* 8, 4173–4181. <https://doi.org/10.1021/pr9004794>
- Hammond, D.E., Simpson, D.M., Franco, C., Wright Muelas, M., Waters, J., Ludwig, R.W., Prescott, M.C., Hurst, J.L., Beynon, R.J., Lau, E., 2022. Harmonizing Labeling and Analytical Strategies to Obtain Protein Turnover Rates in Intact Adult Animals. *Mol. Cell. Proteomics* 21, 100252.
<https://doi.org/10.1016/j.mcpro.2022.100252>
- Hanafusa, K., Wada, I., Hosokawa, N., 2019. SDF2-like protein 1 (SDF2L1) regulates the endoplasmic reticulum localization and chaperone activity of ERdj3 protein. *J. Biol. Chem.* 294, 19335–19348.
<https://doi.org/10.1074/jbc.RA119.009603>
- Jean Beltran, P.M., Mathias, R.A., Cristea, I.M., 2016. A Portrait of the Human Organelle Proteome In Space and Time during Cytomegalovirus Infection. *Cell Syst.* 3, 361-373.e6.
<https://doi.org/10.1016/j.cels.2016.08.012>
- Kim, T.-Y., Wang, D., Kim, A.K., Lau, E., Lin, A.J., Liem, D.A., Zhang, J., Zong, N.C., Lam, M.P.Y., Ping, P., 2012. Metabolic labeling reveals proteome dynamics of mouse mitochondria. *Mol. Cell. Proteomics MCP* 11, 1586–1594. <https://doi.org/10.1074/mcp.M112.021162>
- Kim, Y., Li, C., Gu, C., Fang, Y., Tycksen, E., Puri, A., Pietka, T.A., Sivapackiam, J., Kidd, K., Park, S.-J., Johnson, B.G., Kmoch, S., Duffield, J.S., Bleyer, A.J., Jackrel, M.E., Urano, F., Sharma, V., Lindahl, M., Chen, Y.M., 2023. MANF stimulates autophagy and restores mitochondrial homeostasis to treat autosomal dominant tubulointerstitial kidney disease in mice. *Nat. Commun.* 14, 6493.
<https://doi.org/10.1038/s41467-023-42154-0>
- Krug, K., Jaehnig, E.J., Satpathy, S., Blumenberg, L., Karpova, A., Anurag, M., Miles, G., Mertins, P., Geffen, Y., Tang, L.C., Heiman, D.I., Cao, S., Maruvka, Y.E., Lei, J.T., Huang, C., Kothadia, R.B., Colaprico, A., Birger, C., Wang, J., Dou, Y., Wen, B., Shi, Z., Liao, Y., Wiznerowicz, M., Wyczalkowski, M.A., Chen, X.S., Kennedy, J.J., Paulovich, A.G., Thiagarajan, M., Kinsinger, C.R., Hiltke, T., Boja, E.S., Mesri, M., Robles, A.I., Rodriguez, H., Westbrook, T.F., Ding, L., Getz, G., Clauser, K.R., Fenyö, D., Ruggles, K.V., Zhang, B., Mani, D.R., Carr, S.A., Ellis, M.J., Gillette, M.A., Clinical Proteomic Tumor Analysis Consortium, 2020. Proteogenomic Landscape of Breast Cancer Tumorigenesis and Targeted Therapy. *Cell* 183, 1436-1456.e31. <https://doi.org/10.1016/j.cell.2020.10.036>
- Lam, M.P.Y., Wang, D., Lau, E., Liem, D.A., Kim, A.K., Ng, D.C.M., Liang, X., Bleakley, B.J., Liu, C., Tabaraki, J.D., Cadeiras, M., Wang, Y., Deng, M.C., Ping, P., 2014. Protein kinetic signatures of the remodeling heart following isoproterenol stimulation. *J. Clin. Invest.* 124, 1734–1744.
<https://doi.org/10.1172/JCI73787>
- Lamper, A.M., Fleming, R.H., Ladd, K.M., Lee, A.S.Y., 2020. A phosphorylation-regulated eIF3d translation switch mediates cellular adaptation to metabolic stress. *Science* 370, 853–856.
<https://doi.org/10.1126/science.abb0993>
- Lau, E., Cao, Q., Lam, M.P.Y., Wang, J., Ng, D.C.M., Bleakley, B.J., Lee, J.M., Liem, D.A., Wang, D., Hermjakob, H., Ping, P., 2018. Integrated omics dissection of proteome dynamics during cardiac remodeling. *Nat. Commun.* 9, 120. <https://doi.org/10.1038/s41467-017-02467-3>
- Lau, E., Cao, Q., Ng, D.C.M., Bleakley, B.J., Dincer, T.U., Bot, B.M., Wang, D., Liem, D.A., Lam, M.P.Y., Ge, J., Ping, P., 2016. A large dataset of protein dynamics in the mammalian heart proteome. *Sci. Data* 3, 160015. <https://doi.org/10.1038/sdata.2016.15>
- Li, G.-W., Burkhardt, D., Gross, C., Weissman, J.S., 2014. Quantifying absolute protein synthesis rates reveals principles underlying allocation of cellular resources. *Cell* 157, 624–635.
<https://doi.org/10.1016/j.cell.2014.02.033>
- Liu, C.-L., Li, X., Hu, G.-L., Li, R.-J., He, Y.-Y., Zhong, W., Li, S., He, K.-L., Wang, L.-L., 2012. Salubrinal protects against tunicamycin and hypoxia induced cardiomyocyte apoptosis via the PERK-eIF2 α signaling pathway. *J. Geriatr. Cardiol. JGC* 9, 258–268. <https://doi.org/10.3724/SP.J.1263.2012.02292>
- Lundberg, E., Borner, G.H.H., 2019. Spatial proteomics: a powerful discovery tool for cell biology. *Nat. Rev. Mol. Cell Biol.* 20, 285–302. <https://doi.org/10.1038/s41580-018-0094-y>

- Martin, T.G., Myers, V.D., Dubey, P., Dubey, S., Perez, E., Moravec, C.S., Willis, M.S., Feldman, A.M., Kirk, J.A., 2021a. Cardiomyocyte contractile impairment in heart failure results from reduced BAG3-mediated sarcomeric protein turnover. *Nat. Commun.* 12, 2942. <https://doi.org/10.1038/s41467-021-23272-z>
- Martin, T.G., Tawfik, S., Moravec, C.S., Pak, T.R., Kirk, J.A., 2021b. BAG3 expression and sarcomere localization in the human heart are linked to HSF-1 and are differentially affected by sex and disease. *Am. J. Physiol.-Heart Circ. Physiol.* 320, H2339–H2350. <https://doi.org/10.1152/ajpheart.00419.2020>
- Martínez-Martínez, E., Ibarrola, J., Fernández-Celis, A., Santamaria, E., Fernández-Irigoyen, J., Rossignol, P., Jaisser, F., López-Andrés, N., 2017. Differential Proteomics Identifies Reticulocalbin-3 as a Novel Negative Mediator of Collagen Production in Human Cardiac Fibroblasts. *Sci. Rep.* 7, 12192. <https://doi.org/10.1038/s41598-017-12305-7>
- Mathieson, T., Franken, H., Kosinski, J., Kurzawa, N., Zinn, N., Sweetman, G., Poeckel, D., Ratnu, V.S., Schramm, M., Becher, I., Steidel, M., Noh, K.-M., Bergamini, G., Beck, M., Bantscheff, M., Savitski, M.M., 2018. Systematic analysis of protein turnover in primary cells. *Nat. Commun.* 9, 689. <https://doi.org/10.1038/s41467-018-03106-1>
- Mulvey, C.M., Breckels, L.M., Crook, O.M., Sanders, D.J., Ribeiro, A.L.R., Geladaki, A., Christoforou, A., Britovšek, N.K., Hurrell, T., Deery, M.J., Gatto, L., Smith, A.M., Lilley, K.S., 2021. Spatiotemporal proteomic profiling of the pro-inflammatory response to lipopolysaccharide in the THP-1 human leukaemia cell line. *Nat. Commun.* 12, 5773. <https://doi.org/10.1038/s41467-021-26000-9>
- Naylor, B.C., Anderson, C.N.K., Hadfield, M., Parkinson, D.H., Ahlstrom, A., Hannemann, A., Quilling, C.R., Cutler, K.J., Denton, R.L., Adamson, R., Angel, T.E., Burlett, R.S., Hafen, P.S., Dallon, John.C., Transtrum, M.K., Hyldahl, R.D., Price, J.C., 2022. Utilizing Nonequilibrium Isotope Enrichments to Dramatically Increase Turnover Measurement Ranges in Single Biopsy Samples from Humans. *J. Proteome Res.* 21, 2703–2714. <https://doi.org/10.1021/acs.jproteome.2c00380>
- Pakos-Zebrucka, K., Koryga, I., Mnich, K., Lujic, M., Samali, A., Gorman, A.M., 2016. The integrated stress response. *EMBO Rep.* 17, 1374–1395. <https://doi.org/10.15252/embr.201642195>
- Palomer, X., Capdevila-Busquets, E., Botteri, G., Salvadó, L., Barroso, E., Davidson, M.M., Michalik, L., Wahli, W., Vázquez-Carrera, M., 2014. PPAR β/δ attenuates palmitate-induced endoplasmic reticulum stress and induces autophagic markers in human cardiac cells. *Int. J. Cardiol.* 174, 110–118. <https://doi.org/10.1016/j.ijcard.2014.03.176>
- Sanford, J.A., Nogiec, C.D., Lindholm, M.E., Adkins, J.N., Amar, D., Dasari, S., Drugan, J.K., Fernández, F.M., Radom-Aizik, S., Schenk, S., Snyder, M.P., Tracy, R.P., Vanderboom, P., Trappe, S., Walsh, M.J., Adkins, J.N., Amar, D., Dasari, S., Drugan, J.K., Evans, C.R., Fernandez, F.M., Li, Y., Lindholm, M.E., Nogiec, C.D., Radom-Aizik, S., Sanford, J.A., Schenk, S., Snyder, M.P., Tomlinson, L., Tracy, R.P., Trappe, S., Vanderboom, P., Walsh, M.J., Lee Alekel, D., Bekirov, I., Boyce, A.T., Boyington, J., Fleg, J.L., Joseph, L.J.O., Laughlin, M.R., Maruvada, P., Morris, S.A., McGowan, J.A., Nierras, C., Pai, V., Peterson, C., Ramos, E., Roary, M.C., Williams, J.P., Xia, A., Cornell, E., Rooney, J., Miller, M.E., Ambrosius, W.T., Rushing, S., Stowe, C.L., Jack Rejeski, W., Nicklas, B.J., Pahor, M., Lu, C., Trappe, T., Chambers, T., Raue, U., Lester, B., Bergman, B.C., Bessesen, D.H., Jankowski, C.M., Kohrt, W.M., Melanson, E.L., Moreau, K.L., Schauer, I.E., Schwartz, R.S., Kraus, W.E., Slentz, C.A., Huffman, K.M., Johnson, J.L., Willis, L.H., Kelly, L., Houmard, J.A., Dubis, G., Broskey, N., Goodpaster, B.H., Sparks, L.M., Coen, P.M., Cooper, D.M., Haddad, F., Rankinen, T., Ravussin, E., Johannsen, N., Harris, M., Jakicic, J.M., Newman, A.B., Forman, D.D., Kershaw, E., Rogers, R.J., Nindl, B.C., Page, L.C., Stefanovic-Racic, M., Barr, S.L., Rasmussen, B.B., Moro, T., Paddon-Jones, D., Volpi, E., Spratt, H., Musi, N., Espinoza, S., Patel, D., Serra, M., Gelfond, J., Burns, A., Bamman, M.M., Buford, T.W., Cutter, G.R., Bodine, S.C., Esser, K., Farrar, R.P., Goodyear, L.J., Hirshman, M.F., Albertson, B.G., Qian, W.-J., Piehowski, P., Gritsenko, M.A., Monore, M.E., Petyuk, V.A., McDermott, J.E., Hansen, J.N., Hutchison, C., Moore, S., Gaul, D.A., Clish, C.B., Avila-Pacheco, J., Dennis, C., Kellis, M., Carr, S., Jean-Beltran, P.M., Keshishian, H., Mani, D.R., Clauser, K., Krug, K., Mundorff, C., Pearce, C., Ivanova, A.A., Ortlund, E.A., Maner-Smith, K., Uppal, K., Zhang, T., Sealfon, S.C., Zaslavsky, E., Nair, V., Li, S., Jain, N., Ge, Y., Sun, Y., Nudelman, G., Ruf-zamojski, F., Smith, G., Pincas, N., Rubenstein, A., Anne Amper, M., Seenarine, N., Lappalainen, T., Lanza, I.R., Sreekumaran Nair, K., Klaus, K., Montgomery, S.B., Smith, K.S., Gay, N.R., Zhao, B., Hung, C.-J., Zebarjadi, N., Balliu, B., Fresard, L., Burant, C.F., Li, J.Z., Kachman, M., Soni, T., Raskind, A.B., Gerszten, R., Robbins, J., Ilkayeva, O., Muehlbauer, M.J., Newgard, C.B., Ashley, E.A., Wheeler, M.T., Jimenez-Morales, D., Raja, A., Dalton, K.P., Zhen, J., Suk Kim, Y., Christle, J.W., Marwaha, S., Chin, E.T., Hershman, S.G., Hastie, T., Tibshirani, R., Rivas, M.A., 2020. Molecular Transducers of Physical Activity Consortium (MoTrPAC):

- Mapping the Dynamic Responses to Exercise. *Cell* 181, 1464–1474. <https://doi.org/10.1016/j.cell.2020.06.004>
- Sayers, E.W., Bolton, E.E., Brister, J.R., Canese, K., Chan, J., Comeau, D.C., Connor, R., Funk, K., Kelly, C., Kim, S., Madej, T., Marchler-Bauer, A., Lanczycki, C., Lathrop, S., Lu, Z., Thibaud-Nissen, F., Murphy, T., Phan, L., Skripchenko, Y., Tse, T., Wang, J., Williams, R., Trawick, B.W., Pruitt, K.D., Sherry, S.T., 2022. Database resources of the national center for biotechnology information. *Nucleic Acids Res.* 50, D20–D26. <https://doi.org/10.1093/nar/gkab1112>
- Searle, B.C., Yergey, A.L., 2020. An efficient solution for resolving iTRAQ and TMT channel cross-talk. *J. Mass Spectrom.* 55, e4354. <https://doi.org/10.1002/jms.4354>
- Shin, J.J.H., Crook, O.M., Borgeaud, A.C., Cattin-Ortolá, J., Peak-Chew, S.Y., Breckels, L.M., Gillingham, A.K., Chadwick, J., Lilley, K.S., Munro, S., 2020. Spatial proteomics defines the content of trafficking vesicles captured by golgin tethers. *Nat. Commun.* 11, 5987. <https://doi.org/10.1038/s41467-020-19840-4>
- Siltanen, A., Nuutila, K., Imanishi, Y., Uenaka, H., Mäkelä, J., Pättilä, T., Vento, A., Miyagawa, S., Sawa, Y., Harjula, A., Kankuri, E., 2016. The Paracrine Effect of Skeletal Myoblasts is Cardioprotective against Oxidative Stress and Involves EGFR-ErbB4 Signaling, Cystathionase, and the Unfolded Protein Response. *Cell Transplant.* 25, 55–69. <https://doi.org/10.3727/096368915X688254>
- Stoner, M.W., McTiernan, C.F., Scott, I., Manning, J.R., 2020. Calreticulin expression in human cardiac myocytes induces ER stress-associated apoptosis. *Physiol. Rep.* 8, e14400. <https://doi.org/10.14814/phy2.14400>
- Tan, X., Thapa, N., Sun, Y., Anderson, R.A., 2015. A Kinase-Independent Role for EGF Receptor in Autophagy Initiation. *Cell* 160, 145–160. <https://doi.org/10.1016/j.cell.2014.12.006>
- Taylor, S., Gupta, B.P., 2023. The neurotrophic factor MANF regulates autophagic flux and lysosome function to promote proteostasis in *C. elegans* (preprint). *Genetics*. <https://doi.org/10.1101/2023.07.31.551399>
- Thul, P.J., Åkesson, L., Wiking, M., Mahdessian, D., Geladaki, A., Ait Blal, H., Alm, T., Asplund, A., Björk, L., Breckels, L.M., Bäckström, A., Danielsson, F., Fagerberg, L., Fall, J., Gatto, L., Gnann, C., Hober, S., Hjelmare, M., Johansson, F., Lee, S., Lindskog, C., Mulder, J., Mulvey, C.M., Nilsson, P., Oksvold, P., Rockberg, J., Schutten, R., Schwenk, J.M., Sivertsson, Å., Sjöstedt, E., Skogs, M., Stadler, C., Sullivan, D.P., Tegel, H., Winsnes, C., Zhang, C., Zwahlen, M., Mardinoglu, A., Pontén, F., von Feilitzen, K., Lilley, K.S., Uhlén, M., Lundberg, E., 2017. A subcellular map of the human proteome. *Science* 356, eaal3321. <https://doi.org/10.1126/science.aal3321>
- Toro, R., Pérez-Serra, A., Mangas, A., Campuzano, O., Sarquella-Brugada, G., Quezada-Feijoo, M., Ramos, M., Alcalá, M., Carrera, E., García-Padilla, C., Franco, D., Bonet, F., 2022. miR-16-5p Suppression Protects Human Cardiomyocytes against Endoplasmic Reticulum and Oxidative Stress-Induced Injury. *Int. J. Mol. Sci.* 23, 1036. <https://doi.org/10.3390/ijms23031036>
- Virtanen, P., Gommers, R., Oliphant, T.E., Haberland, M., Reddy, T., Cournapeau, D., Burovski, E., Peterson, P., Weckesser, W., Bright, J., van der Walt, S.J., Brett, M., Wilson, J., Millman, K.J., Mayorov, N., Nelson, A.R.J., Jones, E., Kern, R., Larson, E., Carey, C.J., Polat, İ., Feng, Y., Moore, E.W., VanderPlas, J., Laxalde, D., Perktold, J., Cimrman, R., Henriksen, I., Quintero, E.A., Harris, C.R., Archibald, A.M., Ribeiro, A.H., Pedregosa, F., van Mulbregt, P., SciPy 1.0 Contributors, 2020. SciPy 1.0: fundamental algorithms for scientific computing in Python. *Nat. Methods* 17, 261–272. <https://doi.org/10.1038/s41592-019-0686-2>
- Volonte, D., Liu, Z., Shiva, S., Galbiati, F., 2016. Caveolin-1 controls mitochondrial function through regulation of m-AAA mitochondrial protease. *Aging* 8, 2355–2369. <https://doi.org/10.18632/aging.101051>
- Wang, L.-B., Karpova, A., Gritsenko, M.A., Kyle, J.E., Cao, S., Li, Y., Rykunov, D., Colaprico, A., Rothstein, J.H., Hong, R., Stathias, V., Cornwell, M., Petralia, F., Wu, Y., Reva, B., Krug, K., Pugliese, P., Kawaler, E., Olsen, L.K., Liang, W.-W., Song, X., Dou, Y., Wendl, M.C., Caravan, W., Liu, W., Cui Zhou, D., Ji, J., Tsai, C.-F., Petyuk, V.A., Moon, J., Ma, W., Chu, R.K., Weitz, K.K., Moore, R.J., Monroe, M.E., Zhao, R., Yang, X., Yoo, S., Krek, A., Demopoulos, A., Zhu, H., Wyczalkowski, M.A., McMichael, J.F., Henderson, B.L., Lindgren, C.M., Boekweg, H., Lu, S., Baral, J., Yao, L., Stratton, K.G., Bramer, L.M., Zink, E., Couvillion, S.P., Bloodsworth, K.J., Satpathy, S., Sieh, W., Boca, S.M., Schürer, S., Chen, F., Wiznerowicz, M., Ketchum, K.A., Boja, E.S., Kinsinger, C.R., Robles, A.I., Hiltke, T., Thiagarajan, M., Nesvizhskii, A.I., Zhang, B., Mani, D.R., Ceccarelli, M., Chen, X.S., Cottingham, S.L., Li, Q.K., Kim, A.H., Fenyö, D., Ruggles, K.V., Rodriguez, H., Mesri, M., Payne, S.H., Resnick, A.C., Wang, P., Smith, R.D., Iavarone, A., Chheda, M.G., Barnholtz-Sloan, J.S., Rodland, K.D., Liu, T., Ding, L., Clinical Proteomic Tumor Analysis Consortium, 2021. Proteogenomic and metabolomic characterization of human glioblastoma. *Cancer Cell* 39, 509-528.e20.

<https://doi.org/10.1016/j.ccell.2021.01.006>

West, G.B., Woodruff, W.H., Brown, J.H., 2002. Allometric scaling of metabolic rate from molecules and mitochondria to cells and mammals. *Proc. Natl. Acad. Sci.* 99, 2473–2478.

<https://doi.org/10.1073/pnas.012579799>

REVIEWERS' COMMENTS

Reviewer #1 (Remarks to the Author):

The manuscript is much improved and the MS results are very convincing. The strengths of this manuscript are: the new methodology which combines two approaches which are both high in information content but technically demanding, as well as the demonstration on multiple interesting biological systems showing that the methodology is robust. Such methodology has been elusive to the community so I am grateful this has been tackled. The authors acknowledge limitations of the approach and this is warmly welcomed.

I have minor comments but do not need to review a revised manuscript:

- 1) The abstract is quite long and perhaps could benefit from some prioritisation.
- 2) Some switching between dynamic/dynamical - which do you really mean? (I think dynamic)
- 3) PCA plots: It is perhaps worth introducing some transparency to improve the density of the points; and also ensure the aspect ratio is more faithful to the variance explained in each component.
- 4) (Likely out of scope) It would be extremely interesting if translocating proteins had different conformations, especially if there are heavy and light localisation differences. The experimental methodology here is weak but some hints could be examined using computational approaches (<https://www.biorxiv.org/content/10.1101/2022.10.17.512570v1.full>). I wonder if the authors had thought about this, but I do not expect them to examine this here.

Reviewer #2 (Remarks to the Author):

Comments

The authors have carried out extensive additional work to address the points carefully. This has improved the manuscript a lot. I would not ask for any additional major points to address experimentally. It would be important though to clarify a couple of remaining points:

The authors argue (reviewer 2 comment 2) that they would not want to compare with Prot K digested samples as they also want to be able to see organelle-associated proteins. I can agree with this point. Consequently, it would be important to word throughout the text more clearly that the data (and the use of the word "translocation") does not indicate that there has been an actual import of proteins into the organelles or their membranes. For this, it would be critical to compare whether proteins are attached to or inserted into the organelle, typically done by digestion of associated factors.

I agree with the authors that LOPIT can generally deal with changes in the shape of an organelle,

based on colocalization with markers. For ER stress, subcompartmentalization within the ER has been described, e.g. by forming tubular structures, which may not colocalize with the used marker proteins anymore. Can this be identified by the described approach? If not, it would be useful to discuss this, e.g. in the limitations section.

I may have overlooked, please specify whether there was a cut-off for minimal number of peptides required to include the data.

Side point:

I do not agree with the suggestion that the extensive fractionation and small isolation window mitigate the issues with MS2. However, this is not an issue that should prevent publication of the data in the manuscript. For future consideration, I would like to point the authors to their new supp fig S4B and the much improved separation of compartments by an MS3 method, which strongly supports that ratio compression in the MS2 data has been a large aspect in blurring the compartment boundaries.

Reviewer #4 (Remarks to the Author):

The authors have convincingly addressed the points raised by Reviewer 3.

I have, however, a technical suggestion: the methods are distributed across the Results section, the methods section of the main paper and the supplementary methods. This makes it difficult for the interested reader to find specific technical details.

More specifically, all the MS analysis pipeline steps should be explained in the methods section of the main text. Some specific details from there could be referenced to the supp methods. All the new developments performed in this work in relation to previous ones should be described in the methods of the main text.

Then the authors can comment on these methods from a more general viewpoint in the Results section. For more clarity, this is not a good idea to use the Results as a substitute of the methods. Also:

-what are the changes in the new version of pyTMT tool?

-what does the SPLAT pipeline specifically contain? i.e. RIANNA is included in the SPLAT or used separately? Is SPLAT a software package or a name for the whole technique including separate algorithms? Please explain better

-the isoform-aware rollup of quantitative peptide info into protein info is not fully clear and needs a more elaborate explanation (in the methods section).

-peptide identification is performed using Percolator at 5%FDR, but in another place of the text it is said FDR is 1%, please correct.

-how does the author calculate protein FDR after peptide rollup to proteins?

Response to Reviewers

Reviewer #1 (Remarks to the Author): “The manuscript is much improved and the MS results are very convincing. The strengths of this manuscript are: the new methodology which combines two approaches which are both high in information content but technically demanding, as well as the demonstration on multiple interesting biological systems showing that the methodology is robust. Such methodology has been elusive to the community so I am grateful this has been tackled. The authors acknowledge limitations of the approach and this is warmly welcomed. I have minor comments but do not need to review a revised manuscript:”

Response: We thank Reviewer 1 again for the encouraging comments.

Reviewer 1 Comment 1: *“The abstract is quite long and perhaps could benefit from some prioritisation.”*

Response: We have now shortened the abstract to under 200 words.

Reviewer 1 Comment 2: *“Some switching between dynamic/dynamical - which do you really mean? (I think dynamic)”*

Response: We have now standardized usage to “dynamic”.

Reviewer 1 Comment 3: *“PCA plots: It is perhaps worth introducing some transparency to improve the density of the points; and also ensure the aspect ratio is more faithful to the variance explained in each component.”*

Response: We appreciate Reviewer 1’s suggestion. We have now reduced the point size to improve density. We would like to refrain from introducing transparency out of concern that it will affect color perception. Although we fully agree with Reviewer 1 that the aspect ratio of PCA plots should ideally scale to each PC’s variance explained, in practice we worry this will lead to very many panels with different shapes. Instead, we have made another effort to double check that the variance explained by a PC is clearly labeled for each axis in all PCA plots.

Reviewer 1 Comment 4: *“(Likely out of scope) It would be extremely interesting if translocating proteins had different conformations, especially if there are heavy and light localisation differences. The experimental methodology here is weak but some hints could be examined using computational approaches (<https://www.biorxiv.org/content/10.1101/2022.10.17.512570v1.full>). I wonder if the authors had thought about this, but I do not expect them to examine this here.”*

Response: We appreciate the Reviewer’s insight and for bringing to our attention an interesting article. While we are unable to address this question within the scope of the present work, we would be excited to explore this issue in the future.

Reviewer #2 (Remarks to the Author): *“The authors have carried out extensive additional work to address the points carefully. This has improved the manuscript a lot. I would not ask for any additional major points to address experimentally. It would be important though to clarify a couple of remaining points:”*

Response: We thank Reviewer 2 again for their many useful suggestions, which have helped us tremendously in improving the manuscript. We have included our responses to the remaining points below.

Reviewer 2 Comment 1: *“The authors argue (reviewer 2 comment 2) that they would not want to compare with Prot K digested samples as they also want to be able to see organelle-associated proteins. I can agree with this point. Consequently, it would be important to word throughout the text more clearly that the data (and the use of the word “translocation”) does not indicate that there has been an actual import of proteins into the organelles or their membranes. For this, it would be critical to compare whether proteins are attached to or inserted into the organelle, typically done by digestion of associated factors.”*

Response: We thank Reviewer 2 for raising this salient point and agree it would be important to distinguish association from insertion. In response, we have replaced the wording of “translocation” with “differential localization” or provided clarification where applicable, to avoid misunderstanding.

Reviewer 2 Comment 2: *“I agree with the authors that LOPIT can generally deal with changes in the shape of an organelle, based on colocalization with markers. For ER stress, subcompartmentalization within the ER has been described, e.g. by forming tubular structures, which may not colocalize with the used marker proteins anymore. Can this be identified by the described approach? If not, it would be useful to discuss this, e.g. in the limitations section.”*

Response: We agree with Reviewer 2. We have reorganized the text to more thoroughly discuss additional limitations in the beginning of the Discussions section. We now mention the limitation as recommended that under stress, some organelles may sub-compartmentalize or fragment, which is currently difficult to trace.

Reviewer 2 Comment 3: *“I may have overlooked, please specify whether there was a cut-off for minimal number of peptides required to include the data.”*

Response: A minimum of two peptides is required for inclusion in the data. This was mentioned in the Results section but is now more clearly specified in the Methods section.

Reviewer 2 Comment 4: *“Side point: I do not agree with the suggestion that the extensive fractionation and small isolation window mitigate the issues with MS2. However, this is not an issue that should prevent publication of the data in the manuscript. For future consideration, I would like to point the authors to their new supp fig S4B and the much improved separation of compartments by an MS3 method, which strongly supports that ratio compression in the MS2 data has been a large aspect in blurring the compartment boundaries.”*

Response: We thank Reviewer 2 for sharing further thoughts on this topic. We will strive to consider MS3-based TMT experiments in the future.

Reviewer #4 (Remarks to the Author): *“The authors have convincingly addressed the points raised by Reviewer 3.”*

Response: We thank Reviewer 4 for evaluating the manuscript and providing feedback regarding our response to the previous comments.

Reviewer 4 Comment 1: *“I have, however, a technical suggestion: the methods are distributed across the Results section, the methods section of the main paper and the supplementary methods. This makes it difficult for the interested reader to find specific technical details. More specifically, all the MS analysis pipeline steps should be explained in the methods section of the main text. Some specific details from there could be referenced to the supp methods. All the new developments performed in this work in relation to previous ones should be described in the methods of the main text. Then the authors can comment on these methods from a more general viewpoint in the Results section. For more clarity, this is not a good idea to use the Results as a substitute of the methods.”*

Response: We agree with Reviewer 4 and have now moved part of the method descriptions toward the Methods section from the Results section for a clearer distinction.

Reviewer 4 Comment 2: *“Also: -what are the changes in the new version of pyTMT tool?”*

Response: Changes to the pyTMT tool resulting from this work are (1) functionality to perform isotope contaminant correction using nnls, and (2) summing peptide intensities into proteins based on isoform roll up. During the revision we have also added compatibility with MS3-level TMT data. The changes were mentioned in the Results before, but are now more clearly stated in the Methods section.

Reviewer 4 Comment 3: *“-what does the SPLAT pipeline specifically contains? i.e. RIANNA is included in the SPLAT or used separately? Is SPLAT a software package or a name for the whole technique including separate algorithms? Please explain better.”*

Response: In response to this comment, we clarify that SPLAT is intended to refer to the overall experimental approach described in the manuscript. To facilitate analysis of the generated spatiotemporal data, we also provide a computational pipeline, which is now referred to as “splat-pipeline” in the manuscript to distinguish from the overall experimental approach, to help interested readers perform analysis, should they wish to use it. We now clarify in the Methods that the “splat-pipeline” computational pipeline provides a Snakefile to initiate database search through Comet and Percolator, then calls Riana and pyTMT. Riana is a standalone software and has been used in our previous publication (Hammond et al. Mol Cell Proteomics 2022), which integrates MS1 chromatogram areas under curves for specific peaks upon SILAC or other isotope labeling, then performs signal processing, filtering, and kinetic curve-fitting. pyTMT is a simple Python tool that returns the TMT channel intensity of each scan, performs isotope purity correction, and appends the channel intensity values to the Percolator output file. The splat-pipeline also performs simple operations to merge the MS1 and MS2 quantification results so that they can be used for downstream analysis, such as using established R packages like pRoLoc and BANDLE. While we provide the “splat-pipeline” for convenience, its use is optional for the overall experimental technique. We anticipate an experienced proteomics researcher will be able to substitute the data processing steps with the database search, post-processing, MS1 quantification, MS2 quantification, kinetic modeling, and spatial proteomics analysis software tools of their choice. We now provide additional clarifications in the Methods section.

Reviewer 4 Comment 3: *“-the isoform-aware rollup of quantitative peptide info into protein info is not fully clear and needs a more elaborate explanation (in the methods section).”*

Response: We have now expanded on the explanation in the Methods section. The text now clarifies that the roll up takes a similar approach to that described for protein inference in the PickedProtein procedure

(Savitski et al. 2015), namely, discard peptides that are mappable to two proteins except when those that are mappable to two isoforms in the same protein. Our approach is more conservative in that we also discard a shared peptide between two isoforms of the same protein if the non-canonical isoform protein possesses its own unique peptide, because we wished to avoid the isoform's potential contribution to the canonical protein's spatial profile.

Reviewer 4 Comment 4: *“-peptide identification is performed using Percolator at 5%FDR, but in another place of the text it is said FDR is 1%, please correct.”*

Response: We apologize for the typo. We indeed thresholded the Percolator FDR at 1%, along with using a two-peptide rule. The text has now been corrected.

Reviewer 4 Comment 5: *“-how does the author calculate protein FDR after peptide rollup to proteins?”*

Response: Our main filtering criteria for inclusion into the spatial data uses peptide-level FDR as our goal is to determine which peptides should be considered in using their ultracentrifugation fraction profiles for subcellular compartment assignment. Protein-level false positives were controlled with a very conservative two-peptide filter requiring two confident peptide identifications for a protein to be included.